# Gravity Waves Perturbed Wind Shears Derived from SABER Temperature Observations

Xiao Liu[1, 2], Jiyao Xu[2, 3], Jia Yue[4, 5], Hanli Liu[6]

[1]Henan Engineering Laboratory for Big Data Statistical Analysis and Optimal Control, School of Mathematics and Information Sciences, Henan Normal University, Xinxiang, 453007, China
[2]State Key Laboratory of Space Weather, Center for Space Science and Applied Research, Chinese Academy of Sciences, Beijing, 100190, China
[3] College of Earth Science, University of the Chinese Academy of Science, Beijing, 100049, China
[4] Catholic University of America, Washington, DC 20064, USA
[5] NASA Goddard Space Flight Center, Greenbelt, MD 20771, USA
[6] High Altitude Observatory, National Center for Atmospheric Research, Boulder, CO 80301, USA

*Correspondence to*:  Jiyao Xu (xujy@nssc.ac.cn)

**Abstract.** Large wind shears around the mesopause region play important roles in atmospheric neutral dynamics and ionospheric electrodynamics. Based on previous observations using sounding rockets, lidars, radars and model simulations, large shears are mainly attributed to gravity waves (GWs) and modulated by tides (Liu, 2017). Based on the dispersion and polarization relations of linear GWs and the SABER temperature data from 2002 to 2019, a method of deriving GW-perturbed wind shears is proposed. The zonal mean GW-perturbed shears have peaks (13-17 $ms^{-1}km^{-1}$) at around the mesopause region, i.e., at z=90-100 km at most latitudes and at z=80-90 km around the cold summer mesopause. This latitude-height pattern is robust over the 18 years and agrees with model simulations. The magnitudes of the GW-perturbed shears exhibit year-to-year variations and agree with the lidar and sounding rocket observations on climatology sense but are 60-70% of the model results in the zonal mean sense. The GW-perturbed shears are hemispheric asymmetric and have strong annual oscillation (AO) at around 80 km (above 92 km) at the northern (southern) middle and high latitudes. At middle to high latitudes, the peaks of AO shift from winter to summer and then to winter again with increasing height. However, these GW-perturbed shears may be overestimated because the GW propagation direction cannot be resolved by the method and may be underestimated due to the observational filter, sampling distance and cutoff criterion of the vertical wavelength of GWs.

## 1 Introduction

In the mesosphere and lower thermosphere (MLT), large horizontal winds and their vertical shears (or more precisely vertical wind shears, short for shears) have been revealed from more than 500 wind profiles observed by sounding rockets (Larsen, 2002; Larsen & Fesen, 2009) and from ground-based lidar and radar observations (Larsen & Fesen, 2009; Oppenheim et al., 2009, 2014; Yue el al., 2010). The large horizontal winds ($\geq$100-200 $ms^{-1}$) and shears ($\geq$40 $ms^{-1}km^{-1}$)

occur in the height range of z~95-115 km at lower and middle latitudes. Using the lidar and falling sphere observations at high latitudes (the Andøya Rocket Range and the ALOMAR observatory at 69.3°N) in July 2002, Fritts et al. (2004) showed large winds and shears at z~85-95 km and ascribed them to strong gravity wave (GWs) activity. Most prominent sources of GWs are convection, orography and jets and fronts in the troposphere as well as spontaneous adjustment and secondary wave generation in the stratosphere (Fritts and Alexander, 2003). Amplitudes and shears increase when the GWs propagate into the lower thermosphere. It is now accepted that the large winds and shears are a common phenomenon in the MLT region (Liu, 2007; 2017). The large winds and shears are associated with tides and GWs and play important roles in forming the middle latitudes sporadic E layers and driving the equatorial electrojet (Mathews 1998; Hysell et al., 2002; Arras et al., 2009; Haldoupis, 2012; Shinagawa et al., 2017; Arras and Wickert, 2018; Yu et al., 2019), in controlling atmospheric stabilities and the propagation of GWs, and in transporting and mixing tracers locally and/or globally (Fritts et al., 2004; Liu 2007, 2017; Yue et al., 2013; Stevens et al., 2014).

Based on the definition of the Richardson number $Ri = N^2/S^2$, the atmosphere is dynamic stable when $Ri > 1/4$ and dynamic unstable when $Ri \leq 1/4$. Here $N^2 = g/\bar{T}\left(d\bar{T}/dz + g/c_p\right)$ is the static stability, $S^2 = (\partial u/dz)^2 + (\partial v/dz)^2$ is the wind shear; $\bar{T}$ is the background temperature; $g$ and $c_p$ are the gravitational acceleration and specific heat for dry air at constant pressure, respectively; $u$ and $v$ are the zonal and meridional winds, respectively. The threshold of dynamic instability is $Ri = 1/4$, which means that the maximum wind shear allowed by the background static stability should be $S = 2N$. According to $S = 2N$ and the simulated temperature, Liu (2007) showed that the maximum wind shears agree well with the observed large wind shears. However, the global scale models (e.g., TIME-GCM: Thermosphere, Ionosphere, Mesosphere, Electrodynamics General Circulation Model) cannot reproduce the observed large winds and wind shears but can increase the amplitudes of winds and wind shears with finer spatial resolutions (Larsen & Fesen, 2009). This indicates that the model resolution and thus tides and small-scale waves (e.g., GWs) are the cause of large winds and shears seen in the observations. Using a local numerical model, Liu et al. (2014a) showed that the nonlinear interactions between GWs and tides can produce large winds and shears in the MLT region. The tidal phases modulate the peak height of large winds and shears. Using the spectral element version of NCAR Whole Atmosphere Community Climate Model (WACCM) with horizontal resolution of ~25 km and vertical resolution of 0.1 scale height, Liu (2017) reproduced the large wind shears, which are in good agreement with observations. Through scale separation, Liu (2017) proposed that small-scale waves (with zonal wavenumber > 6), likely GWs, play a dominant role in producing large shears. The high resolution WACCM can resolve GWs with scales longer than ~100 km. Tidal waves make secondary contribution to the magnitudes of shears but can modulate the shears produced by GWs.

Observations on large winds and shears were made by sounding rocket and ground-based lidar and radar for a very limited number of locations and hence cannot provide a global morphology. High resolution GCM (e.g., WACCM) simulations can provide a global picture of large winds and shears but need be validated through global observations (e.g.,

satellites). Moreover, it is still challenging to study the intra-annual and/or the inter-annual variations of large winds and shears through high resolution GCM simulations due to the computational cost (Liu, 2017, 2019).

Satellite observations provide a good opportunity to study the climatology of global winds. The Wind Imaging Interferometer (WINDII) and the High Resolution Doppler Imager (HRDI) instruments onboard the Upper Atmospheric Research Satellite (UARS) provide global observations of winds in the MLT region (McLandress et al, 1996; Zhang et al., 2007; Shepherd et al., 2012). Combining the winds observed by HRDI and data assimilation system, Swinbank & Ortland (2003) developed a climatology that describes the monthly zonal mean zonal winds from the surface to the upper mesosphere. The TIMED Doppler Interferometer (TIDI) instrument on board the Thermosphere Ionosphere Mesosphere Energetics and Dynamics (TIMED) satellite measures global winds in the MLT region (Killeen et al., 2006; Niciejewski et al., 2006). These observations advanced our knowledge on the global winds in the MLT region. However, the global characteristics of shears are poorly known due to either the limited altitude coverage or the altitude resolution or large noise of the satellite observations. Since large shears play important roles in the atmospheric dynamics in the MLT region and ionospheric E region, and since they are likely caused by GWs (Fritts, 2004; Liu, 2007, 2017; Yue et al., 2010), it should be possible to derive wind shears by combining the GW theory and GWs derived from other observed physical quantities (e.g., temperature).

The focus of this paper is to propose a method of deriving shears from GW analyses of temperature observations. Temperature profiles measured by the Sounding of the Atmosphere using Broadband Emission Radiometry (SABER) instrument (Russell et al., 1999) onboard the TIMED satellite from 2002 to 2019 are used for this study providing a 18-year period. Temperature profiles are remarkably stable until now (Mlynczak et al., 2020). This indicates the stability of the SABER instrument is very high over the recent 18 years of measurements. These profiles cover an altitude range of ~15-110 km and latitude range of 53°S-83°N or 83°S-53°N. The temperature accuracy is 1-3 K from 30 to 80 km and 5-10 K from 90 to 100 km as reported at http://saber.gats-inc.com/website based on Remsburg et al. (2008).

The remainder of this paper is organized as follows. Section 2 presents the method of deriving wind shears induced by GWs, and the uncertainties will be presented. Section 3 presents the comparisons of the GW-perturbed shears with the model and observational results. Then the latitudinal and intra-annual variations of the GW-perturbed shears are presented in Section 4. The limitations of the method and their possible influences on the GW-perturbed shears are discussed in Section 5. A short summary is given in Section 6.

## 2 Method of Deriving GW-perturbed Wind Shears and Validations

### 2.1 Theory of Deriving GW-perturbed Shears

The basic idea of deriving GW-perturbed shears is the linear GW theory, which includes the dispersion and polarization relations of a monochromatic GW. For conservative propagation and without refraction, a linear monochromatic GW can be described as (Fritts & Alexander, 2003),

$$\left(u_j', v_j' \; T_j'/\bar{T}\right) = \left(\tilde{u}_j, \tilde{v}_j, \tilde{T}_j\right) \times e^{i\varphi_j + z/2H},$$  (1)

here, $i = \sqrt{-1}$ is the imaginary unit. The subscript $j$ denotes a monochromatic GW. $u_j'$ and $v_j'$ are the horizontal wind perturbations parallel and perpendicular to the wavevector of the GW, respectively. $T_j'$ and $\bar{T}$ are the perturbation and background temperatures, respectively. $\tilde{u}_j$, $\tilde{v}_j$ and $\tilde{T}_j$ are the amplitudes of $u_j'$, $v_j'$ and $T_j'/\bar{T}$, respectively. $\varphi_j = k_j x + l_j y + m_j z - \omega_j t$ is the phase of GW. $k_j$, $l_j$ and $m_j$ are the wavenumbers in the horizontal $(x, y)$ and vertical $(z)$ directions, respectively. $\omega_j$ and $t$ are the ground-based frequency and time.

Based on the polarization of the monochromatic linear GWs with lower and medium frequencies (Fritts & Rastogi, 1985; Fritts & Alexander, 2003), the relations between $\tilde{u}_j$, $\tilde{v}_j$ and $\tilde{T}_j$ can be derived as (Eckermann et al., 1995; Liou et al., 2003; Gubenko et al., 2008),

$$\tilde{u}_j \approx i\frac{g}{N}\left(1 - f^2/\hat{\omega}_j^2\right)^{-1/2}\tilde{T}_j,$$  (2)

$$\tilde{v}_j = \frac{f}{\hat{\omega}_j}\frac{g}{N}\left(1 - f^2/\hat{\omega}_j^2\right)^{-1/2}\tilde{T}_j.$$  (3)

here $\hat{\omega}_j$ and $f = 2\Omega \sin\phi$ ($\Omega = 7.292 \times 10^{-5}\text{s}^{-1}$, $\phi$ is latitude) are the intrinsic and inertial frequencies, respectively. The wind shear of each monochromatic GW can be written as,

$$\frac{\partial u_j'}{\partial z} = \frac{g}{N}m_j\left(1 - f^2/\hat{\omega}_j^2\right)^{-1/2}\tilde{T}_j e^{i(\varphi_j + \pi)},$$  (4)

$$\frac{\partial v_j'}{\partial z} = \frac{f}{\hat{\omega}_j}\frac{g}{N}m_j\left(1 - f^2/\hat{\omega}_j^2\right)^{-1/2}\tilde{T}_j e^{i(\varphi_j + \pi/2)}.$$  (5)

Here we only consider the GWs propagating or projecting in the along-track direction. In reality, usually spectrum of GWs is observed formed by superposition of several monochromatic GWs. For each monochromatic GW, Eq. (2-5) are valid. Here, we use $u'$ to represent observed GWs, which contains multiple monochromatic GWs (e.g., many $u_j'$). These monochromatic GWs may not propagate in the same direction. However, from two adjacent GW profiles, one cannot get the actual horizontal wavenumber $(k_{j,hr})$ of each monochromatic, but can get the projection of actual wave vector in the along-track direction $(k_{j,ha})$. The inconsistency between $k_{j,hr}$ and $k_{j,ha}$ introduces uncertainties of GW-perturbed shears, which are discussed in Sec. 2.3. Thus, $u'$ can be expressed as the vector sum of the projections of all the actual monochromatic GWs in the along-track direction and formulated as,

$$u' = \sum_j u_j' = \sum_j \tilde{u}_j e^{i\varphi_j} = \frac{g}{N}\sum_j\left[\left(1 - f^2/\hat{\omega}_j^2\right)^{-1/2}\tilde{T}_j e^{i(\varphi_j + \pi/2)}\right].$$  (6)

In the same way, the component $v'$ of an observed GWs can be expressed as,

$$v' = \frac{g}{N}\sum_j\left[\frac{f}{\hat{\omega}_j}\left(1 - f^2/\hat{\omega}_j^2\right)^{-1/2}\tilde{T}_j e^{i\varphi_j}\right].$$  (7)

For the GW-perturbed shears of $u'$ and $v'$, we have

$$\frac{\partial u'}{\partial z} = \frac{\partial}{\partial z}\left(\sum_j u_j'\right) = \sum_j \frac{\partial u_j'}{\partial z} = \frac{g}{N}\sum_j\left[m_j\left(1 - f^2/\hat{\omega}_j^2\right)^{-1/2}\tilde{T}_j e^{i(\varphi_j + \pi)}\right],$$  (8)

$$\frac{\partial v'}{\partial z} = \frac{\partial}{\partial z}\left(\sum_j v'_j\right) = \sum_j \frac{\partial v'_j}{\partial z} = \frac{g}{N}\sum_j\left[\frac{fm_j}{\widehat{\omega}_j}\left(1 - f^2/\widehat{\omega}_j^2\right)^{-1/2}\tilde{T}_j e^{i(\varphi_j + \pi/2)}\right]. \tag{9}$$

Finally, the magnitude of GW-perturbed shear can be written as

$$S = \sqrt{(\partial u'/\partial z)^2 + (\partial v'/\partial z)^2}. \tag{10}$$

Due to the uncertainties in determining the GWs' propagation direction, only the magnitudes of GW-perturbed shears are

analyzed in this work.

## 2.2 GW-perturbed Shear Derived from Synthesized GW Profiles and Validations

To demonstrate the applicability of the theory and the procedure of retrieval the GW-perturbed shears, we construct two synthesized temperature perturbation profiles (shown in Fig. 1a). Each profile is the sum of three monochromatic GWs with three different vertical wavelengths and height dependent amplitudes,

$$\begin{cases} T'_{w1}(z) = \sum_{j=1,3} T'_j(z) = \sum_{j=1,3}\tilde{T}_j \cos\left(m_j z + \varphi_j\right) \\ m_1 = 2\pi/(6\,\text{km}), m_2 = 2\pi/(10\,\text{km}), m_3 = 2\pi/(15\,\text{km}) \\ \varphi_1 = \varphi_2 = \varphi_3 = 0 \\ A_1 = 2\times exp[(z - 70\,\text{km})/(25\,\text{km})] \\ A_2 = 2\times exp[-(z - 60\,\text{km})^2/(20\,\text{km})^2], A_3 = 1 \end{cases} \tag{11}$$

The background temperature $\bar{T}(z) = 240$ K. The black line in Fig. 1a show the $T'_{w1}(z)$ and noted as $T'_{w1}$. The profile $T'_{w2}$ (red line in Fig. 1a) has the same amplitudes and vertical wavenumbers as $T'_{w1}$ but setting $\varphi_1 = \varphi_2 = \varphi_3 = \pi/2$. $T'_{w1}(z)$ and $T'_{w2}(z)$ are used to represent two adjacent SABER measurements. It should be noted that we set the three monochromatic GWs in $T'_{w2}(z)$ having the same phase shift of $\pi/2$ only for the convenience of theoretical representation. In real atmosphere,

GWs with different vertical wavelengths may have different horizontal wavelengths and thus different phase shifts. These phase shifts can be calculated by comparing the phases of the two monochromatic GWs with same vertical wavelength embedded in the two adjacent GW profiles. The phase of each monochromatic GW can be derived through discrete Fourier transformation (in short DWT, Torrence & Compo, 1998). Then the phase shifts can be used to estimate horizontal wavenumber (e.g., Preusse et al., 2002; Ern et al., 2004; Alexander et al., 2008, 2018; Wang and Alexander, 2010). For

example, if we assume the horizontal distance ($\Delta r$) between the two profiles is 300 km, the phase shift of $\Delta\varphi = \pi/2$ indicates the horizontal wavenumber $k_h = \Delta\varphi/\Delta r = (\pi/2)/(300\,\text{km}) = 2\pi/(1200\,\text{km})$. For the lower and medium frequency GWs, the dispersion relation can be simplified as (Fritts & Alexander, 2003),

$$\widehat{\omega}_j^2 = N^2\frac{k_{j,h}^2}{m_j^2} + f^2. \tag{12}$$

After we get the horizontal wavenumber $k_{j,h}$ from a GW profiles pair, then the intrinsic frequency for $T'_j(z)$ can be

calculated by Eq. (12). It should be noted that the $k_{j,h}$ is the projection of the actual GW's horizontal wavenumber in the along-track direction and is underestimated. This underestimates the intrinsic frequency and thus overestimates the wind and shears based on Eq. (6-9) and will be discussed detail in Sec. 2.3. Using Eq. (2-5) and the prescribed amplitude and vertical wavenumber, which will be determined from the satellite observation and described below, for the individual

monochromatic GW, we can get $u'_j$ and $v'_j$ and their shears. Then, according to Eq. (6-9), the vector sum of the three monochromatic GW-perturbed wind profiles can be obtained and are shown as black lines in Fig. 1b for zonal wind and 1(c) for meridional wind. Here we assume the latitude is at 30ºN, which is a typical latitude at middle latitudes. The corresponding shears are also shown as black lines in Fig. 1d-e. Since the amplitude, vertical and horizontal wavenumbers of

5 GW profile pairs are prescribed and are not derived through DWT, which is a key step of the spectral decomposition method described below, the retrieved winds and shears are referred to as theoretical results. These theoretical results can be used to measure the results obtained from the spectral decomposition method proposed below. Here we assume that $u'_j$ and $v'_j$ are the winds along and cross the orbit track directions, respectively, and may not coincide with the eastward and northward directions, but are the winds along and cross the orbit track directions, respectively. The along-track direction is points from

10 the location of $T'_{w1}(z)$ to that of $T'_{w2}(z)$. The cross track direction is 90° counterclockwise from the zonal direction. However, GWs may not propagate only in the along-track direction. The assumption will introduce uncertainties and will be discussed below.

Now we describe the method of retrieving winds and wind shears induced by GW profile pairs, whose amplitudes, vertical and horizontal wavenumbers are not prescribed but should be evaluated. We named this method as "spectral

decomposition method" since the principle ideas are: (1) decomposing an observed GW profile into multiple monochromatic waves; (2) applying linear GW theory on each monochromatic wave to get monochromatic wind and shear of each wave component; (3) finding the vector sum of monochromatic winds and shears to get the wind perturbations and shears induced by the observed GW. The detailed application of this method is described as the following three steps.

The first step is to evaluate the amplitude and vertical wavenumbers of each GW profile by the method of DWT such

that we can get the height dependent amplitudes and vertical wavelengths as well as the phase shifts. For the GW profiles of $T'_{w1}(z)$ and $T'_{w2}(z)$, then at each height and vertical wavelength ($\lambda_z = 2\pi/m$), their DWT are $\hat{T}_{w1}(z, \lambda_z)$ and $\hat{T}_{w2}(z, \lambda_z)$. Then, their cospectrum $C_{1,2}$ is computed,

$$C_{1,2} = \hat{T}_{w1}\hat{T}^*_{w2} = \tilde{T}_{w2}\tilde{T}_{w2} \exp(i\Delta\varphi_{1,2}), \tag{13}$$

Here, $\tilde{T}_{w1} = |\hat{T}_{w1}|$ and $\tilde{T}_{w2} = |\hat{T}_{w2}|$ are the amplitudes of $T'_{w1}(z)$ and $T'_{w2}(z)$, $\hat{T}^*_{w2}$ is the complex conjugation of $\hat{T}_{w2}$. The

25 phase shift is calculated by $\Delta\varphi_{1,2} = \tan^{-1}[\text{Im}(C_{1,2})/\text{Re}(C_{1,2})]$. When performing DWT, we restrict the vertical wavelength ranging from ~5 km to ~30 km for a vertical extent of 90 km, which is the height coverage (18-108 km) of the SABER temperature profiles.

The second step is to evaluate the horizontal wavenumber through the phase shift between two adjacent GW profiles. According to the distance between the two adjacent profiles $\Delta r = 300$ km and phase shift, we can get $k_h = \Delta\varphi_{1,2}/\Delta r =$

$(\Delta\varphi_{1,2})/(300 \text{ km})$ (Ern et al., 2004, Alexander, 2008). Then the intrinsic frequency $\hat{\omega}$ for the component of $m$ can be calculated by Eq. (12). Here we note that the horizontal wavelengths of GWs in real atmosphere may be shorter than $2\Delta r$, only GWs with horizontal wavelengths longer than $2\Delta r$ are considered due to the sampling distances and the limb scanning mode of the SABER instrument (Preusse et al., 2002; Ern et al., 2004).

The third step is to calculate the GW-perturbed wind (shown as red dashed lines in Fig. 1b-c) and shears perturbations (red dashed lines in Fig. 1d-e) by Eq. (6-9). Then we can get the amplitudes of wind shears, which are the modules of $\partial u'/\partial z$ and $\partial v'/\partial z$, respectively. Finally, the GW-perturbed shear ($S$) can be calculated by Eq. (10).

A brief summary of the results from the spectral decomposition method (red) and theory (black) are shown in Fig. 1. From Fig. 1, we can see that the GW-perturbed winds and shears derived from spectral decomposition method (red dashed lines) agree well with the theoretical results (black solid lines) below 100 km. The bad consistencies occur at around the upper boundary. Thus, we will focus on the results in $z$=30-100 km in the following analysis.

## 2.3 GW-perturbed Shear Derived from SABER GW profiles and Uncertainties

A key step to derive the GW-perturbed shears is the extraction of GWs from the SABER temperature profile. The extraction methods of GWs from satellite data have been developed by Fetzer and Gille (1994) and improved greatly since (e.g., Preusse et al., 2002; Ern et al., 2004, 2011, 2018; Chen et al., 2019; Alexander et al., 2008, 2018, Wang & Alexander, 2010; Alexander, 2015). We have developed a similar method in our previous studies (Liu et al., 2014b, 2017, 2019), which is summarized here. First, the daily SABER temperature profiles $T(z)$ in a latitude band of 5° are selected. Second, at each altitude, these selected data are fitted by harmonics with zonal wavenumbers ranging from 0 to 6, which are mainly planetary waves and non-migrating tides and are removed from $T(z)$ to get the residual temperature $T_r(z)$. The component of wavenumber 0 is considered as the zonal mean temperature $\bar{T}(z)$. Due to the slowing precessing of the SABER measurement (a full cycle is about 120 days), migrating tides will appear as stationary zonal wave patterns if data from ascending and descending nodes are taken separately. The above two steps are applied on both the ascending and descending nodes, respectively, such that it minimizes the influences of migrating tides on the residual temperature $T_r(z)$ and thus GW profiles (Preusse et al., 2002, Ern et al., 2004, 2018). Third, DWT (Eq. (13)) is applied on each residual profile to get its wavelet transform. When applying DWT, we restrict the vertical wavelengths in the range from ~5 km to ~30 km. From these monochromatic waves, we reconstruct a GW profile $T'_w(z)$.

A GW profiles pair is defined as the two adjacent SABER GW profiles, whose along-track distance is less than 400 km. The 400 km criterion is fulfilled only for the short-distance pairs for SABER measurement (Fig. 1 of Ern et al. 2011). Then from a given GW profiles pair, we can get pairs of GW-perturbed winds and shears by the spectral decomposition method. Figure 2 shows the procedure of derivation winds and shears from a GW profiles pair at (42.55ºN, 56.48ºE) and (44.97ºN, 56.34ºE) on 1$^{st}$ January 2018. Figures 2(b) and 2(c) show the GW-perturbed winds in the along and cross track directions, respectively, through Eq. (6) and (7). Figures 2(d) and 2(e) show the GW-perturbed shears in the along and cross track directions, respectively, through Eq. (8) and (9). Figure 2(f) shows the GW-perturbed shears calculated by Eq. (10). From Fig. 2, we can see that the peak heights of the temperature, wind and shear are not at the same height due to their phase differences shown by the polarization relations. The winds and shears in the along-track direction are larger than those in the cross-track direction due to the factor of $f/\hat{\omega} < 1$ in Eq. (3).

The horizontal wavenumber $k_h$, which is derived from GW profiles pair, is in the along-track direction. It is smaller than the real wavenumbers since GWs may not propagate exactly in the along-track direction. This will induce uncertainties in deriving the GW-perturbed shears since the GW propagation direction cannot be determined from a GW profiles pair (Ern et al., 2004). The uncertainties of the GW-perturbed shears are estimated in below.

The relation of the horizontal wavenumber in the along-track direction ($k_{ha}$) and that of in the GW propagation direction ($k_{hr}$) can be expressed as $k_{ha} = k_{hr} \cos \alpha$. The subscripts "a" and "r" denote the along-track and real physical quantities, respectively. The angle ($\alpha$) between the along-track direction and GW propagation direction can vary from 0 º to 360º. Here, the angle $\alpha$ is restricted in the range of 0-90º. This restriction maps the angle of four quadrants into the first quadrant since only the magnitude of $k_{ha}$ and $k_{hr}$ are considered here. If $\alpha \neq 0$, this will induce uncertainties of intrinsic

frequencies ($\widehat{\omega}$) and thus GW-perturbed shear ($S$). Here the uncertainty of $S$ (noted by $Se$) is defined as ratio between the along-track $S$ (noted by $S_a$) and real $S$ (noted by $S_r$). According to Eq. (10) and (4-5), we get,

$$S_e = \frac{S_a}{S_r} = \frac{\widetilde{u}_a'}{\widetilde{u}_r'} \sqrt{\frac{1+f^2/\widehat{\omega}_a^2}{1+f^2/\widehat{\omega}_r^2}} = \sqrt{\frac{(N^2 k_{hr}^2/m^2)+2f^2/\cos^2 \alpha}{(N^2 k_{hr}^2/m^2)+2f^2}}. \qquad (14)$$

Equation (14) shows that if $\alpha = 0$ or $f = 0$, then $S_e = 1$ and the GW-perturbed shear is accurate. if $\alpha \neq 0$ and $f = 2\Omega \sin \phi \neq 0$, then $Se > 1$ and the total wind shear is overestimated.

Figure 3 shows the dependencies of $S_e$ on horizontal ($\lambda_h$) and vertical wavelengths ($\lambda_z$), the angle $\alpha$ and latitude $\phi$. From Fig. 3a, we can see that $S_e$ increases with the increasing $\lambda_h$ and the decreasing $\lambda_z$. Figure 3(b) shows that $S_e$ increases with the increasing $\lambda_h$ and the increasing $\phi$. Figure 3(c) shows that $S_e$ increases with the increasing $\lambda_h$ and the increasing $\alpha$. Comparisons the relative importance of $\lambda_z$, $\lambda_h$, $\alpha$, and $\phi$ in changing $S_e$, the angle $\alpha$ is the most important one. If we assume that GW propagates in an arbitrary direction, $S_e$ is less than 1.2 (indicated by a red contour line) for a large fraction of GWs

at $\phi = 30°$. The fractions of GWs, which are overestimated by 20%, increase with the increasing latitudes. This indicates that the method proposed here can be used to estimate GW-perturbed shears even though the wave propagation direction cannot be determined from a GW profiles pair. When we analysed the derived total wind shears, the overestimation at high latitudes should be considered.

     The large winds and shears in the MLT region, where both tides and GWs reach large amplitudes and may interact

nonlinearly (Fritts and Vincent, 1987; Li et al., 2009; Liu et al., 2014a) and break (Fritts et al., 2004; Liu and Vadas, 2013; Vadas and Liu, 2013). The large winds may rotate with height and act as critical levels, which filter out a broad spectral range of GWs. These filtered GWs may break and deposit their momenta in the background atmosphere, which create body force to general secondary GWs. Moreover, the vertical wavelengths of these filtered GWs change rapidly with height and in the nonlinear regime. These nonlinear GWs may also produce large winds and shears around the mesopause regions.

However, these nonlinear GWs cannot be described by linear GW theory proposed here. Thus, the GW-perturbed shears underestimate the actual shears in the MLT region due to the unrepresented nonlinear GWs.

**3 Comparisons GW-perturbed Shears with Model and Observational Results**

Due to the uncertainties, the GW-perturbed shears will be further examined by comparing with models and observational results. According to the wind shears during 1-10 July simulated by WACCM (Liu, 2017) and observations, we take the GW-perturbed shears during May-August of 2018 as an example for the purpose of comparison. The longer date coverage is chosen to include data from both north-viewing and south-viewing yaw periods and hence to cover a wider latitude range. Figure 5 shows the zonal mean and standard deviations (stds) of GW-perturbed shears and the top 10% largest shears during the three periods of 2018. The latitude band used for calculating zonal means has a width of 5° with overlap of 2.5°. The latitude coverage is from 52.5°S-82.5°N or 82.5°S-52.5°N due to the yaw cycle of the SABER measurement. The three periods have centers in July and extend one or two months, such that the results can illustrate the main features during summer.

Three main features can be found in the zonal mean $S$ shown in Fig. 4. Firstly, the maxima of $S$ are ~13-17 ms$^{-1}$km$^{-1}$ with stds of ~9-13 ms$^{-1}$km$^{-1}$ at z~90-100 km (around the mesopause) at all latitudes of the three different time intervals. Especially, the GW-perturbed shears reach their maxima at around 70°S. The maxima $S$ are ~10-13 ms$^{-1}$km$^{-1}$ with std of ~8-10 ms$^{-1}$km$^{-1}$ at $z$~80-90 km at latitudes higher than 40°N, where is near the summer mesopause (cf. contour lines of the zonal mean temperature). Secondly, the zonal mean shears have similar latitude-height patterns during the three different time intervals although they are averaged over different time interval lengths (e.g., 31 days during 01 July-31 July in 2018 and 62 days during 29 June-30 August in 2018). The same is true during the intervals of 06 May-27 June and 29 June-30 August in 2018, respectively, at latitudes of 52.5°S-52.5°N. The GW-perturbed shears during the continuous two yaw cycles (06 May-27 June and 29 June-30 August) exhibit a smooth extension from 52.5°S-52.5°N to higher latitudes, respectively. This shows GW-perturbed shears are a common phenomenon around the mesopause region (Larson, 2002; Fritts et al., 2004; Larsen & Fesen, 2009; Yue el al., 2010; Liu, 2007, 2017). Thirdly, the top 10% largest $S$ reach maxima of ~30 ms$^{-1}$km$^{-1}$ around the mesopause. In general, at each height, the GW-perturbed shears reach their minima at lower latitudes and reach their maxima at high latitudes.

**3.1 Comparisons with Model Results**

The latitude-height patterns of $S$ derived here agree well those simulated by WACCM during 1-10 July (Liu, 2017). Specifically, the GW-perturbed shears derived here have peaks at ~$z$=80-90 km at latitudes higher than 50°N during 06 May-27-June in 2018. Moreover, there is another peak at ~$z$=90-100 km of the southern high latitudes during 29 June-30 August in 2018. These peaks agree with the WACCM simulation results that the large shears have peaks at around the mesopause (Xu et al., 2007; Fig. 2a of Liu (2017)). However, the shear peaks derived here are at a slightly lower height than the WACCM results (Liu, 2017). The GW-perturbed shears might be influenced by temperature uncertainties of SABER measurements, which are much larger at around the cold summer mesopause (Remsberg et al., 2008). We removed waves

with vertical wavelengths shorter than 5 km, to minimize the noise introduced by uncertainties of SABER measurements (Ern et al., 2011, 2018).

Compared to the WACCM simulation results, the differences pertain to the magnitudes of the zonal mean $S$ and specifically: (1) the maxima of the zonal mean $S$. Figure 2(a) of Liu (2017) showed that the maxima of the zonal mean $S$ are 20–40 ms$^{-1}$km$^{-1}$ near the mesopause at latitudes higher than 50°N/S and are ~20-25 ms$^{-1}$km$^{-1}$ at latitudes lower than 50°N/S. This indicates that the GW-perturbed zonal mean $S$ derived here are about 70% smaller than those of the WACCM simulation results. (2) the top 10% largest $S$. The simulated top 10% largest $S$ (e.g., the minima of the total 10% largest $S$) are larger than 50 ms$^{-1}$km$^{-1}$ at high latitudes and 35 ms$^{-1}$km$^{-1}$ over the equator (Fig. 5 of Liu, 2017). Thus, the top 10% largest $S$ derived here (with maxima of ~30 ms$^{-1}$km$^{-1}$) is about 60% of the simulated results. (3) the stds. The simulated stds reach their maxima of ~1.4 times of the shears (Fig. 2b of Liu, 2017). The ratio of 1.4 indicates that the maxima of the simulated std is about 40 ms$^{-1}$km$^{-1}$, which is larger than those derived from observations (~11-13 ms$^{-1}$km$^{-1}$, see the middle row of Fig. 4) at latitudes lower than 50°N/S. (4) the structures of $S$. The structures in Fig. 2b of Liu (2017) have much finer scales than those in Fig. 4. The different spatial scale coverages, which will be discussed in Sect. 5, might be responsible for the smaller values of wind shears derived from observations.

In general, the GW-perturbed shears derived here can reproduce the latitude-height pattern and 60-70% of the simulated shear magnitudes in the zonal mean sense.

## 3.2 Comparisons with Observational Results

To compare the GW-perturbed shears derived here and those observed by ground-based lidar and sounding rockets (Larsen, 2002; Yue et al., 2010), we show in Fig. 5 the profiles of $S$ and their zonal means as well as the top 10% and 1% largest $S$ during January and July at around 40°N and Equator. The January and July are representative months for winter and summer, respectively. The latitudes of Equator and 40°N may be representative for low and middle latitudes, where sounding rocket measurements were performed (Fig. 1 of Larsen, 2002; Larsen & Fesen, 2009). The 40°N is near the latitude of 41°N, where the Colorado State University (CSU) Na lidar observations were performed (Yue et al., 2010).

Figure 5 shows that the magnitudes of the GW-perturbed $S$ profiles increase with height and can be larger than 40 ms$^{-1}$km$^{-1}$ above 90 km (80 km) during January (July). The height variations and magnitudes of GW-perturbed $S$ profiles derived here compare well with the over 400 chemical tracer measurements by sounding rocket (Fig. 10 of Larsen (2002)) below 100 km. Moreover, the height variations and magnitudes of GW-perturbed $S$ profiles derived here compare well with the CSU lidar observations below 100 km. Specifically, the magnitudes of $S$ profiles observed by CSU lidar increase with height and have maxima of ~40 ms$^{-1}$km$^{-1}$ (Fig. 1c-d of Yue et al. (2010)). This agrees well with the $S$ profiles shown in Fig. 5. After averaging the shears observed by the CSU lidar during summer and winter months, the zonal mean shears are ~12-17 ms$^{-1}$km$^{-1}$ with stds of ~10 ms$^{-1}$km$^{-1}$ (Fig. 11c of Yue et al. (2010)). They are slightly larger than the magnitudes of 10-13 ms$^{-1}$km$^{-1}$ derived here (the right column of Fig. 5). The zonal mean shears (the right column and upper row of Fig. 5) derived

here have similar magnitudes during January and July above 90 km. However, the mean shears observed by the CSU lidar the have similar magnitudes in winter and summer at ~$z$=80-105 km. We note that the magnitudes of $S$ derived here are larger in July than in January at Equator. This might be related to the intra-annual variations of $S$ and will be studied in Sect. 4.

A short summary of the above comparisons is below. The GW-perturbed shears derived from SABER observations agree with the previous observations and model results in general. This provides a global climatology of large shears around the mesopause region partially based on observations. The magnitudes of $S$ derived from SABER observations are similar to those observed by lidar and sounding rocket but are about 60-70% of the high resolution WACCM results in the zonal mean sense. The difference probably comes from (1) the coarse horizontal samplings ~250-350 km of satellite observations; (2) only GWs with $\lambda_z \geq 5$ km and $\lambda_h \geq 2\Delta r$ ($\Delta r$ is the distance between the two GW profiles in a pair) used to derived shears and (3) the observational filter should underestimate the amplitudes of GWs, especially for GWs with shorter vertical wavelengths (Ern et al., 2018), and thus underestimate the GW-perturbed shears. This will be further discussed in Section 5.

## 4 Climatology of GW-perturbed Shears

With the advantage of 18-year SABER observations, the climatology of the GW-perturbed shears can be explored on the aspects of their latitudinal variations and intra-annual variations. The four seasons in the northern hemisphere (NH) are: spring (MAM: March, April, and May), summer (JJA: June, July, and August), autumn (SON: September, October, and November), winter (DJF: December, January, and February).

### 4.1 Latitudes Variations of GW-perturbed Shears

Figure 6 shows the latitude-height contours of the zonal mean and stds of the GW-perturbed shears and the top 10% largest shears during four composite seasons. Each composite season is made up by the superpositions of the corresponding seasons from 2002 to 2019. The numbers of profiles used to derive GW-perturbed wind shears in each season (the fourth row of Fig. 6) have peaks around 50°N/S due to the changes of ascending and descending nodes. The sharp changes of the profile numbers around 50°N/S might induce the discontinuity of the latitudinal variations of GW-perturbed shears. By further examination of the discontinuities in Fig. 6, we find that the zonal means exhibit more obvious discontinuities at around 50°N (50°S) than those at around 50°S (50°N) during spring and summer (autumn and winter). This is because there are fewer samplings (the fourth row of Fig. 6) at around 50°N (50°S) than those at around 50°S (50°N) during spring and summer (autumn and winter).

The hemispheric asymmetry of the sampling is induced by the inconsistency of the date coverages of yaw cycle and season. Consequently, we show in Fig. 7 the zonal means and stds of GW-perturbed shears and the top 10% largest shears during six composite yaw cycles. The composite yaw cycle is the superposition of all the yaw cycles, which have nearly identical date coverage relative to the beginning of each calendar year. For example, the first yaw cycle of each year covers 0125-0318 in 2002, 0116-0318 in 2003, …, and 20181228-20190226. These dates are mainly in January and February, with

a few extending to March and December. We label "1228-0318" on the top of the first column of Fig. 7 to note the all the dates covered by the first yaw cycle. The results in the other five composite yaw cycles are shown in a same manner. The continuous two composite yaw cycles may have overlaps, with the longest overlap time of about 20 days. Thus, the composite yaw cycle can represent the results during two months around the center date of each composite yaw cycle.

From Fig. 6, we can see that the zonal means of the GW-perturbed shears (the first row) increase with the increasing height and latitude in general. The peaks are ~10-15 ms$^{-1}$km$^{-1}$ above 90 km at latitudes of 82.5°S-50°N (50°S-82.5°N) during spring and summer (autumn and winter). Moreover, the wind shears have peaks at a lower height ($z$~80-90 km) and at latitudes of 82.5°S-50°S (50°N-82.5°N) during autumn and winter (spring and summer). These lower height peaks during spring and autumn (highlighted by blue rectangles) are weaker than those during summer and winter (highlighted by red

rectangles). Comparing with Fig. 7 (the second and third columns, 0228-0512, 0502-0715), we find that the weak peak at $z$~80-90 km during spring is contributed from the wind shears during May (highlighted by a red rectangle in Fig. 7 during the yaw cycle of 0502-0715) since there is no peak at similar location during the yaw cycle of 0228-0512. The same is true during autumn, when the weak peak at z~80-90 km is contributed from the wind shears during November (highlighted by a red rectangle in Fig. 7 during the yaw cycle 1031-114) since there is no peak at similar location during the yaw cycle of

1228-0318. The stronger peaks during summer and winter in Fig. 6 are contributed from those during the yaw cycles of 0502-0715 and 1031-0114, respectively. The stronger peaks above 90 km and at ~$z$=80-90 km (marked by red rectangles) are both at around the mesopause as referred to the zonal mean temperature (contour lines in the second rows of Figs. 6 and 7).

      The std and the top 10% largest shears, which are shown in the second and third rows of Figs. 6 and 7, respectively,

have similar patterns as that of zonal mean shears. The maxima of the std and the top 10% largest shears are, respectively, ~12 ms$^{-1}$km$^{-1}$ and ~30 ms$^{-1}$km$^{-1}$, which are slightly less than that shown in Fig. 4. This is because the sampling profiles in Figs. 6 and 7 (composite season or yaw cycle over 18-year) are much larger than those in Fig. 4 (only one yaw cycle in one year).

      Since the patterns of zonal mean and stds of the GW-perturbed shears and the top 10% largest shears are similar to each

25 other (as shown in Figs. 4, 6, 7), only the zonal mean shears during each summer from 2002 to 2019 are shown in Fig. 8. It can be seen that the latitude-height distributions of GW-perturbed shears, including the peaks at lower heights (around the mesopause region) of high latitudes, are similar to the 18-year's mean results shown in Figs. 6 and 7. However, the GW-perturbed shear magnitudes (shown in Fig. 8) exhibit year-to-year variations. For example, at the SH high latitudes, the wind shears above 90 km are strongest during 2008 and 2019 and weakest during 2002. At the NH high latitudes, the GW-

30 perturbed shears at ~$z$=85-95 km vary with year more greatly and have smaller values, as compared to those at around 80 km.

**4.2 Intra-annual Oscillations of GW-perturbed Shears**

Since the GW-perturbed shears are prominent around the mesopause region, their intra-annual oscillations will be studied at ~$z$=60-100 km. Figure 9 shows the monthly zonal mean GW-perturbed shears at four latitudes bands of the NH and SH from

2002 to 2019. A general feature of time-height variations GW-perturbed shears are the annual (AO) and semiannual oscillations (SAO). To quantify the exact amplitudes and phases of AO and SAO, harmonic fitting is applied on the GW-perturbed shears. The fitting function has periods both AO and SAO. Figure 10 shows the amplitudes and phases of both AO and SAO at four latitude bands of northern and southern hemispheres (SH).

At 50°N/S (the first row of Fig. 9 and the first column of Fig. 10), the GW-perturbed shears exhibit different height dependencies of AO and SAO. At 50°N, both AO and SAO reach their maxima at 80 km, while SAO has another peak at 97 km. At $z$=75-92 km, the AO is dominant and peaks in June. Below 75 km and above 92 km, the AO and SAO are almost equal partitioned. At 50°S, both AO and SAO reach their maxima at ~81 km, while AO has another peak at 98 km. Above 92 km and below 68 km, the AO is dominant and has phase in June. At $z$=80-90 km, the AO and SAO are almost equal

partitioned and have peaks in December and June, respectively. The amplitudes of SAO at 50°N/S have similar amplitudes and height variations. However, the amplitude of AO at 50°N is smaller than that at 50°S. This makes the GW-perturbed shears hemispherical asymmetric.

     It should be noted that the phase of AO at 50°N shifts from December at ~65 km to June at ~75 km and then shifts from June at ~88 km to December at ~100 km. Whereas, the phase of AO at 50°S shifts from June at ~70 km to December at ~77

15   km and then shifts from December at ~85 km to June at ~65 km. In summary, in each hemisphere, the phase of AO shifts from winter below ~65 km (~70 km) to summer at ~$z$=75-88 km (~$z$=77-85 km) and then shift to winter again above 95 km at 50°N (50°S), respectively.

     At 35°N/S (the second row of Fig. 9 and the second column of Fig. 10), the GW-perturbed shears exhibit both AO and SAO. At 35°N, the amplitudes of AO and SAO vary with height in a similar pattern as those at 50°N but have smaller values.

The phases of AO and SAO are also in June when their amplitudes are prominent at $z$=75-92 km and then shift to winter above ~92 km. At 35°S, the amplitude and phase of AO vary with height in a similar pattern as those at 50°S. The amplitude of SAO is dominant below 90 km with peak in June.

     At 20°N (the third row of Fig. 9 and the third column of Fig. 10), the amplitudes of AO and SAO exhibit similar height variations as those at 50°N and 35°N but have smaller values. The AO and SAO reach their peaks at around 81 km in June

and have values of ~0.7 ms$^{-1}$km$^{-1}$ and ~0.5 ms$^{-1}$km$^{-1}$, respectively. At 20°S, the SAO is in the dominant position and has peaks in June at ~92 km and ~67 km. At 5°N/S and at $z$=85-98 km, the AO is in the dominant position and has peak shifting from July to March. At ~$z$=71-80, the SAO is in the dominant position and has peak shifting from March to January.

     The AO and SAO of the GW-perturbed shears are summaries as below. The amplitudes of AO have peaks at around 80 km and decrease with the decreasing latitudes. The phases of AO shift from winter to summer and then to winter again with

the increasing height. The amplitudes of SAO decrease with the decreasing latitudes. The phases of SAO are in May and June when the SAOs reach their peaks at 50°N/S, 35°N/S and 20°N/S. At 5°N/S, the SAOs shift their phase from March to January in their peak height. The AO and SAO are hemispheric asymmetry. At ~$z$=75-90 km, the AOs (SAOs) are in the dominant position at latitudes higher than 20°N (20°S). Above ~90 km, the AO and SAO are almost equal partitioned at 50°N and 35°N, whereas, the AO is in the dominant position at 50°S and 35°S.

The GW-perturbed shears, which is derived from the projection of actual GWs in the along-track direction, are overestimated as compared to the actual GW-perturbed shears. The extent of overestimation depends the actual GW propagations, which have seasonal and latitudinal preferences. Thus the seasonal and latitudinal variations of GW-perturbed shears may be influenced by the preferred GWs propagation directions to some extent. Comparing with the AO and SAO in the GW square temperature amplitude (GWSTA) and absolute momentum flux (GWMF) presented by Chen et al. (2019), we find that the AO and SAO of GW-perturbed shears agree with GWSTA and GWMF on the aspects of phase shifts and hemispheric asymmetry (Fig. 2 and 3 of Chen et al., 2019). However, the heights at which phase shifts occur are different. One reason is that the AO and SAO in the background temperature and static stabilities (Liu et al., 2020). The other reason is that the GWMF is inverse proportion to the vertical wavenumber $m$, while the GW-perturbed shear is proportion to the $m$. The resulting height of the phase shift of GWMF at a lower height than that of GW-perturbed shears since $m$ increases with height below z=90 km (Ern et al. (2018).

## 5 Discussions

To determine the horizontal wavenumbers in the zonal and meridional directions, at least three profiles should be sampled at different locations of the same wave (Wang & Alexander, 2010, Alexander, 2015, Schmidt et al., 2016; Alexander et al., 2018). For the SABER measurement, there are 15 orbits in the ascending and descending nodes, respectively. The nearest distance between two orbits is about 24°, which is much longer than the horizontal wavelengths of most of GWs. This limits our ability to deduce the zonal and meridional horizontal wavenumbers and leads to the uncertainties in deriving the GW-perturbed shears. The horizontal wavenumber derived from a GW profiles pair is in the along-track direction. It is in general smaller than the horizontal wavenumbers of GWs in reality. The angle ($\alpha$) between the along-track direction and real GW propagation direction is the dominant source of the uncertainties. This will overestimate the GW-perturbed shears as shown by Eq. (14) and Fig. 3. According to Eq. (14), the influences of $\alpha$ on $S_e$ increase with the increasing latitudes due to increasing inertial frequency $f$. For the extreme case, $S_e = 1$ for any $\alpha$ since $f = 0$ over Equator. On the other hand, at the high latitudes, GWs propagate mostly in the zonal direction since their sources are mainly jet/front, topography (Fritts & Alexander, 2003; Plougonven & Zhang, 2014). Fortunately, the SABER orbit track intersects with zonal direction at a smaller angle at high latitudes than that at lower latitudes due to the changes of ascending/descending nodes. This reduces the uncertainties of the GW-perturbed shears for the zonally propagating waves. Thus, the uncertainties might be smaller than 1.2 as shown in Fig. 3c.

Even with an over-estimation of 1.2, the GW-perturbed shears derived here are smaller than those of high-resolution model simulations by 60-70% (Liu, 2017). The smaller GW-perturbed shears might be induced by the following two reasons: (1) the coarse horizontal samplings of satellite observations and (2) only GWs with $\lambda_h \geq 2\Delta r$ (Nyquist limit) and $\lambda_z \geq 5$ km used to derived shears. The reason for (1) is that the latitude-height variations of stds in Figs. 4, 6, 7 are smoother than those in Fig. 2b of Liu (2017). The small-scale variations in Fig. 2b of Liu (2017) might be smoothed out due to the observational filter of SABER limb sounding pattern (Preusse et al., 2002; Ern et al., 2018).

According to the SABER sampling (Fig. 1 of Ern et al. 2011), the sampling distance of a GW profiles pair $\Delta r \sim 250 - 350$ km limits the resolved GWs with $\lambda_h \geq 500 - 700$ km. Whereas, the horizontal resolution of WACCM is about 25 km (Liu, 2017), this can resolve GWs with $\lambda_h \geq 50$ km according to Nyquist limit, though waves with $\lambda_h \leq 200$ km are excessively damped by numerical diffusion in WACCM (Liu et al., 2014c). The longer sampling distances miss the GWs with shorter horizonal wavelengths, which might also contribute the GW-perturbed shears and thus reduces the magnitudes of GW-perturbed shears. The influence of the horizontal sampling distances on wind shears can be further confirmed by the simulation results presented by Shinagawa et al. (2017), who showed that the longitude-latitude distributions of the zonal wind shears have peak values of 16-18 ms$^{-1}$km$^{-1}$ at 100 km during summer and winter (Figs. 6 and 7 of their paper). Certainly, the zonal mean of the zonal wind shears should be much smaller than 16-18 ms$^{-1}$km$^{-1}$ at 100 km. This magnitude is also smaller than the simulation results presented by Liu (2017). The smaller magnitude of the wind shears might result from the different resolution used by their models. The Ground-to-topside model of Atmosphere and Ionosphere for Aeronomy (GAIA) used by Shinagawa et al. (2017) has a grid size of 2.8° longitude by 2.8° latitude horizontally and 0.2 scale height vertically. Whereas, the WACCAM used by Liu et al. (2014c) and Liu (2017) has a quasi-uniform horizontal resolution of~25 km, and a 0.1 scale height vertically.

For reason (2), the cutoff criterion of $\lambda_z \geq 5$ km is used here to get more reliable GW profile through DWT. This cutoff criterion is related to the vertical resolution of the SABER measurement and is the same as that used by Ern et al. (2018) to remove the artificial oscillations. To test the influences of the cutoff criterions on the GW-perturbed shears, we perform a same procedure as that described in Sect. 2 and 3 but relax the cutoff criterion to $\lambda_z \geq 3$ km. The GW-perturbed shears derived with cutoff criterion of $\lambda_z \geq 3$ km are shown in Fig. 11. It can be seen that the latitude-height patterns of the GW-perturbed shears are the same as those shown in Fig. 4. However, the maxima of the zonal mean GW-perturbed shears increase from 12-17 ms$^{-1}$km$^{-1}$ for $\lambda_z \geq 5$ km to 18-24 ms$^{-1}$km$^{-1}$ for $\lambda_z \geq 3$ km. This illustrates that magnitudes of the GW-perturbed shears increase with the decreasing cutoff vertical wavelengths.

The uncertainties in the theory presented in Sec.2 arise from the fact that we assume $f/\widehat{\omega}_j \neq 0$ except at the equator. For the mid-frequency GWs ($\widehat{\omega}_j \gg f$ or $f/\widehat{\omega}_j \approx 0$), we get

$$\frac{\partial u'_j}{\partial z} \approx m_j \frac{g}{N} \tilde{T}_j e^{i(\varphi_j + \pi)}, \tag{15}$$

$$\frac{\partial v'_j}{\partial z} = 0. \tag{16}$$

The GW-perturbed shears are contributed only by $\partial u'_j/\partial z$. Then $S$ derived under the mid-frequency assumption of $f/\widehat{\omega}_j \approx 0$ is less than that derived under the assumption of $f/\widehat{\omega}_j \neq 0$ by a factor of

$$R = \left[ \left(1 - f^2/\widehat{\omega}_j^2\right) / \left(1 + f^2/\widehat{\omega}_j^2\right) \right]^{1/2}. \tag{17}$$

To fully explore the differences of magnitudes of $S$ derived under the assumptions of $f/\widehat{\omega}_j \neq 0$ and $f/\widehat{\omega}_j \approx 0$, we show the GW-perturbed shears for $f/\widehat{\omega}_j \approx 0$ in a same manner as those in Fig. (4-9). Such that we can judge that whether the assumptions which need to be made. In a same manner as Fig. 4, we show in Fig. 12 the latitude-height contours of the

zonal means of $S$ derived under the assumption of $f/\widehat{\omega}_j = 0$. The stds and the top 10% largest shears are not shown here since they have similar patterns as the zonal means of $S$ but have smaller maxima than those shown in Fig. 4. From Fig. 12 we can see that the latitude-height patterns of the $S$ are the same as those shown in Fig. 4. However, the maxima of the zonal means of $S$ decrease from 12-17 ms$^{-1}$km$^{-1}$ for $f/\widehat{\omega}_j \neq 0$ to 11-15 ms$^{-1}$km$^{-1}$ for $f/\widehat{\omega}_j \approx 0$. The maxima of the zonal means of $S$ at latitudes higher than 50°N are at a higher height for $f/\widehat{\omega}_j \approx 0$ than that for $f/\widehat{\omega}_j \neq 0$. In a same manner as Fig. 5, we show in Figure 13 the profiles of $S$ as well as the top 10% and 1% largest $S$ derived under the assumption of $f/\widehat{\omega}_j = 0$ during January and July at around 40°N. From Fig. 13 we can see that the height variations of these profiles are similar to those shown in Fig. 5. However, the magnitudes of these profiles are slightly smaller than those shown in Fig. 5.

In a same manner as Fig. (6-7), we show in Fig. (14-15) the zonal means of $S$ during the four composite seasons and six yaw cycles, respectively, derived under the medium frequency assumption of $f/\widehat{\omega}_j = 0$. From Fig. (14-15) we can see these results have similar patterns as those for $f/\widehat{\omega}_j \neq 0$ but have slightly smaller magnitudes. Moreover, the maxima of these results at high latitudes of summer hemispheres are at a higher height for $f/\widehat{\omega}_j \approx 0$ than those for $f/\widehat{\omega}_j \neq 0$. The peaks circled by blue rectangles are very weak and almost disappear in Fig. (14-15) as compared to those shown in Fig. (6-7). The stds and the top 10% largest $S$ are not shown here since they have similar patterns as the zonal means of $S$ but have smaller maxima than those shown in Fig. (6-7).

In a same manner as Fig. (8-9), we show in Fig. (16-17) the latitude-height contours and time-height contours of the zonal means of $S$, respectively, derived under the medium frequency assumption of $f/\widehat{\omega}_j = 0$. Comparing between Fig. (8-9) and Fig. (16-17), we can see that the zonal means of $S$ derived under the medium frequency assumption of $f/\widehat{\omega}_j = 0$ have similar patterns as those for $f/\widehat{\omega}_j \neq 0$ but have slightly smaller magnitudes.

In summary, the GW-perturbed shears derived under the assumptions of $f/\widehat{\omega}_j \neq 0$ and $f/\widehat{\omega}_j = 0$ have similar patterns on the aspects of latitude-height, time-height contours. The magnitudes of the GW-perturbed shears derived under the assumption of $f/\widehat{\omega}_j = 0$ are slightly smaller than those under the assumption of $f/\widehat{\omega}_j \neq 0$.

**6 Summary**

Due to the important role the large vertical wind shears play in in the dynamics and electrodynamics of the MLT, there is a need for global observation. In response to this need, a method of deriving GW-perturbed shears is proposed in the work. The theorical basis of the method is the dispersion and polarization relations of linear GWs. Data employed are SABER temperature profiles measured over the past 18-year. Based on the method and the data, the global GW-perturbed shears are studied over a time span of 18-year.

The GW-perturbed shears derived here agree with previous lidar and sounding rocket observations in the aspects of height structures and magnitudes on the climatology sense. Moreover, the GW-perturbed shears derived here agree with the high-resolution model simulation results in the aspects of latitude-height patterns but have smaller magnitudes in the zonal mean sense. The GW-perturbed shears reach their maxima around the mesopause region and increase with the increasing

latitudes. At most latitudes and during all seasons, the maxima of GW-perturbed shears are at $\sim z$=90-100 km. At high latitudes of the summer hemisphere, the maxima of GW-perturbed shears at a lower height ($\sim z$=80-90 km). This latitude-height pattern of GW-perturbed shears is independent on years. The magnitudes of the GW-perturbed shears exhibit year-to-year variations.

The GW-perturbed shears exhibit more prominent AO and SAO at high latitudes than those at lower latitudes. The height variations of the amplitudes AO and SAO are hemispheric asymmetric. The strong AO occurs at around 80 km in the NH and above 92 km in the SH. At middle to high latitudes, the phases of AO shift from winter to summer and then to winter again with the increasing height. The amplitudes of SAO decrease with the decreasing latitudes. The phases of SAO are in May and June when the SAO reach its peak at middle to high latitudes.

The main limitation of the method is the overestimation of the GW-perturbed shears due to the unresolved GW propagation direction by the method. The other limitations, such as the observational filter, long sampling distance and cutoff criterion of the vertical wavelength, will underestimate the GW-perturbed shears. To overcome these limitations, it is necessary to develop new techniques for remote sensing temperatures or winds from space, such as limb imaging techniques allowing to infer temperatures in three dimensions along the orbital track at high horizontal and vertical resolution.

*Data availability.* The SABER data are downloaded from ftp://saber.gats-inc.com/Version2_0/Level2A/

*Author contribution.* XL and JX designed the study and wrote the manuscript. JY and HL contributed to the discussion of the results and the preparation of the manuscript. All authors discussed the results and commented on the manuscript at all
stages.

*Competing interests.* The authors declare that they have no conflict of interest.

*Acknowledgments.* The SABER data were obtained from ftp://saber.gats-inc.com/Version2_0/Level2A/. This work was
supported by the National Natural Science Foundation of China (41831073, 41874182). This work was also supported in part by the Specialized Research Fund and the Open Research Program of the State Key Laboratory of Space Weather. National Center for Atmospheric Research is a major facility sponsored by the National Science Foundation under Cooperative Agreement No. 1852977.

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

**Figures and Captions**

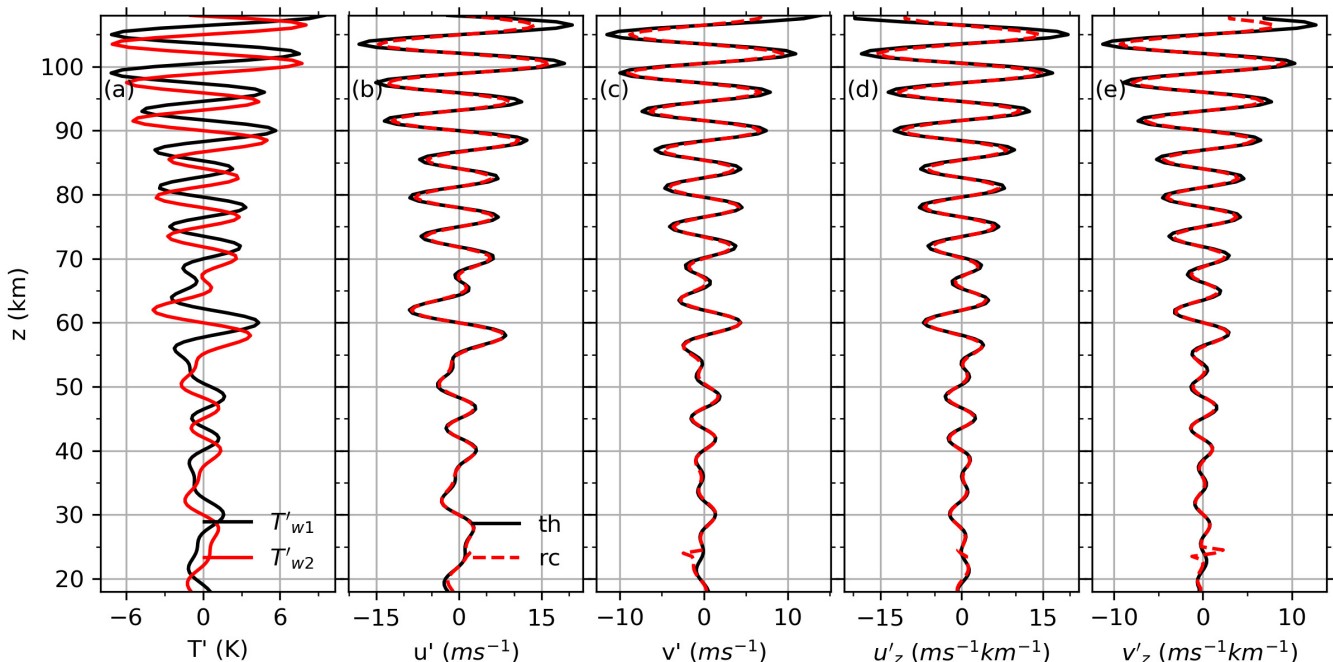

**Figure 1:** Synthetic temperature perturbation profiles (a, $T'_{w1}(z)$ and $T'_{w2}(z)$ are represented by black and red lines, respectively) and the corresponding winds (b, zonal; c, meridional) and shears (d, zonal; e, meridional). The winds and shears are, respectively, calculated from the prescribed amplitudes and wavenumbers (black line, labelled as "th") and reconstructed by spectral decomposition method (red dashed line, labelled as "rc"). All panels have the same y-axis scale.

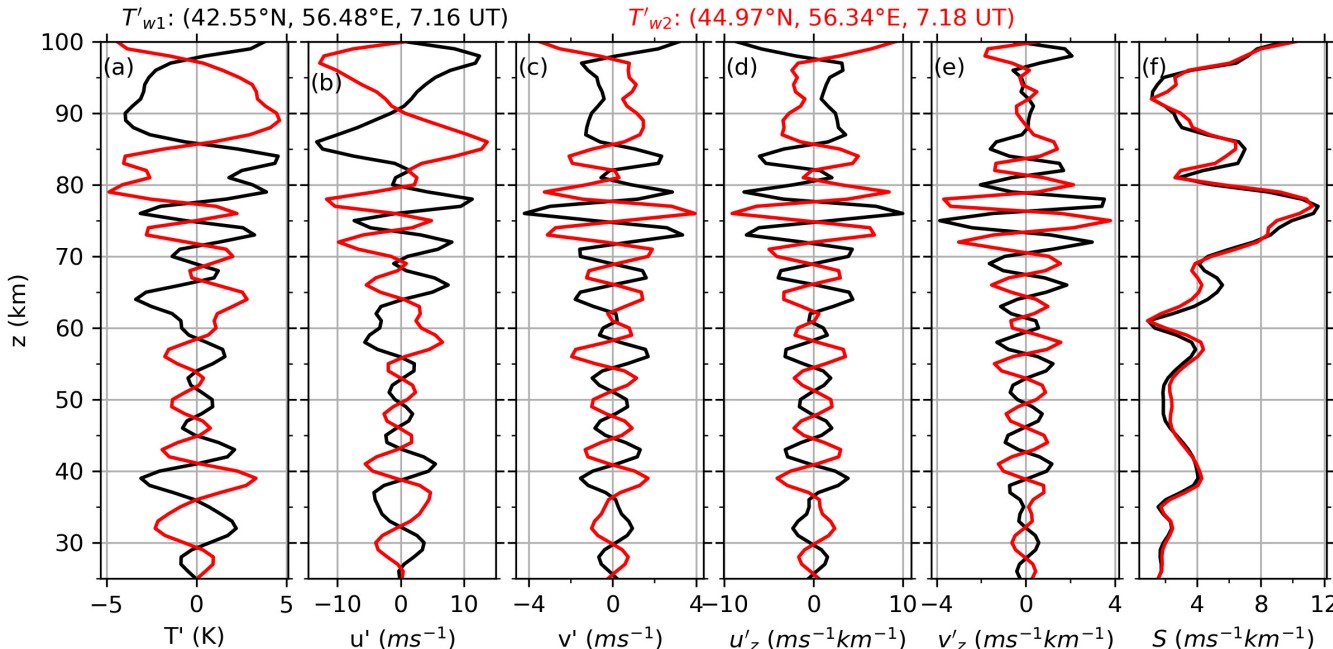

**Figure 2:** Procedure of derivation winds and shears induced by GW profiles pair (red and black lines indicate the results from $T'_{w1}$ and $T'_{w2}$, respectively). (a): $T'_{w1}$ and $T'_{w2}$ were measured at (42.55ºN, 56.48ºE, 7.16 UT) and (44.97ºN, 56.34ºE, 7.18 UT), respectively, on 1st January 2018. The distance between the two adjacent profiles is 269.8 km. (b) and (c): the GW induced winds in the along and cross track directions, respectively; (d) and (e): the GW induced shears in the along and cross track directions, respectively; (f): the GW-perturbed shear ($S$). All panels have the same y-axis scale.

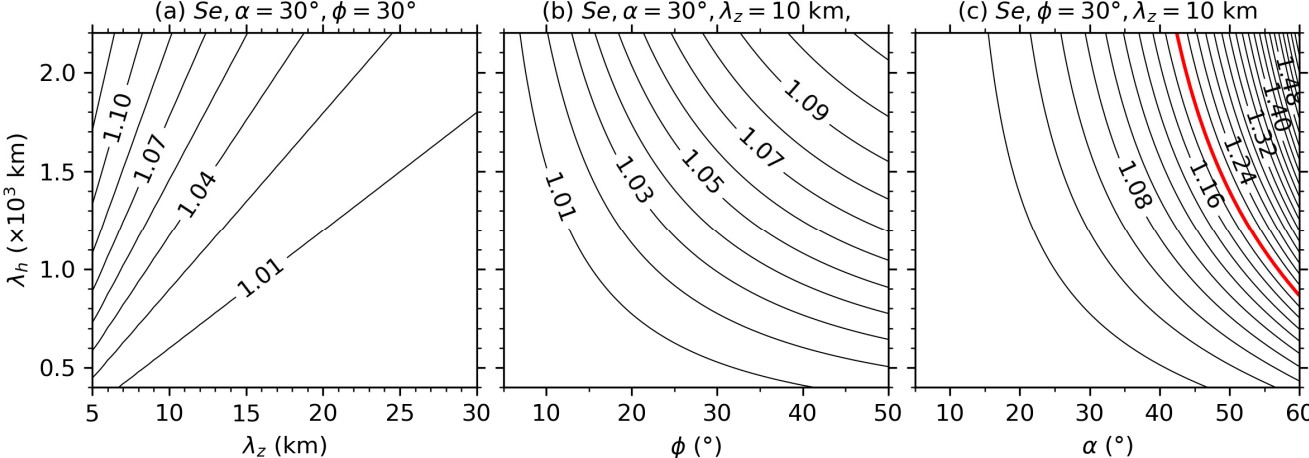

**Figure 3:** Dependencies of $S_e$ on the horizontal and vertical wavelengths for $\alpha = 30°$ and $\phi = 30°$ (a), on the latitude $\phi$ for $\alpha = 30°$ and $\lambda_z = 10$ km (b) and on the angle $\alpha$ for $\phi = 30°$ (c). The red contour line in (c) indicates $S_e = 1.2$. All panels have the same y-axis scale.

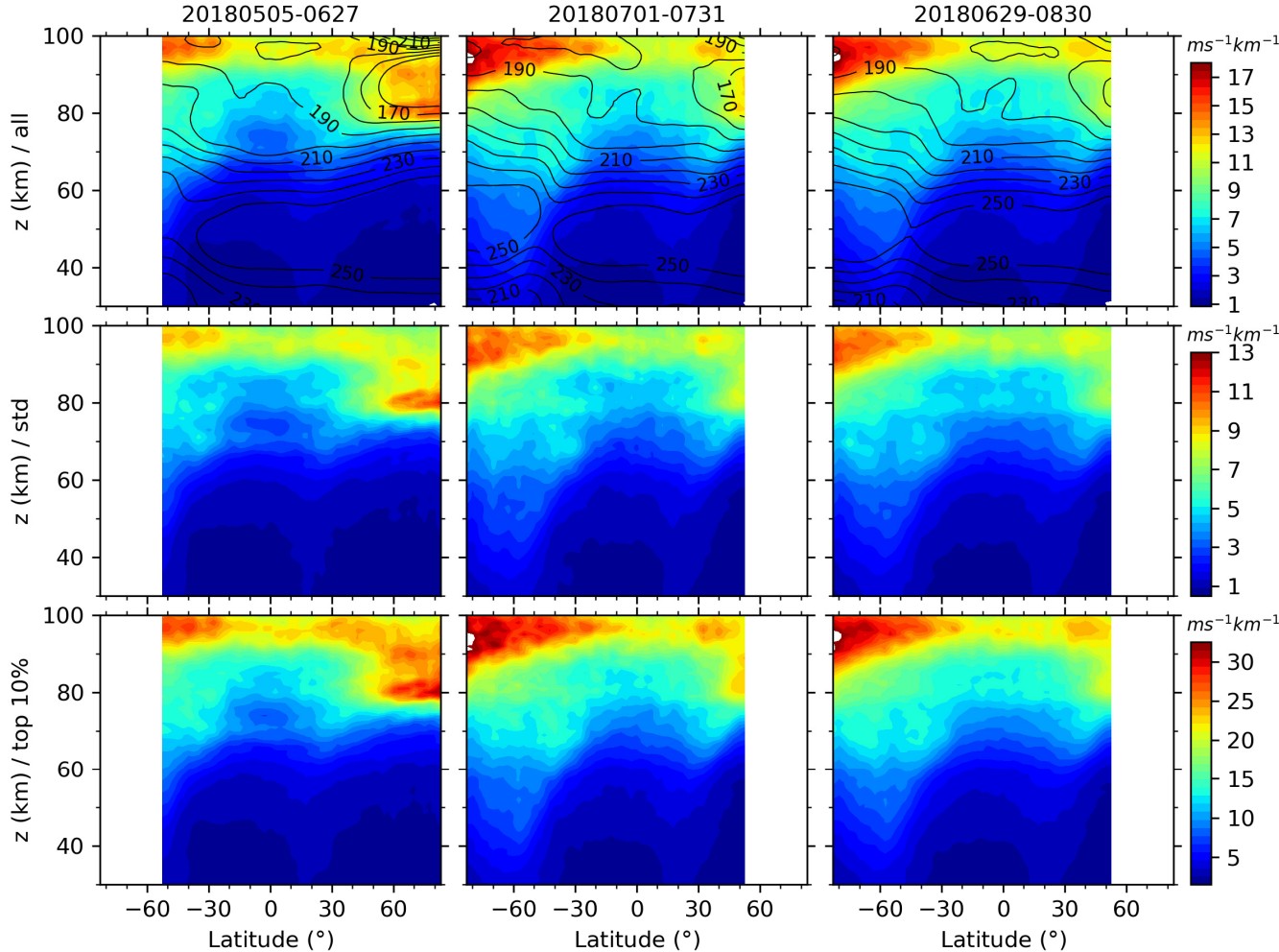

**Figure 4:** Latitude-height contours of the zonal means (upper row) and standard deviations (std, middle row) of GW-perturbed shears and the top 10% largest shears (bottom row) during three periods (left column: 0506-0627; middle column: 0701-0731; right column: 0629-0830, the four numbers mean mmdd). The contour lines shown in the upper row are the corresponding zonal mean temperature. Same color scale is used in each row with unit of ms⁻¹km⁻¹. All panels have the same scales in both x-axis and y-axis.

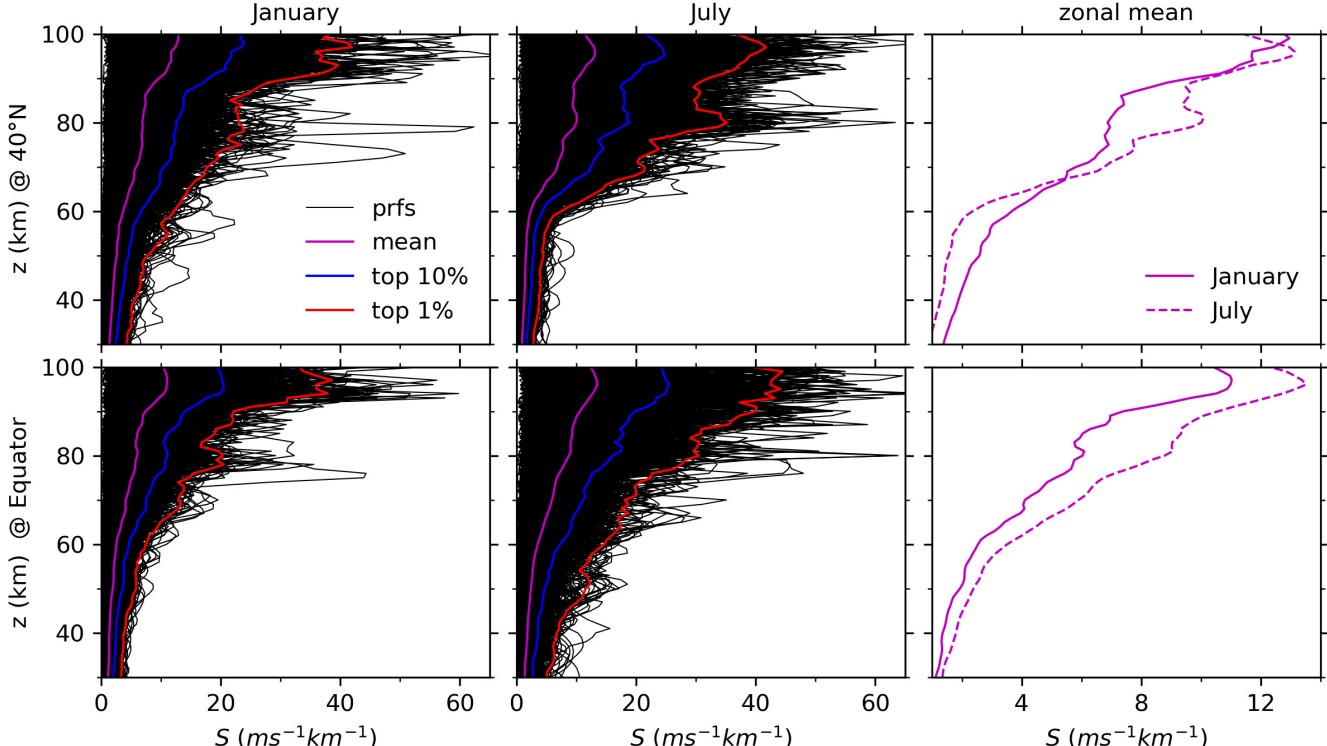

**Figure 5:** Profiles of S (black) and their mean (magenta) as well as the top 10% (blue) and 1% (red) largest S during January (left column) and July (middle) at around 40°N (upper) and Equator (below). The zonal means of $S$ during January (solid) and July (dashed line) at the two latitudes are also shown in the right column for comparison. Same x-axis (y-axis) scale is used in each row (column).

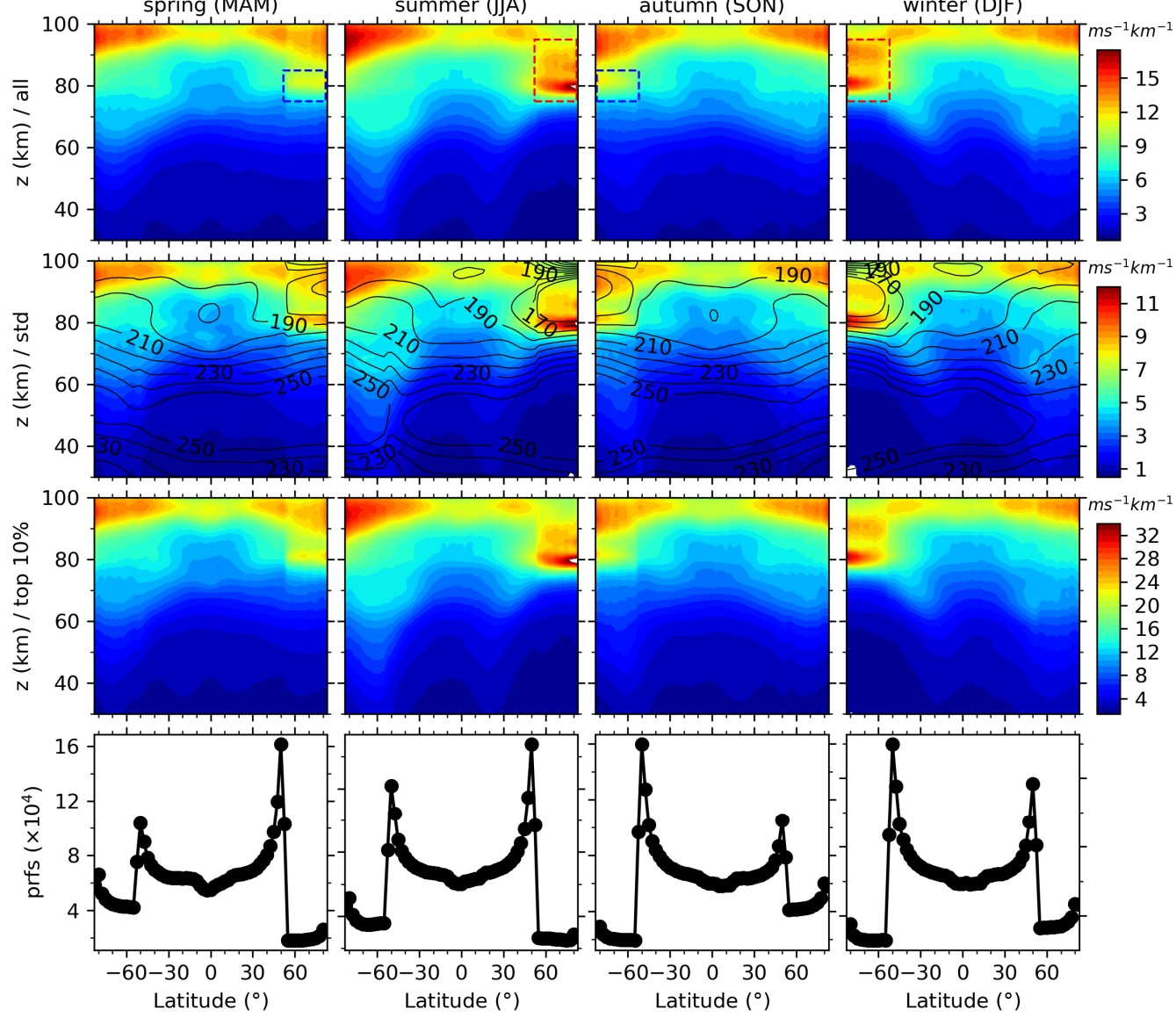

**Figure 6:** Latitude-height contours of the zonal means (the first row) and standard deviations (std, the second row) of GW-perturbed wind shears and the top 10% largest shears (the third row) during four composite seasons (noted on the top of each column). The composite season is the superposition of the corresponding season from 2002 to 2019. For more readable, the zonal mean temperatures are shown as contour lines only in the second row. The rectangles are to highlight the peak at a lower height. The numbers of profiles used to derive GW-perturbed wind shears in each season are shown in the fourth row. Same color scale is used in each row with unit of ms$^{-1}$km$^{-1}$. Each row has the same y-axis scale. All panels have the same x-axis scale.

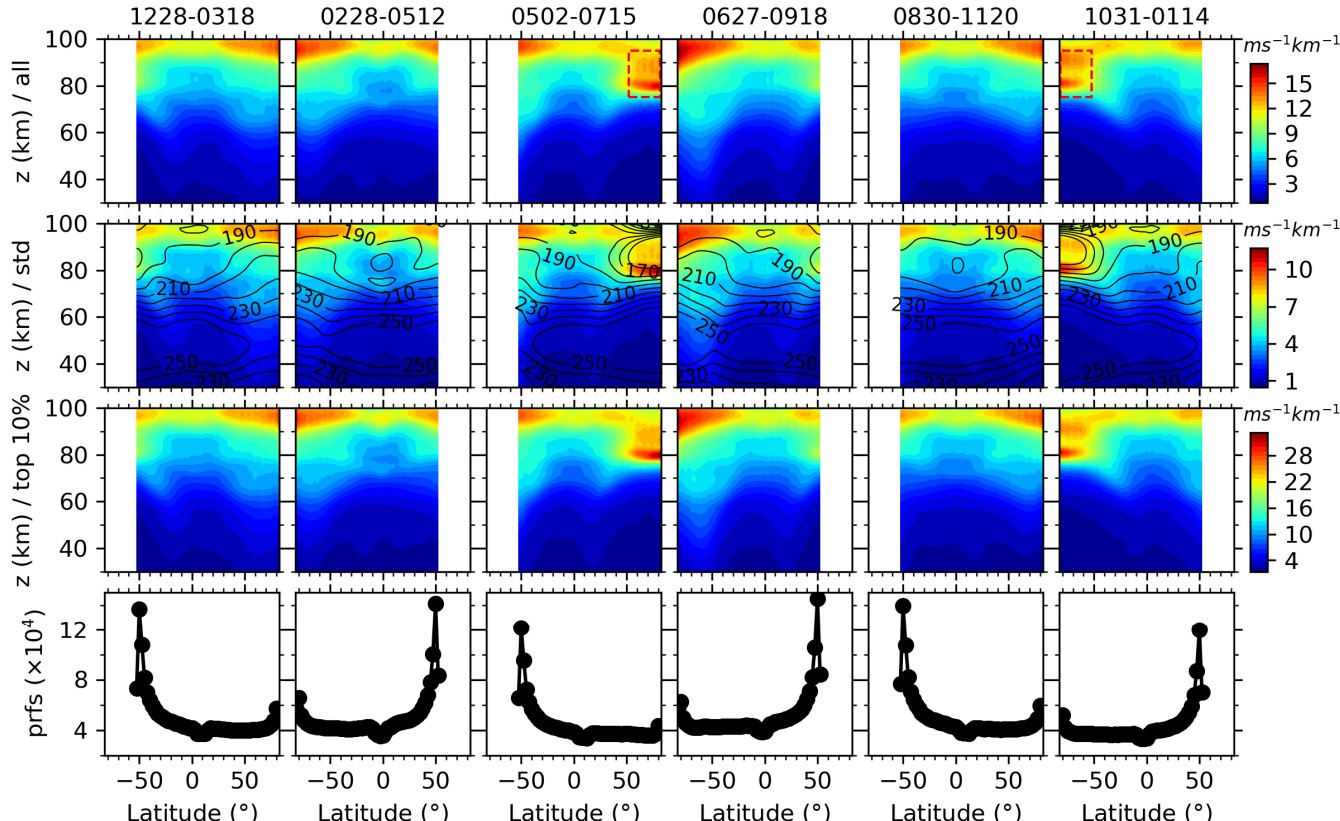

**Figure 7:** Same caption as Fig. 6 but during six composite yaw cycles from 2002 to 2019. The composite yaw cycle is the superposition of all the yaw cycles, which have nearly identical date coverage relative to the beginning of each calendar year, from 2002 to 2019. The date coverage of each yaw cycle is labelled on the top of each column (see text for detail). For more readable, the zonal mean temperatures are shown as contour lines only in the second row. The rectangles are to highlight the peak at a lower height.

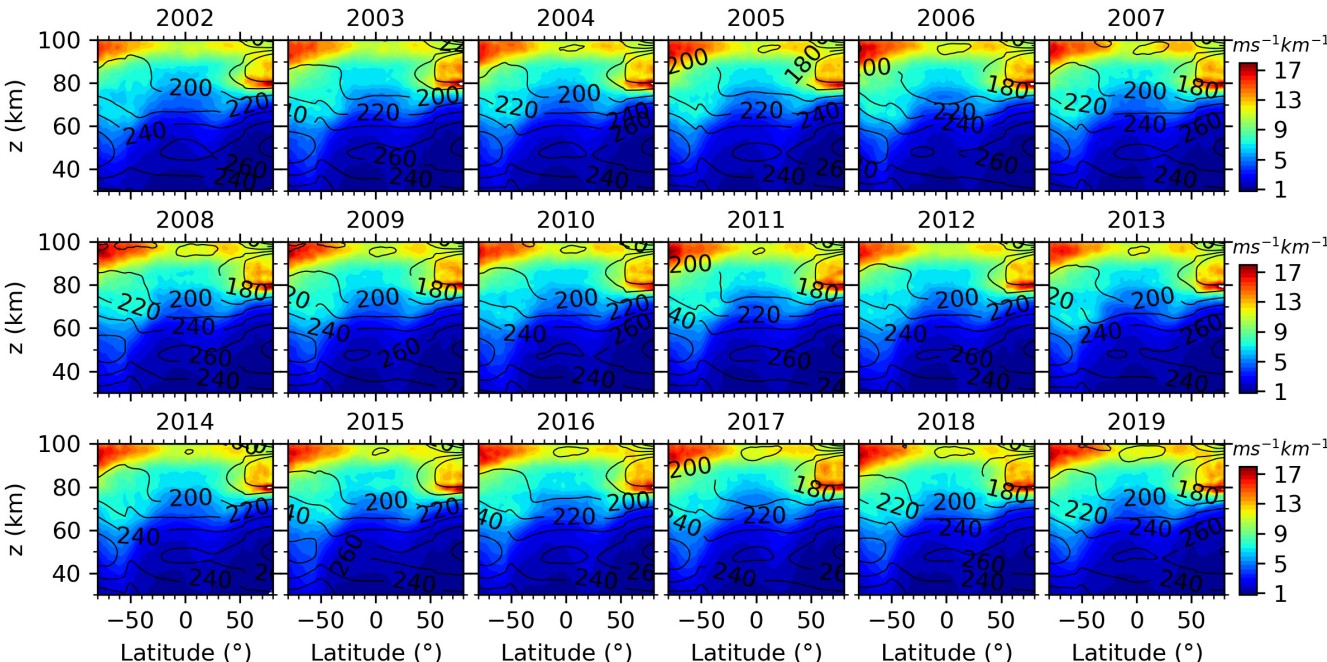

**Figure 8:** Latitude-height contours of the zonal mean GW-perturbed wind shears (color filled contour) and temperature (contour lines) during each summer from 2002 to 2019. All panels have the same color scales with unit of ms⁻¹km⁻¹. All panels have the same x-axis and y-axis scales.

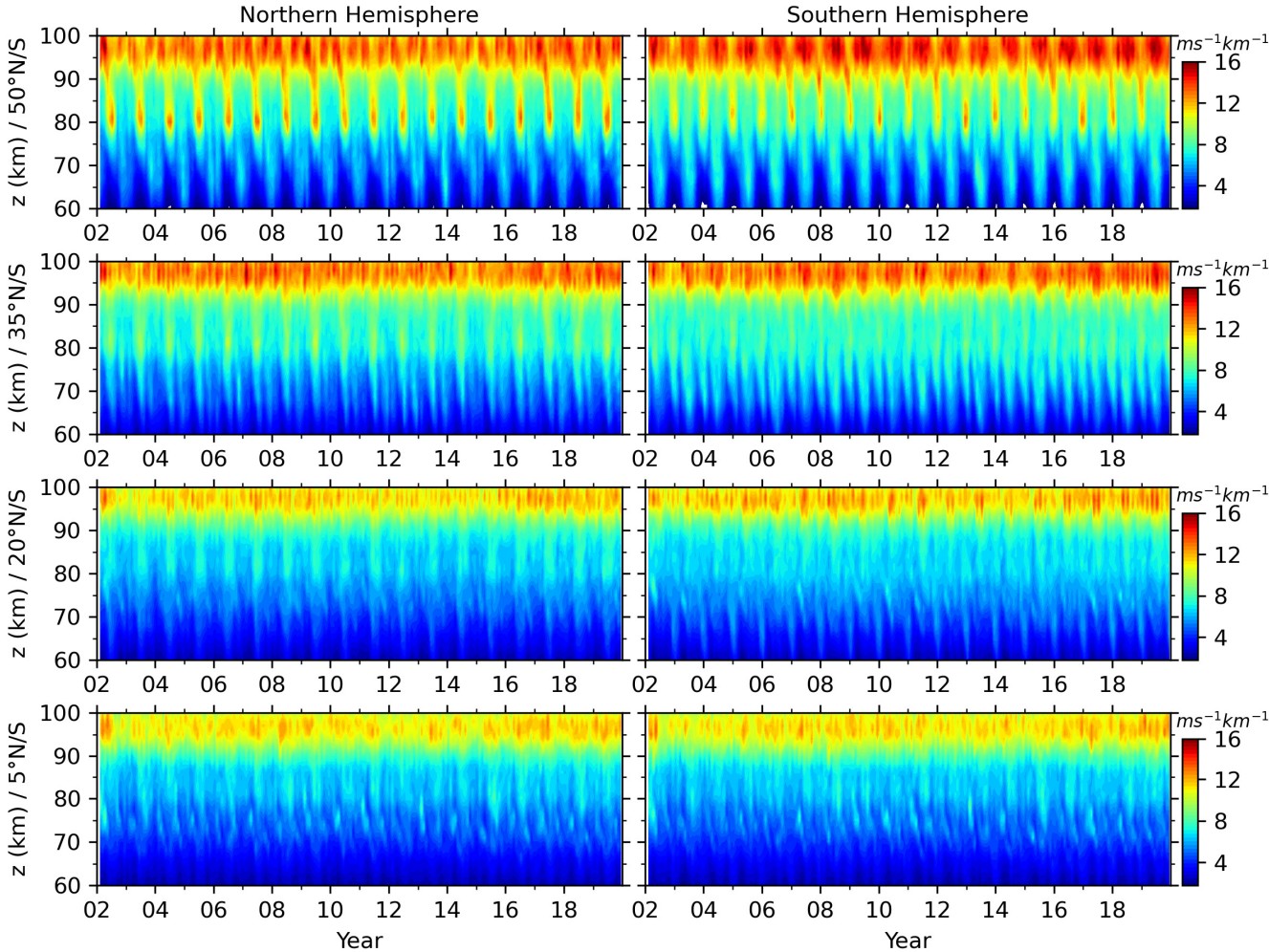

**Figure 9:** Time-height contours of the monthly zonal mean GW-perturbed shears ($S$) at four latitudes (50°N/S, 35°N/S, 20°N/S, 5°N/S, see the y-label of each row) of the NH (left column) and SH (right column) from 2002 to 2019. All panels have the same color scales with unit of $ms^{-1}km^{-1}$. All panels have the same x-axis and y-axis scales.

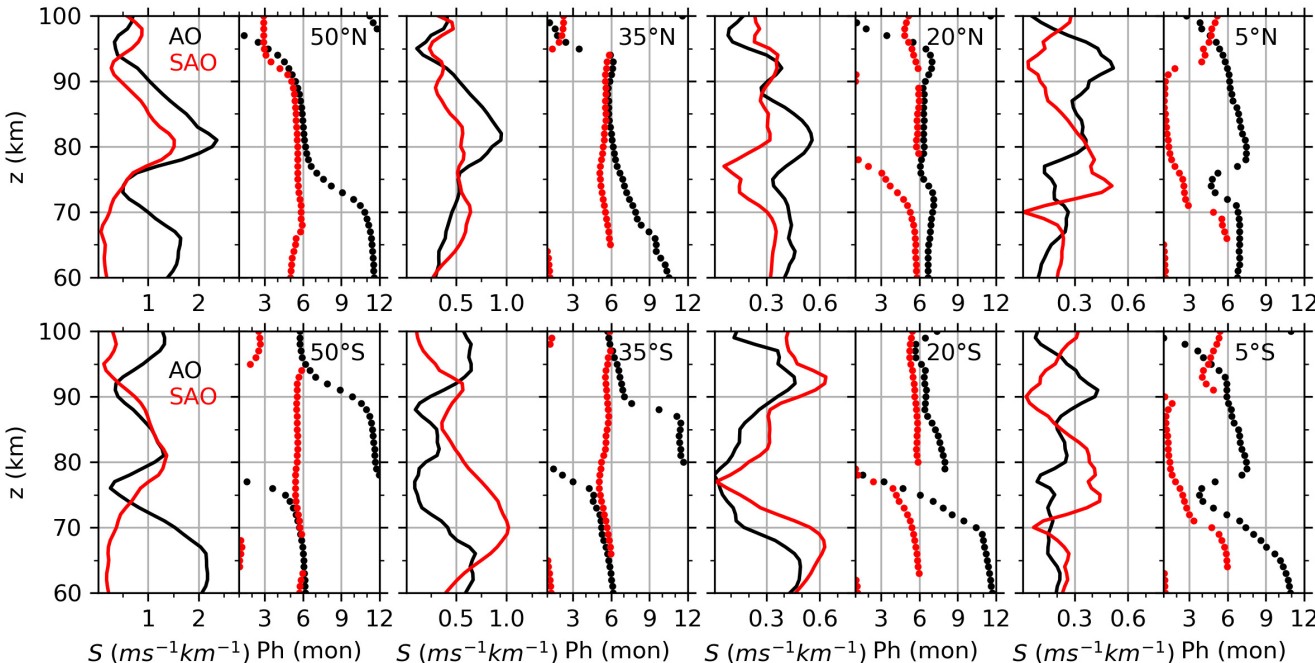

**Figure 10:** Amplitudes (left column of each panel) and phases (right column of each panel) of AO (black) and SAO (red) of the GW-perturbed wind shears at four latitude bands of NH (upper row) and SH (lower row). The phase is defined as the month when the oscillation reaches its peak. Same x-axis is used at each latitude band. Same y-axis scale is used for all panels.

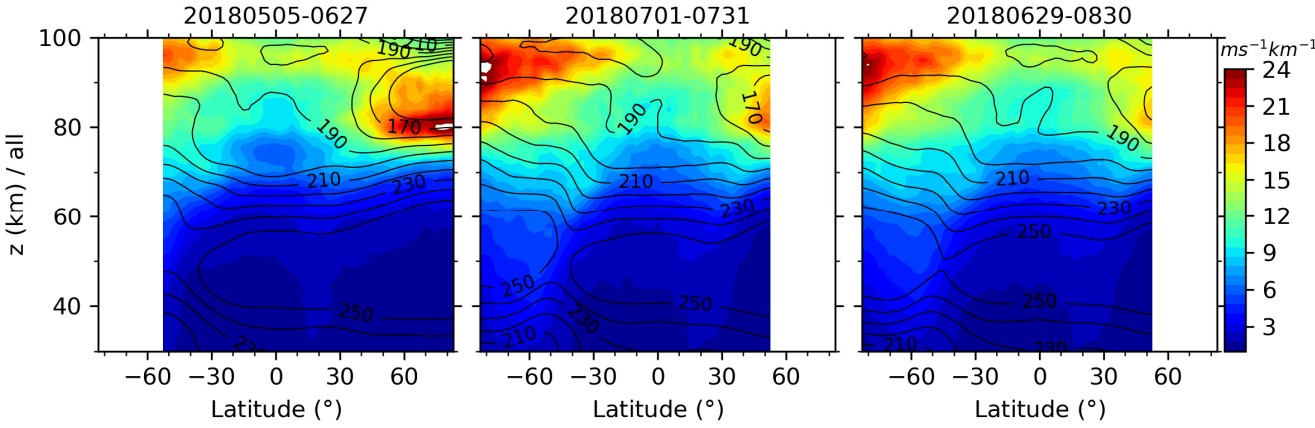

**Figure 11.** Same caption as Fig. 4 but for the zonal mean GW-perturbed shears with cutoff criterion of 3 km.

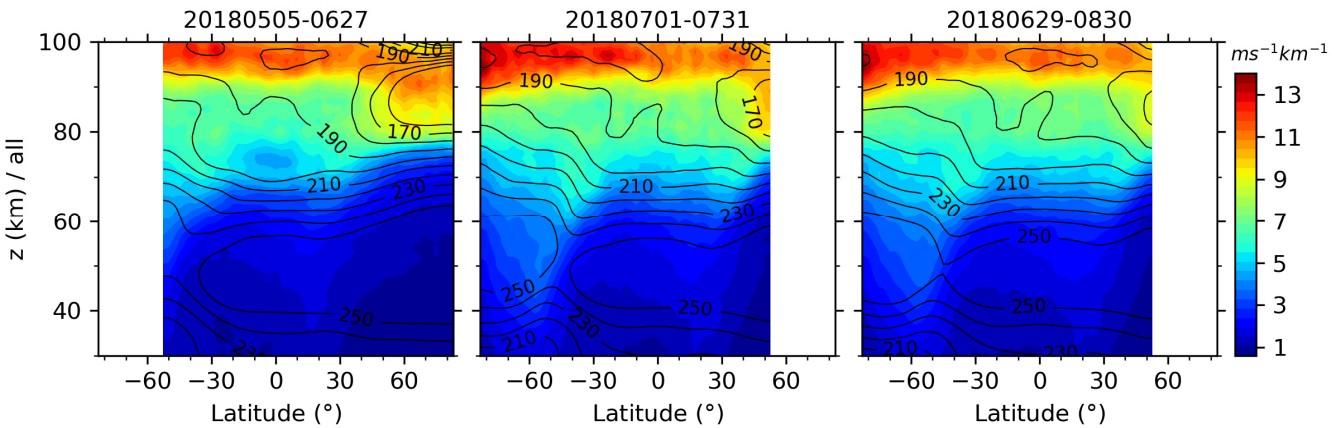

**Figure 12.** Same caption as Fig. 4 but for the zonal mean of GW-perturbed shears derived under the medium frequency assumption $f/\widehat{\omega}_j = 0$.

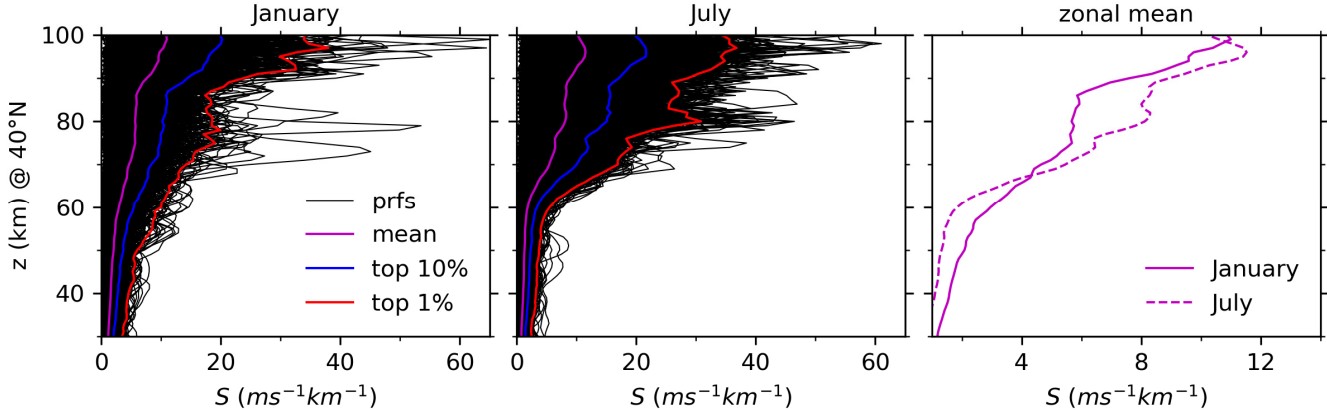

**Figure 13.** Same caption as Fig. 5 but for the GW-perturbed profiles derived under the medium frequency assumption $f/\hat{\omega}_j = 0$ and at 40°N

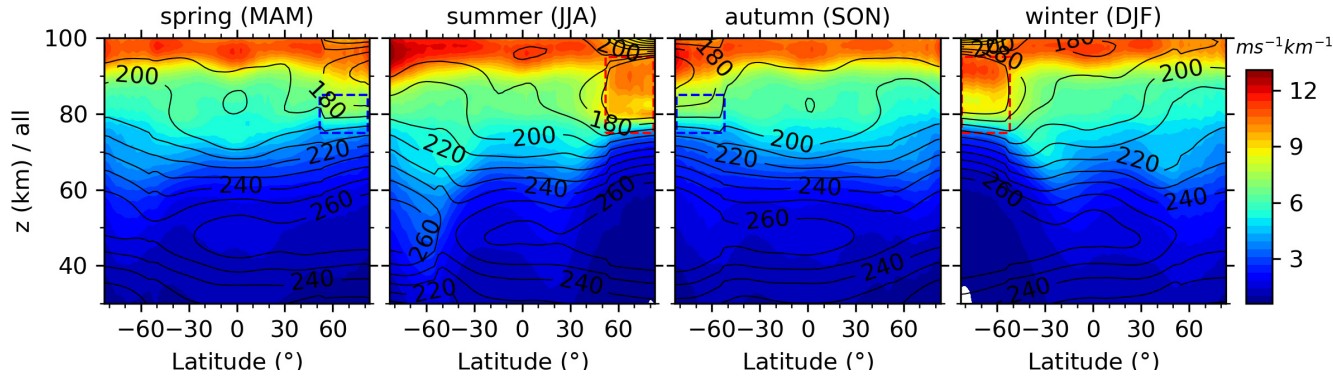

**Figure 14.** Same caption as Fig. 6 but for the zonal means of GW-perturbed shears derived under the medium frequency assumption $f/\widehat{\omega}_j = 0$.

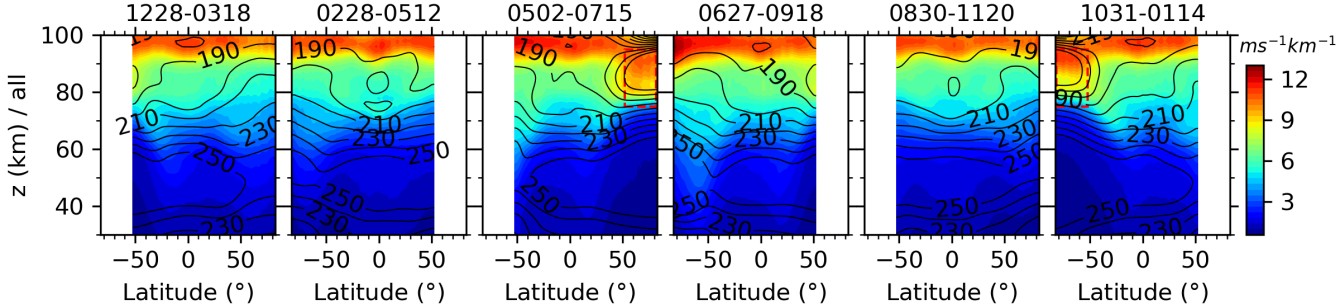

**Figure 15.** Same caption as Fig. 7 but for the zonal means of GW-perturbed shears derived under the medium frequency assumption $f/\hat{\omega}_j = 0$.

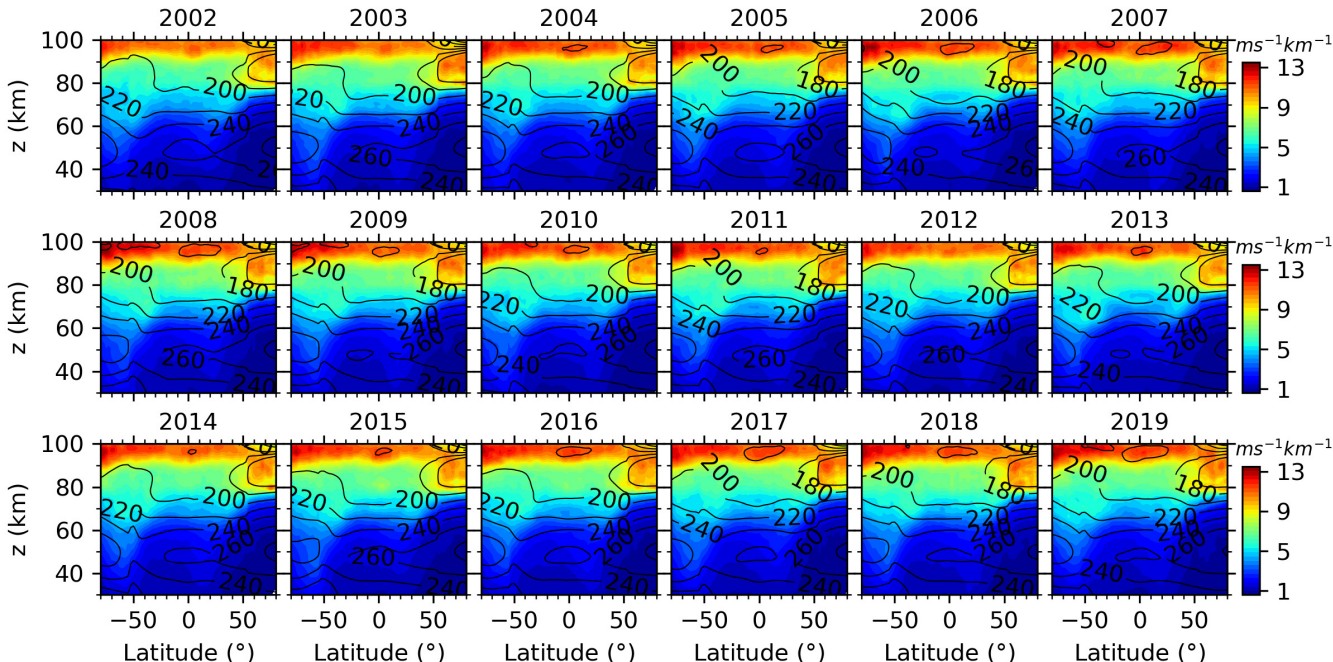

**Figure 16.** Same caption as Fig. 8 but for the GW-perturbed shears derived under the medium frequency assumption $f/\widehat{\omega}_j = 0$.

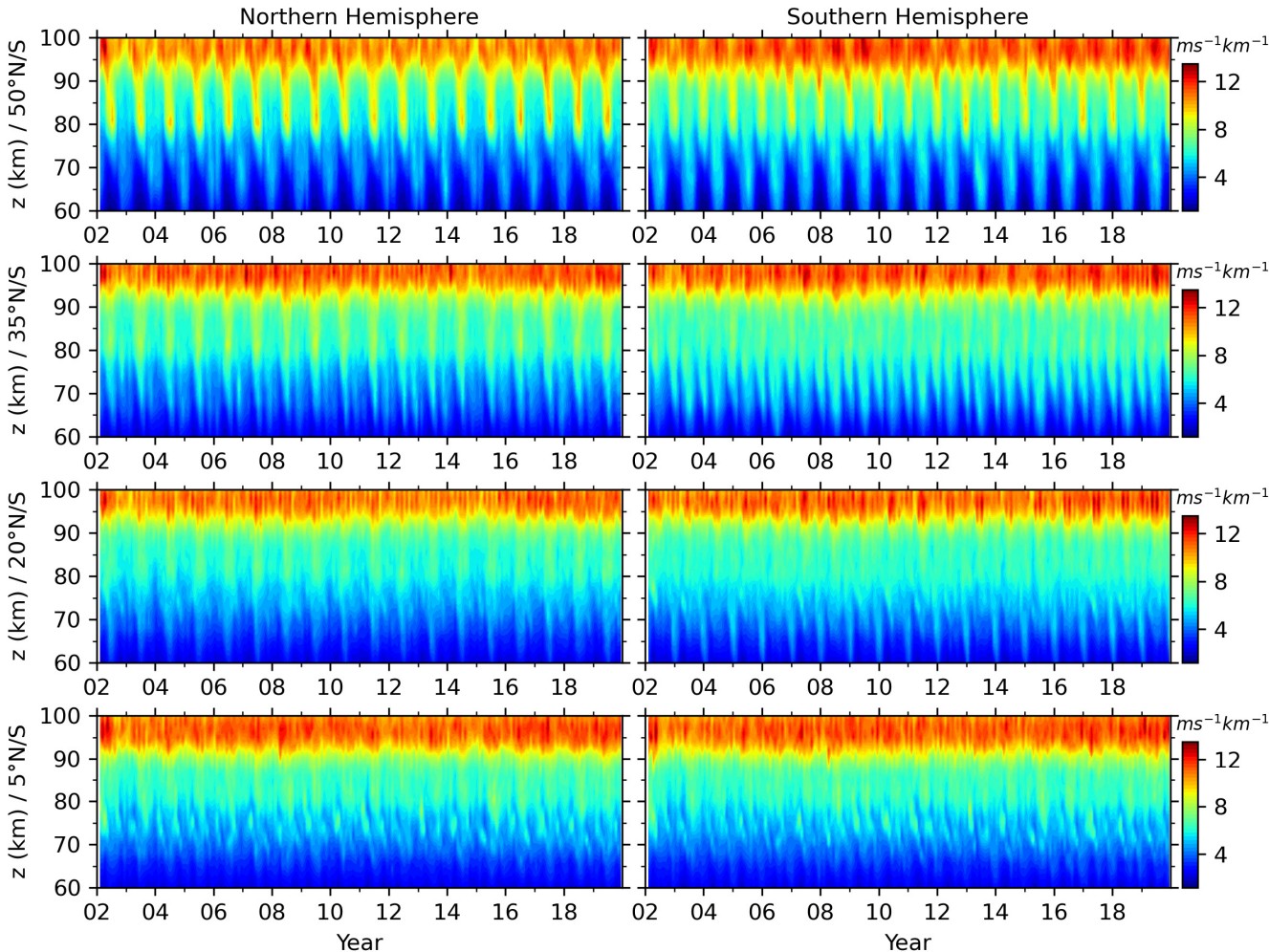

**Figure 17.** Same caption as Fig. 9 but the GW-perturbed shears derived under the medium frequency assumption $f/\widehat{\omega}_j = 0$.