# Peer review of "Gravity Waves Perturbed Wind Shears Derived from SABER Temperature Observations"

_Atmospheric Chemistry and Physics, 2020_

## Referee Comment (RC1) · Anonymous Referee #1 · 3 Aug 2020

The paper by Liu et al. provides climatologies of vertical wind shear. This can be obtained essentially from temperature amplitude multiplied by vertical wave number and is thus different from climatologies of GW amplitudes (without the m) and GW momentum flux which scales m*Tˆ2. Both m and T-amplitude can, in principle gained from vertical profiles obtained from satellites at good accuracy. Corrections are applied for the ratio of intrinsic frequency to Coriolis parameter. For these, however, quite a number of assumptions have to be made. These are either not explained at all or not very well in the paper. A further major point is that the scaling with m emphasizes the noise of the observations. That fact needs to be discussed. My recommendation is therefore to return to the authors for major revisions.

Major comments:

[Figure]

1. All the problems in the theory part arise out of the fact that you apply the omega/f correction. My recommendation therefore is to present also the results for mid-frequency assumption (omega » f), for figure 4 in the main text, for figures 5-9 in the appendix. That would allow the reader to estimate for herself/himself whether the assumptions which need to be made (at least for current satellites) are important to the findings. Are they just amplifying the shears in an about uniform way or are they contributing to the shape of the distributions.

2. Rewrite the theory part and properly name the assumptions (see also detailed comments below). How does your picture change, if you assume that the different waves you superpose have different propagation directions? The following more in the line of suggestions, which may be beyond the scope of the paper: It would be nice to have a Monte Carlo simulation showing the effect of superposing waves of different omega/f and different propagation direction, but I admit that the result would again depend on some assumptions about the real distributions. You could also try to use simulated observations through the model (just sampling, no radiative transfer) to get some idea on the size of effects. Maybe it also would be worthwhile to assume first that one could infer propagation direction as well, as in principle such instruments could be build.

3. Discuss the influence of noise. In particular it is known that SABER temperature retrievals have a noise problem at the summer mesopause. That is stated on the SABER web-page and also discussed for GW analyses by Ern et al. 2018. The results for this particular region have to be treated with care. At least some discussion must be included in the paper.

4. The paper must be self explaining. Explain how you arrive at your vertical wavelengths and amplitudes. Anyway: You should not use FFT over the whole altitude regime. There is so much Doppler shift, critical level filtering and oblique propagation going on that the case where you observe the same gravity wave at all altitudes and furthermore with similar vertical wavelength is the absolute exception.

5. In particular: how do you deal with tides? All tides must be removed before the analysis as they are global wave modes with wind amplitudes maximizing at a different latitude than temperature amplitudes.

6. Discussions: for GWMF I think it is a valid assumption that wind modulation causes most of the relevant waves to propagate opposite to the winds. You can find some of this expressed in AIRS results and also GW reolving global models. However, for shear I don't think that is further valid. Therefore I would include some of the discussion (very much shortened) in sec. 2.

Specific comments:

P2L4 It is now accepted ... A larger part of the shear and the winds is associated with tides. You need to mention them here. In particular sporadic E is primarily associated with tides (e.g. Arras et al., 2009, Arras et al. 2018).

P2L9 Please explain this equation: what is the motivation? Static stability of saturated mid frequency GW?

P2L34 "poorly known" Motivate. What are the limitations of the wind observations? Noise? Altitude resolution? Combination to vector winds?

P3 Equ 1: Under the assumption of conservative propoagation and no refraction. Reasonable for some waves in a limited altitude range, but must not be taken over the whole MA as is done below.

P3L20/L25 These polarization relations are for winds u parallel and v perpendicular to the wavevector of the GW. You can do the assumption, but must say so in the text.

P4 Equ. 6 and Equ. 7 These equations make the much stronger assumption that the wave vectors are all pointing in the same direction. While that seems a plausible assumption, if you look for the largest amplitudes (preferrential orientation against the mean wind), I don't see why this should be the case for wind shear. You shown in Equs 4 and 5 that shear is proportional to m as well as to the amplitude. Handwaving, these

two effects should largely cancel with regards to phase speed. Thus, you would not expect a direction preference.

P4 Equ. 8 and 9 then are not valid

P5 Equ 12 is fine, but is introduced in a way that makes belief that you could get the horizontal wavenumber from the satellite. Instead it is the one projected to the track. This will introduce overestimate of lambda_h, underestimate of k_h and thus underestimate of $\hat{\omega}$. You introduce some of these arguments below, but this must be more clearly ordered.

P5L8 No you can not! This is not the zonal and meridional component!

P5L15 And why should the wavevector point along the orbital track? That is as unlikely.

P5L25 There is no reason to believe that the waves at 20km have anything to do with the waves at 100km. Please explain your method. Phase differences should be gained from "local" phases determined for a limited altitude range

P8L6 can you please make these number codes plain text 01 July - 31 July which Year?

P8L15 This is a region problematic for retrievals due to very cold temperatures and low IR signals (cf. Remsberg, Ern). Some of what you see there may be real, but some is very likely due to increased noise. A discussion is needed.

P9L16 You mean the zonal mean of the individual shear profiles? Please be precise, as the zonal mean shear due to GWs should be zero. But then: what does the lidar provide? Average of abs.(shears) or shear of average?

P9L25 You do not discuss here the visibility filter, which also should reduce amplitudes

P12L17 And also on what you emphasize due to the scaling with m: shear (short vertical wavelength), amplitudes (mid lz) or GWMF (long lz).

P12L21 also Schmidt et al.

P13L11 SABER "sees" GWs with horizontal wavelengths > 200km, much the same as WACCM, though with somewhat reduced amplitude. I would check by mid-frequency approach, whether the scaling and profile distance is really of importance. In addition, as you said above, if there is an influence, it would be an overestimation.

Technical corrections:

P1L28 ... (or more precisely vertical wind shears) ...

P1L30 ... and from ground-based ...

P2L2 ... strong gravity wave (GW) activity.

P2L2 You probably want to refer to a generally knowledge on GWs rather than on evidence from the aforementioned papers for the sources: Most prominent sources of GWs are convection, orography and jets and fronts in the troposphere as well as spontaneous adjustment and secondary wave generation in the stratosphere. Amplitudes and shears increase when the GWs propagate ...

P2L13 driving -> are the cause of

P2L17 reproduced the large wind shears seen in the observations.

P2L22 Observations ... were made by sounding ... for a very limited number of locations only and hence cannot ...

P2L24 still challenging

P3L4 of deriving shears from GW anylses of temperature observations.

P3L6 SABER luckily still measures: Temperature profiles measured by .... from 2002 to 2019 are used for this study providing an 18 year period.

P3L8 ... website based on Remsberg et al. (2008).

P3L10 omit () at this point

P4L3 In reality, usually spectrum of GWs is observed formed by superposition of several monochromatic GWs.

P5L7 and the amplitude and vertical wavenumber determined from the satellite observations for the individual monoc. GWs, we ...

P6L26 https://en.wikipedia.org/wiki/Flowchart Please use different wording here

P7L20 a factor 0.2 would be underestimation overestimated by 20% ?

P8L9 comment -> common

P9L15 slightly larger

---

## Referee Comment (RC2) · Anonymous Referee #2 · 9 Aug 2020

SABER temperature profiles are used as input to a calculation of gravity wave parameters, including the vertical and horizontal wavenumbers, the intrinsic frequency, and the horizontal velocity perturbations associated with each wave component in the decomposed wave spectrum. The goal is to estimate the vertical shears associated with the wave spectrum over the extended time periods and greater geographical extent provided by the satellite data.

The presentation of the material and methodology is clear, the discussion of past work is useful and very complete, and the figures represent the results well. Nonetheless, there are several major questions related to the validity of the analysis.

The shears obtained from the SABER data are calculated and are not directly measured. A broader global climatology, including the seasonal variations, is valuable

to the community and represents a worthwhile goal. There are a number of critical assumptions required, however, for the calculations to be valid and meaningful. Those assumptions are especially likely to break down in the altitude range around the mesopause, which is where the largest shears occur that are a focus of the analysis. The assumptions are not adequately addressed in the paper, and the effect of those assumptions are therefore also not addressed. The recommendation is to return the manuscript to the authors for revisions.

Specific comments are as follows:

Tides are in some sense global-scale gravity waves constrained by spherical boundary conditions, but they clearly do not conform to the simple linearized polarization relations given in equations (2) and (3). The authors introduce some filtering to reduce or eliminate larger-scale modes, but it is not clear that the tidal modes, especially the higher order modes, are eliminated prior to extracting the gravity wave spectral components. This is especially important since the altitude range where the large shears occur near the mesopause is also an altitude range where a number of tidal modes have large amplitudes, perhaps not coincidentally.

A further concern is that the region of large shears near the mesopause is also a region of large winds that represents a height range with critical levels for a large fraction of the gravity wave spectrum, especially since the background winds rotate with height through that altitude range, so there are critical levels for a broad range of propagation directions. It appears questionable to apply the linearized polarization relations in regions with critical levels where the vertical wavelength is changing rapidly with height and the dynamics are almost certainly nonlinear. The validity of the analysis in those regions is not discussed.

The large shear region is also a region where many gravity waves break, producing turbulence and, in some cases, secondary waves, as discussed in at least one of the references cited in the manuscript. The critical levels mentioned in the previous comment can produce breaking, but breaking waves can also be a result of the amplitude growth with height, which by itself leads to amplitudes large enough to produce such effects at heights near the mesopause.

If we assume a localized source in the lower atmosphere that produces a spectrum of waves, then the lowest-frequency gravity waves propagate at a lower elevation angle while the higher-frequency waves propagate at angles closer to vertical. A vertical profile extending from the ground to the lower thermosphere is therefore unlikely to have the contents from a single set of monochromatic waves contributing at all heights. As an example, several studies have been published in the past with special cases in which waves have been traced from the troposphere to the thermosphere when there has been an especially strong source, such as a strong line of thunderstorms, but those waves become displaced horizontally as they propagate vertically so that a vertical sample would only intercept the wave packet in a small part of the complete altitude range. In the general case the altitude profile measured by the satellite will likely have contributions from an extended range of geographically-distributed sources and therefore also different phasing dominating the vertical wavelength contribution in each part of the altitude range. If so, it is not clear that an individual wavelength contribution extracted via a DFT represents the wave structure accurately and applying the polarization relations as if that particular component represents a single monochromatic wave seems questionable.

Finally, since only the along-track direction is sampled, the horizontal wavelength will be overestimated for any waves propagating in an off-track direction. The resulting underestimate of the horizontal wavenumber artificially increases the value of omega prime (equation 12) and leads to an overestimate of the vertical shear (equations 8 and 9). Given that, it seems possible that the apparent seasonal variations and latitudinal variations in the derived shears are a reflection of seasonal and latitudinal changes in preferred wave propagation directions rather than actual changes in the magnitudes of the shears.
* * *
Interactive
comment

A minor comment is that the title of the paper, which uses the terms "gravity wave induced shears", suggests an effect in which waves are accelerating the flow to produce the shears, i.e., inducing a shear by momentum deposition for example, but that is not what the manuscript describes since the analysis focuses only on the superposition of shears that are naturally a part of the wave spectrum without any mean-flow acceleration.

---

## Short Comment (SC1) · 21 Aug 2020

**Responses to Referee#1:**

The paper by Liu et al. provides climatologies of vertical wind shear. This can be obtained essentially from temperature amplitude multiplied by vertical wave number and is thus different from climatologies of GW amplitudes (without the m) and GW momentum flux which scales m*T^2. Both m and T-amplitude can, in principle gained from vertical profiles obtained from satellites at good accuracy. Corrections are applied for the ratio of intrinsic frequency to Coriolis parameter. For these, however, quite a number of assumptions have to be made. These are either not explained at all or not very well in the paper. A further major point is that the scaling with m emphasizes the noise of the observations. That fact needs to be discussed. My recommendation is therefore to return to the authors for major revisions.

**Response:** Thanks for your valuable comments on our manuscript. According to your comments, the main revisions are: (1) wavelet transformation method is used to decompose GWs and reconstruct shears; (2) the results under the medium frequency assumption are provided. These revisions do not change our conclusion. Please see below for the point-to-point responses.

**Major comments**

1. All the problems in the theory part arise out of the fact that you apply the omega/f correction. My recommendation therefore is to present also the results for mid-frequency assumption (omega » f), for figure 4 in the main text, for figures 5-9 in the appendix. That would allow the reader to estimate for herself/himself whether the assumptions which need to be made (at least for current satellites) are important to the findings. Are they just amplifying the shears in an about uniform way or are they contributing to the shape of the distributions?

   **Response:** Sure. According to Eq. (4-5),

   $$\frac{\partial u_j'}{\partial z} = \frac{g}{N} m_j \left(1 - f^2/\widehat{\omega}_j^2\right)^{-1/2} \tilde{T}_j e^{i(\varphi_j + \pi)}, \tag{4}$$

   $$\frac{\partial v_j'}{\partial z} = \frac{f}{\widehat{\omega}_j} \frac{g}{N} m_j \left(1 - f^2/\widehat{\omega}_j^2\right)^{-1/2} \tilde{T}_j e^{i(\varphi_j + \pi/2)} \tag{5}$$

and the mid-frequency assumption $\widehat{\omega}_j \gg f$, we get $f/\widehat{\omega}_j \approx 0$ and

$$\frac{\partial u_j'}{\partial z} \approx m_j \frac{g}{N} \tilde{T}_j e^{i(\varphi_j + \pi)}, \tag{15}$$

$$\frac{\partial v_j'}{\partial z} = 0. \tag{16}$$

The GW-perturbed shears are contributed only by $\partial u_j'/\partial z$. Under the assumption of $f/\widehat{\omega}_j \neq 0$, from Eq. (4-5), the GW-perturbed shear is,

$$S_A = \sqrt{\left(\frac{\partial u_j'}{\partial z}\right)^2 + \left(\frac{\partial v_j'}{\partial z}\right)^2} = \frac{g m_j}{N} \left(1 - f^2/\widehat{\omega}_j^2\right)^{-1/2} \left(1 + f^2/\widehat{\omega}_j^2\right)^{-1/2}.$$

Under the mid-frequency assumption of $f/\widehat{\omega}_j \approx 0$, from Eq. (4a), the GW-perturbed shear is,

$$S_B = \left|\frac{\partial u_j'}{\partial z}\right| = \frac{gm_j}{N}.$$

Thus, the ratio between $S_B$ and $S_A$ is

$$R = \left[\left(1 - f^2/\widehat{\omega}_j^2\right)/\left(1 + f^2/\widehat{\omega}_j^2\right)\right]^{1/2}. \tag{17}$$

The ratio $R$ is less than 1 if $f/\widehat{\omega}_j \neq 0$ and is equal to 1 if $f/\widehat{\omega}_j = 0$. Thus, the medium frequency reduces the GW-perturbed shears as compared to that of $f/\widehat{\omega}_j \neq 0$. Fig. R1 shows the variations of the ratio ($R$) with vertical and horizontal wavelength at several latitudes for the background temperatures of 180 K (upper row) and 280 K (lower row). It can be seen that the ratio ($R$) decreases sharply with the decreasing vertical wavelengths, especially at high latitudes. The ratio ($R$) is approximately independent of background and thus the buoyancy frequency.

[Figure]

**Fig. R1.** The variations of the ratio (R) with vertical and horizontal wavelength at several latitudes for the background temperatures of 180 K (upper row) and 280 K (lower row). The contour interval is 0.025.

Following your suggestion, we performed same calculation of the GW-induces shears under the mid-frequency assumption. The results are shown in below and are added in the discussion part of the text.

[revised manuscript text omitted]

2. Rewrite the theory part and properly name the assumptions (see also detailed comments below). How does your picture change, if you assume that the different waves you superpose have different propagation directions? The following more in the line of suggestions, which may be beyond the scope of the paper: It would be nice to have a Monte Carlo simulation showing the effect of superposing waves of different omega/f and different propagation direction, but I admit that the result would again depend on some assumptions about the real distributions. You could also try to use simulated observations through the model (just sampling, no radiative transfer) to get some idea on the size of effects. Maybe it also would be worthwhile to assume first that one could infer propagation direction as well, as in principle such instruments could be build.

**Response:** Thanks for your good suggestion. According to your suggestion, we have clarified the assumptions in the theory part (Sec. 2). The main revisions are:

(1) The assumptions on GW propagation directions: These monochromatic GWs may not propagate in the same direction. However, from two adjacent GW profiles, one cannot get the actual horizontal wavenumber ($k_{j,hr}$) of each monochromatic, but can get the projection of actual wave

vector in the along-track direction ($k_{j,ha}$). The inconsistency between $k_{j,hr}$ and $k_{j,ha}$ introduces uncertainties of GW-perturbed shears, which are discussed in Sec. 2.3. Thus, $u'$ can be expressed as the vector sum of the projections of all the actual monochromatic GWs in the along-track direction and formulated as,

$$u' = \sum_j u'_j = \sum_j \tilde{u}_j e^{i\varphi_j} = \frac{g}{N} \sum_j \left[ \left( 1 - f^2/\hat{\omega}_j^2 \right)^{-1/2} \tilde{T}_j e^{i(\varphi_j + \pi/2)} \right].$$

(2) Due to the uncertainties in determining the GWs' propagation direction, only the magnitudes of GW-perturbed shears are analyzed in this work.

(3) It should be noted that the $k_{j,h}$ is the projection of the actual GW's horizontal wavenumber in the along-track direction and is underestimated. This underestimates the intrinsic frequency and thus overestimates the wind and shears based on Eq. (6-9) and will be discussed detail in Sec. 2.3.

(4) However, GWs may not propagate only in the along-track direction. Our assumption will introduce some uncertainties and will be discussed below.

It is a good suggestion to perform the Monte Carlo simulation to show the effect of superposing waves of different omega/f and different propagation direction. We plan to perform the simulation in our future work.

3.  Discuss the influence of noise. In particular it is known that SABER temperature retrievals have a noise problem at the summer mesopause. That is stated on the SABER web-page and also discussed for GW analyses by Ern et al. 2018. The results for this particular region have to be treated with care. At least some discussion must be included in the paper.

    **Response:** We have added some discussion on this point in Sec. 3.1 as below:

    The GW-perturbed shears might be influenced by temperature uncertainties of SABER measurements, which are much larger at around the cold summer mesopause (Remsberg et al., 2008). We removed waves with vertical wavelengths shorter than 5 km, to minimize the noise introduced by uncertainties of SABER measurements (Ern et al., 2011, 2018).

4.  The paper must be self-explaining. Explain how you arrive at your vertical wavelengths and amplitudes. Anyway: You should not use FFT over the whole altitude regime. There is so much Doppler shift, critical level filtering and oblique propagation going on that the case where you observe the same gravity wave at all altitudes and furthermore with similar vertical wavelength is the absolute exception.

    **Response:** You are right. For a single GW, its amplitudes and wavelengths may vary when they propagate due to the Doppler shift, critical level filtering and oblique propagation, etc. A GW profile

extracted from observations is a superposition of many GWs, which may come from different locations and have height dependent amplitudes and wavelengths. We have replaced the DFT with discrete wavelet transform (DWT, Torrence & Compo, 1998) in this version. Such that we can get the height dependent vertical wavelengths and amplitudes. Here we show in Figure 1(DWT) the GW-perturbed shears derived through DWT. For comparison, we show in Figure 1(DFT) the GW-perturbed shears derived through DFT. Comparing between Figure 1(DWT) and Figure 1(DFT), similar GW-perturbed shears can be obtained from both methods. However, the slightly difference occurs at z=85-95 km, where the small-scale perturbations of GW-perturbed shears are clear for the DFT method but not for the DWT method. On the statistical sense, the GW-perturbed shears exhibit similar magnitudes and latitude-height and time-height patterns (see text for detailed results).

We have revised the method in the Sec.2.2.

Torrence, C. and Compo, G. P.: A Practical Guide to Wavelet Analysis, Bull. Am. Meteorol. Soc., 79(1), 61–78, doi:10.1175/1520-0477(1998)079<0061:APGTWA>2.0.CO;2, 1998.

[Figure]

**Figure 1(DWT):** Synthetic temperature perturbation profiles (a, $T'_{w1}(z)$ and $T'_{w2}(z)$ are represented by black and red lines, respectively) and the corresponding winds (b, zonal; c, meridional) and shears (d, zonal; e, meridional). The winds and shears are, respectively, calculated from the prescribed amplitudes and wavenumbers (black line, labelled as "th") and reconstructed by spectral decomposition method (red dashed line, labelled as "rc"). All panels have the same y-axis scale.

[Figure]

**Figure 1(DFT):** Same caption as Figure 1(DWT) but for the results derived from DFT.

5.  In particular: how do you deal with tides? All tides must be removed before the analysis as they are global wave modes with wind amplitudes maximizing at a different latitude than temperature amplitudes.

    **Response:** According to Preusse et al. (2002) and Ern et al. (2004, 2018), due to the slowing precessing of the SABER measurement (a full cycle is about 120 days), migrating tides will appear as stationary zonal wave patterns if data from ascending and descending nodes are taken separately. By removing these stationary wave patterns separately for ascending and descending nodes, tides can be removed from the observed temperature fluctuations.

    In the first paragraph of Sec. 2.3, we present the method of extraction methods of GW profiles: "First, the daily SABER temperature profiles $T(z)$ in a latitude band of 5° are selected. Second, at each altitude, these selected data are fitted by harmonics with zonal wavenumbers ranging from 0 to 6, which are mainly planetary waves and non-migrating tides and are removed from $T(z)$ to get the residual temperature $T_r(z)$. The component of wavenumber 0 is considered as the zonal mean temperature $\bar{T}(z)$. Due to the slowing precessing of the SABER measurement (a full cycle is about 120 days), migrating tides will appear as stationary zonal wave patterns if data from ascending and descending nodes are taken separately. The above two steps are applied on both the ascending and descending nodes, respectively, such that it minimizes the influences of migrating tides on the residual temperature $T_r(z)$ and thus GW profiles (Preusse et al., 2002, Ern et al., 2004, 2018)".

6.  Discussions: for GWMF I think it is a valid assumption that wind modulation causes most of the relevant waves to propagate opposite to the winds. You can find some of this expressed in AIRS

results and also GW resolving global models. However, for shear I don't think that is further valid. Therefore, I would include some of the discussion (very much shortened) in sec. 2.

**Response:** For the wave propagation directions, we have added some discussions in Sec. 2 as below:

These monochromatic GWs may not propagate in the same direction. However, from two adjacent GW profiles, one cannot get the actual horizontal wavenumber ($k_{j,hr}$) of each monochromatic, but can get the projection of actual wave vector in the along-track direction ($k_{j,ha}$). The inconsistency between $k_{j,hr}$ and $k_{j,ha}$ introduces uncertainties of GW-perturbed shears, which are discussed in Sec. 2.3. Thus, $u'$ can be expressed as the vector sum of the projections of all the actual monochromatic GWs in the along-track direction and formulated as,

$$u' = \sum_j u'_j = \sum_j \tilde{u}_j e^{i\varphi_j} = \frac{g}{N} \sum_j \left[ \left(1 - f^2/\widehat{\omega}_j^2\right)^{-1/2} \tilde{T}_j e^{i(\varphi_j + \pi/2)} \right].$$

**Specific comments**

1.  P2L4 It is now accepted ... A larger part of the shear and the winds is associated with tides. You need to mention them here. In particular sporadic E is primarily associated with tides (e.g. Arras et al., 2009, Arras et al. 2018).

    **Response:** We have revised "The large winds and shears play important…" as "The large winds and shears are associated with tides and GWs and play important…". Moreover, we have added the references of Arras et al. (2009), Arras and Wickert (2018), and Jacobi et al. (2019) to show the relations between sporadic E and tides.

    Arras, C., Jacobi, C., and Wickert, J.: Semidiurnal tidal signature in sporadic E occurrence rates derived from GPS radio occultation measurements at higher midlatitudes, Ann. Geophys., 27, 2555–2563, https://doi.org/10.5194/angeo-27-2555-2009, 2009.

    Arras, C. and Wickert, J.: Estimation of ionospheric sporadic E intensities from GPS radio occultation measurements, J. Atmos. Sol.-Terr. Phys., 171, 60–63, https://doi.org/10.1016/j.jastp.2017.08.006, 2018.

    Jacobi, C., Arras, C., Geißler, C., and Lilienthal, F.: Quarterdiurnal signature in sporadic E occurrence rates and comparison with neutral wind shear, Ann. Geophys., 37, 273–288, https://doi.org/10.5194/angeo-37-273-2019, 2019.

2.  P2L9 Please explain this equation: what is the motivation? Static stability of saturated mid frequency GW?

**Response:** "Based on the theory of dynamic instability, Liu (2007) showed that the maximum wind shears derived from $S = 2N$ ($N$: the buoyancy frequency, $S$: the vertical shear) coincide well with the observed large wind shears" have been revised as:

Based the definition of the Richardson number $Ri = N^2/S^2$, the atmosphere is dynamic stable when $Ri > 1/4$ and dynamic instable when $Ri \leq 1/4$. Here $N^2 = g/\bar{T}\left(d\bar{T}/dz + g/c_p\right)$ is the static stability, $S^2 = (\partial u/dz)^2 + (\partial v/dz)^2$ is the wind shear; $\bar{T}$ is the background temperature; $g$ and $c_p$ are the gravitational acceleration and specific heat for dry air at constant pressure, respectively; $u$ and $v$ are the zonal and meridional winds, respectively. The threshold of dynamic instability is $Ri = 1/4$, which means that the maximum wind shear allowed by the background static stability should be $S = 2N$. According to $S = 2N$, Liu (2007) showed that the maximum wind shears coincide well with the observed large wind shears.

3. P2L34 "poorly known" Motivate. What are the limitations of the wind observations? Noise? Altitude resolution? Combination to vector winds?

    **Response:** We have clarified this point as "However, the global characteristics of shears are poorly known due to either the limited altitude coverage or the altitude resolution or large noises of the satellite observations".

4. P3 Equ 1: Under the assumption of conservative propagation and no refraction. Reasonable for some waves in a limited altitude range, but must not be taken over the whole MA as is done below.

    **Response:** We have added the assumption as "For a conservative propagation and no refraction monochromatic linear GW, it has the form of...".

5. P3L20/L25 These polarization relations are for winds u parallel and v perpendicular to the wavevector of the GW. You can do the assumption, but must say so in the text.

    **Response:** We have revised this as "$u_j'$ and $v_j'$ are the horizontal wind perturbations parallel and perpendicular to the wavevector of the GW, respectively".

6. P4 Equ. 6 and Equ. 7 These equations make the much stronger assumption that the wave vectors are all pointing in the same direction. While that seems a plausible assumption, if you look for the largest amplitudes (preferential orientation against the mean wind), I don't see why this should be the case for wind shear. You showed in Equs 4 and 5 that shear is proportional to m as well as to the amplitude. Handwaving, these two effects should largely cancel with regards to phase speed. Thus, you would not expect a direction preference.

**Response:** You are right. The wave vectors are not pointing in the same direction and determined by the horizontal wavenumber $k_{j,h}$. However, we cannot get the actual wave vector using a GWs' pair but get the projection of actual wave vector in the along-track direction. Thus, Eq. (6-7) are the vector sum of the projection of all the actual monochromatic GWs in the along-track direction. The inconsistency between the actual wave vector and the projected wave vector induces the uncertainties of shears as discussed in Sec. 2.3

According to your comment, we clarified this point before Eq. (6) as below:

These monochromatic GWs may not propagate in the same direction. However, from two adjacent GW profiles, one cannot get the actual horizontal wavenumber ($k_{j,hr}$) of each monochromatic GW, but can get the projection of actual wave vector in the along-track direction ($k_{j,ha}$). The inconsistency between $k_{j,hr}$ and $k_{j,ha}$ introduces uncertainties of GW-perturbed shears, which are discussed in Sec. 2.3. Thus, $u'$ can be expressed as the vector sum of the projections of all the actual monochromatic GWs in the along-track direction and formulated as,

$$u' = \sum_j u'_j = \sum_j \tilde{u}_j e^{i\varphi_j} = \frac{g}{N} \sum_j \left[ \left( 1 - f^2 / \hat{\omega}_j^2 \right)^{-1/2} \tilde{T}_j e^{i(\varphi_j + \pi/2)} \right]. \tag{6}$$

7. P4 Equ. 8 and 9 then are not valid

**Response:** Eq. (8-9) are the vertical gradient of $u'$ and $v'$. Due to the linearity of vector sum operation, the partial derivation the vector sum is equivalent to the vector sum of all the components. To clarify this point, we revised Eq. (8-9) as:

$$\frac{\partial u'}{\partial z} = \frac{\partial}{\partial z} \left( \sum_j u'_j \right) = \sum_j \frac{\partial u'_j}{\partial z} = \frac{g}{N} \sum_j \left[ m_j \left( 1 - f^2 / \hat{\omega}_j^2 \right)^{-1/2} \tilde{T}_j e^{i(\varphi_j + \pi)} \right], \tag{8}$$

$$\frac{\partial v'}{\partial z} = \frac{\partial}{\partial z} \left( \sum_j v'_j \right) = \sum_j \frac{\partial v'_j}{\partial z} = \frac{g}{N} \sum_j \left[ \frac{fm_j}{\hat{\omega}_j} \left( 1 - f^2 / \hat{\omega}_j^2 \right)^{-1/2} \tilde{T}_j e^{i(\varphi_j + \pi/2)} \right]. \tag{9}$$

8. P5 Equ 12 is fine, but is introduced in a way that makes belief that you could get the horizontal wavenumber from the satellite. Instead it is the one projected to the track. This will introduce overestimate of lambda_h, underestimate of k_h and thus underestimate of nhat{nomega}. You introduce some of these arguments below, but this must be more clearly ordered.

**Response:** We have added a note to clarify this point as "It should be noted that the $k_{j,h}$ is the projection of the actual GW's horizontal wavenumber in the along-track direction and is underestimated. This underestimates the intrinsic frequency and thus overestimates the wind and shears based on Eq. (6-9) and will be discussed detail in Sec. 2.3."

9. P5L8 No you can not! This is not the zonal and meridional component!

**Response:** We have revised this as "we can get $u'_j$ and $v'_j$ and their shears".

10. P5L15 And why should the wavevector point along the orbital track? That is as unlikely.

   **Response:** This point has been clarified as "Here we assume that $u'_j$ and $v'_j$ are the winds along and cross the orbit track directions, respectively". Moreover, we have added "However, GWs may not propagate only in the along-track direction. The assumption will introduce uncertainties and will be discussed below" in the end of this paragraph.

11. P5L25 There is no reason to believe that the waves at 20 km have anything to do with the waves at 100 km. Please explain your method. Phase differences should be gained from "local" phases determined for a limited altitude range.

   **Response:** You are right. We used discrete wavelet transform to decompose GWs and reconstruct GW-perturbed shears. Such that we can get the height dependent amplitudes and vertical wavelengths. The local phase shift is computed by the cospectrum of two adjacent GW profiles (Ern et al., 2004, Alexander, 2008). In the text we have revised as:

   The first step is to evaluate the amplitude and vertical wavenumbers of each GW profile by the method of discrete wavelet transform (short for DWT, Torrence & Compo, 1998) such that we can get the height dependent amplitudes and vertical wavelengths. For the GW profiles of $T'_{w1}(z)$ and $T'_{w2}(z)$, then at each height and vertical wavelength ($\lambda_z = 2\pi/m$), their DWT are $\hat{T}_{w1}(z, \lambda_z)$ and $\hat{T}_{w2}(z, \lambda_z)$. Then, their cospectrum $C_{1,2}$ is computed,

$$C_{1,2} = \hat{T}_{w1}\hat{T}^*_{w2} = \tilde{T}_{w2}\tilde{T}_{w2}\exp(i\Delta\varphi_{1,2}), \tag{13}$$

Here, $\tilde{T}_{w1} = |\hat{T}_{w1}|$ and $\tilde{T}_{w2} = |\hat{T}_{w2}|$ are the amplitudes of $T'_{w1}(z)$ and $T'_{w2}(z)$, $\hat{T}^*_{w2}$ is the complex conjugation of $\hat{T}_{w2}$. The phase shift is calculated by $\Delta\varphi_{1,2} = \tan^{-1}[\text{Im}(C_{1,2})/\text{Re}(C_{1,2})]$. When performing DWT, we restrict the vertical wavelength ranging from ~5 km to ~30 km for a vertical extent of 90 km, which is the height coverage (18-108 km) of the SABER temperature profiles.

   It should be noted that, the GW-perturbed shears derived from DWT and DFT are very similar. Please see the response to the **Major commets#4**. Thus, changing the method from DFT to DWT does not change our conclusions.

12. P8L6 can you please make these number codes plain text 01 July - 31 July which Year?

   **Response:** We have revised this as "(e.g., 31 days during 01 July-31 July in 2018 and 62 days during 29 June-30 August in 2018). The same is true during the intervals of 06 May-27 June and 29 June-30 August in 2018, respectively, at latitudes of 52.5°S-52.5°N. The GW-perturbed shears during the continuous two yaw cycles (06 May-27 June and 29 June-30 August)". We also revised the number codes as plain text at other places in the text.

13. P8L15 This is a region problematic for retrievals due to very cold temperatures and low IR signals (cf. Remsberg, Ern). Some of what you see there may be real, but some is very likely due to increased noise. A discussion is needed.

**Response:** We have added a discussion on this point in the end of this paragraph as "The GW-perturbed shears might be influenced by temperature uncertainties of SABER measurements, which are much larger at around the cold summer mesopause (Remsberg et al., 2008). We removed waves with vertical wavelengths shorter than 5 km, to minimize the noise induced by uncertainties of SABER measurements (Ern et al., 2011, 2018)".

14. P9L16 You mean the zonal mean of the individual shear profiles? Please be precise, as the zonal mean shear due to GWs should be zero. But then: what does the lidar provide? Average of abs.(shears) or shear of average?

**Response:** This is my mistake and should be clarified. Both the means shears and the zonal mean shears are the means of the shear's magnitude (i.e., mean of abs.(shears), no direction) throughout the manuscript. We have added a note on Equation (10) as:

Finally, the magnitude of GW-perturbed shear can be written as

$$S = \sqrt{(\partial u'/\partial z)^2 + (\partial v'/\partial z)^2}. \tag{10}$$

Due to the uncertainties in determining the GWs' propagation direction, only the magnitudes of GW-perturbed shears are analyzed in this work."

15. P9L25 You do not discuss here the visibility filter, which also should reduce amplitudes

**Response:** We have added this point in the text as "(3) the observational filter should underestimate the amplitudes of GWs, especially for GWs with shorter vertical wavelengths (Ern et al., 2018), and thus underestimate the GW-perturbed shears".

16. P12L17 And also on what you emphasize due to the scaling with m: shear (short vertical wavelength), amplitudes (mid lz) or GWMF (long lz).

**Response:** We have added the point of "influences of the vertical wavenumber m on the differences of the heights at which phase shifts occur" in the text and revised as:

"However, the heights at which phase shifts occur are different. One reason is that the background temperature and static stabilities (Liu et al., 2020). The other reason is that the GWMF is inverse proportion to the vertical wavenumber $m$, while the GW-perturbed shear is proportion to the $m$. The resulting height of the phase shift of GWMF at a lower height than that of GW-perturbed shears since $m$ increases with height below z=90 km (Ern et al. (2018)."

17. P12L21 also Schmidt et al.

    **Response:** We have added "Schmidt et al. (2016)" here.

Schmidt, T., Alexander, P., and de la Torre, A.: Stratospheric gravity wave momentum flux from radio occultations, J. Geophys. Res.-Atmos., 121, 4443–4467, https://doi.org/10.1002/2015 JD024135, 2016.

18. P13L11 SABER "sees" GWs with horizontal wavelengths > 200km, much the same as WACCM, though with somewhat reduced amplitude. I would check by mid-frequency approach, whether the scaling and profile distance is really of importance. In addition, as you said above, if there is an influence, it would be an overestimation.

    **Response:** Below are Equation (8) and (12),

$$\frac{\partial u'}{\partial z} = \sum_j \frac{\partial u'_j}{\partial z} = \frac{g}{N} \sum_j \left[ m_j \left(1 - f^2/\widehat{\omega}_j^2\right)^{-1/2} \tilde{T}_j e^{i(\varphi_j + \pi)} \right], \tag{8}$$

$$\widehat{\omega}_j^2 = N^2 \frac{k_{j,h}^2}{m_j^2} + f^2. \tag{12}$$

According to Equation (8), we can see that the longer horizontal wavelength (i.e., smaller horizontal wavenumber) induces smaller intrinsic frequency $\widehat{\omega}_j$. This increases the magnitude of $\partial u'/\partial z$ and overestimates the GW-perturbed shears based on Equation (8).

However, what we want to express is that the longer sampling distances miss the GWs with shorter horizonal wavelengths, which might also contribute the GW-perturbed shears.

We did not express this point clearly in the text and revised it in the new version as "The longer sampling distances miss the GWs with shorter horizonal wavelengths, which might also contribute the GW-perturbed shears and thus reduces the magnitudes of GW-perturbed shears".

**Technical corrections**

1. P1L28 ... (or more precisely vertical wind shears) ...

    **Response:** This has been revised as "…large horizontal winds and their vertical shears (or more precisely vertical wind shears, short for shears)…".

2. P1L30 ... and from ground-based ...

    **Response:** We have added "from" before "ground-based".

3. P2L2 ... strong gravity wave (GW) activity.

    **Response:** We have revised "gravity waves (GWs)" as "gravity wave (GW)".

4. P2L2 You probably want to refer to a generally knowledge on GWs rather than on evidence from the aforementioned papers for the sources: Most prominent sources of GWs are convection, orography and jets and fronts in the troposphere as well as spontaneous adjustment and secondary wave generation in the stratosphere. Amplitudes and shears increase when the GWs propagate ...

   **Response:** Thanks. We have revised "These GWs had increasing wave amplitude and shears when they propagated into the lower thermosphere" as "Most prominent sources of GWs are convection, orography and jets and fronts in the troposphere as well as spontaneous adjustment and secondary wave generation in the stratosphere (Fritts and Alexander, 2003). Amplitudes and shears increase when the GWs propagate into the lower thermosphere".

5. P2L13 driving -> are the cause of

   **Response:** We have revised "important in driving" as "are the cause of".

6. P2L17 reproduced the large wind shears seen in the observations.

   **Response:** We have revised "reproduced the large wind shears" as "reproduced the large wind shears seen in the observations".

7. P2L22 Observations ... were made by sounding ... for a very limited number of locations only and hence cannot ...

   **Response:** We have revised "Large winds and shears have been observed both by sounding rocket and ground based lidar and radar, all locally and cannot provide a global morphology" as "Observations on large winds and shears were made by sounding rocket and ground-based lidar and radar for a very limited number of location only and hence cannot provide a global morphology".

8. P2L24 still challenging

   **Response:** We have revised "still a challenging" as "still challenging".

9. P3L4 of deriving shears from GW analyses of temperature observations.

   **Response:** We have revised "…. of deriving shears from GWs, while the GWs are derived from temperature profiles. The temperature profiles, which are measured by the Sounding…" as "…. of deriving shears from GW analyses of temperature observations…".

10. P3L6 SABER luckily still measures: Temperature profiles measured by .... from 2002 to 2019 are used for this study providing an 18 year period.

**Response:** We have revised "The temperature profiles, which are measured by the Sounding of the Atmosphere using Broadband Emission Radiometry (SABER) instrument (Russell et al., 1999) onboard the TIMED satellite, have covered a period of 18 years (2002-2019) and are remarkably stable until now (Mlynczak et al., 2020)" as "Temperature profiles measured by the Sounding of the Atmosphere using Broadband Emission Radiometry (SABER) instrument (Russell et al., 1999) onboard the TIMED satellite from 2002 to 2019 are used for this study providing an 18-year period". Temperature profiles are remarkably stable until now (Mlynczak et al., 2020)."

11. P3L8 ... website based on Remsberg et al. (2008).

    **Response:** We have revised "...website (Remsburg et al., 2008)" as "...website based on Remsberg et al. (2008)".

12. P3L10 omit () at this point

    **Response:** We have deleted "(short for GW-induced shears)".

13. P4L3 In reality, usually spectrum of GWs is observed formed by superposition of several monochromatic GWs.

    **Response:** We have revised "In real atmosphere, a GW profile consists multiple spectra and can be regarded as a superposition of monochromatic GWs" as "In reality, usually spectrum of GWs is observed formed by superposition of several monochromatic GWs".

14. P5L7 and the amplitude and vertical wavenumber determined from the satellite observations for the individual monoc. GWs, we ...

    **Response:** We have revised "and the prescribed amplitude and vertical wavenumber of each monochromatic GW, we can get the zonal and meridional winds components and their shears" as "and the prescribed amplitude and vertical wavenumber, which will be determined from the satellite observation and described below, for the individual monochromatic GW, we can get $u_j'$ and $v_j'$ and their shears".

15. P6L26 https://en.wikipedia.org/wiki/Flowchart Please use different wording here

    **Response:** We replaced "flow chart" by "procedure" here and figure caption.

16. P7L20 a factor 0.2 would be underestimation overestimated by 20%?

    **Response:** We have revised "which are overestimated by a factor of 0.2" as "which are overestimated by 20%".

17. P8L9 comment -> common

    **Response:** We have revised "comment" as "common".

18. P9L15 slightly larger

    **Response:** We have revised "slight larger" as "slightly larger".

[revised manuscript text omitted]

---

## Short Comment (SC2) · 21 Aug 2020

**Responses to Referee#2:**

SABER temperature profiles are used as input to a calculation of gravity wave parameters, including the vertical and horizontal wavenumbers, the intrinsic frequency, and the horizontal velocity perturbations associated with each wave component in the decomposed wave spectrum. The goal is to estimate the vertical shears associated with the wave spectrum over the extended time periods and greater geographical extent provided by the satellite data.

The presentation of the material and methodology is clear, the discussion of past work is useful and very complete, and the figures represent the results well. Nonetheless, there are several major questions related to the validity of the analysis.

The shears obtained from the SABER data are calculated and are not directly measured. A broader global climatology, including the seasonal variations, is valuable to the community and represents a worthwhile goal. There are a number of critical assumptions required, however, for the calculations to be valid and meaningful. Those assumptions are especially likely to break down in the altitude range around the mesopause, which is where the largest shears occur that are a focus of the analysis. The assumptions are not adequately addressed in the paper, and the effect of those assumptions are therefore also not addressed. The recommendation is to return the manuscript to the authors for revisions.

**Response:** Thanks for your valuable comments on our manuscript. Following your comments below, we have addressed the assumptions and used wavelet transform to reconstruct the wind shears in this version. Please see below for detailed responses.

**Specific comments are as follows**

1.  Tides are in some sense global-scale gravity waves constrained by spherical boundary conditions, but they clearly do not conform to the simple linearized polarization relations given in equations (2) and (3). The authors introduce some filtering to reduce or eliminate larger-scale modes, but it is not clear that the tidal modes, especially the higher order modes, are eliminated prior to extracting the gravity wave spectral components. This is especially important since the altitude range where the large shears occur near the mesopause is also an altitude range where a number of tidal modes have large amplitudes, perhaps not coincidentally?

    **Response:** According to Preusse et al. (2002) and Ern et al. (2004, 2018), due to the slowing precessing of the SABER measurement (a full cycle is about 120 days), migrating tides will appear as stationary zonal wave patterns if data from ascending and descending nodes are taken separately.

By removing these stationary wave patterns separately for ascending and descending nodes, tides can be removed from the observed temperature fluctuations.

In the first paragraph of Sec. 2.3, we present the method of extraction methods of GW profiles: "First, the daily SABER temperature profiles $T(z)$ in a latitude band of 5° are selected. Second, at each altitude, these selected data are fitted by harmonics with zonal wavenumbers ranging from 0 to 6, which are mainly planetary waves and non-migrating tides and are removed from $T(z)$ to get the residual temperature $T_r(z)$. The component of wavenumber 0 is considered as the zonal mean temperature $\bar{T}(z)$. Due to the slowing precessing of the SABER measurement (a full cycle is about 120 days), migrating tides will appear as stationary zonal wave patterns if data from ascending and descending nodes are taken separately. The above two steps are applied on both the ascending and descending nodes, respectively, such that it minimizes the influences of migrating tides on the residual temperature $T_r(z)$ and thus GW profiles (Preusse et al., 2002, Ern et al., 2004, 2018)".

2. A further concern is that the region of large shears near the mesopause is also a region of large winds that represents a height range with critical levels for a large fraction of the gravity wave spectrum, especially since the background winds rotate with height through that altitude range, so there are critical levels for a broad range of propagation directions. It appears questionable to apply the linearized polarization relations in regions with critical levels where the vertical wavelength is changing rapidly with height and the dynamics are almost certainly nonlinear. The validity of the analysis in those regions is not discussed.

**Response:** Thanks for your good suggestion. According to your suggestion, we have added a paragraph in the end of Sec. 2.3 as below:

The large winds and shears in the MLT region, where both tides and GWs reach large amplitudes and may interact nonlinearly (Fritts and Vincent, 1987; Li et al., 2009; Liu et al., 2014a) and break (Fritts et al., 2004; Liu and Vadas, 2013; Vadas and Liu, 2013). The large winds may rotate with height and act as critical levels, which filter out a broad spectral range of GWs. These filtered GWs may break and deposit their momenta in the background atmosphere, which create body force to general secondary GWs. Moreover, the vertical wavelengths of these filtered GWs change rapidly with height and in the nonlinear regime. These nonlinear GWs may also produce large winds and shears around the mesopause regions. However, these nonlinear GWs cannot be described by linear GW theory proposed here. Thus, the GW-perturbed shears underestimate the actual shears in the MLT region due to the unrepresented nonlinear GWs.

3. The large shear region is also a region where many gravity waves break, producing turbulence and, in some cases, secondary waves, as discussed in at least one of the references cited in the

manuscript. The critical levels mentioned in the previous comment can produce breaking, but breaking waves can also be a result of the amplitude growth with height, which by itself leads to amplitudes large enough to produce such effects at heights near the mesopause.

**Response:** We added the discussion on the limitation of our method, which is only valid for the linear GW. For the GW break and turbulence, as well as the secondary waves, they are in the nonlinear regime and out of the scope of the linear GW theory. This point has been addressed in the response to comment#2 and added a paragraph in the end of Sec. 2.3.

4. If we assume a localized source in the lower atmosphere that produces a spectrum of waves, then the lowest-frequency gravity waves propagate at a lower elevation angle while the higher-frequency waves propagate at angles closer to vertical. A vertical profile extending from the ground to the lower thermosphere is therefore unlikely to have the contents from a single set of monochromatic waves contributing at all heights. As an example, several studies have been published in the past with special cases in which waves have been traced from the troposphere to the thermosphere when there has been an especially strong source, such as a strong line of thunderstorms, but those waves become displaced horizontally as they propagate vertically so that a vertical sample would only intercept the wave packet in a small part of the complete altitude range. In the general case the altitude profile measured by the satellite will likely have contributions from an extended range of geographically-distributed sources and therefore also different phasing dominating the vertical wavelength contribution in each part of the altitude range. If so, it is not clear that an individual wavelength contribution extracted via a DFT represents the wave structure accurately and applying the polarization relations as if that particular component represents a single monochromatic wave seems questionable.

**Response:** You are right. For a single GW, its amplitudes and wavelengths may vary when they propagate due to the Doppler shift, critical level filtering and oblique propagation, etc. A GW profile extracted from observations is a superposition of many GWs, which may come from different locations and have height dependent amplitudes and wavelengths. We have replaced the DFT with discrete wavelet transform (DWT, Torrence & Compo, 1998) in this version. Such that we can get the height dependent vertical wavelengths and amplitudes. Here we show in Figure 1(DWT) the GW-perturbed shears derived through DWT. For comparison, we show in Figure 1(DFT) the GW-perturbed shears derived through DFT. Comparing between Figure 1(DWT) and Figure 1(DFT), similar GW-perturbed shears can be obtained from both methods. However, the slightly difference occurs at z=85-95 km, where the small-scale perturbations of GW-perturbed shears are clear for the DFT method but not for the DWT method. On the statistical sense, the GW-perturbed shears exhibit similar magnitudes and latitude-height and time-height patterns (see text for detailed results).

We have revised the method in the Sec.2.2.

Torrence, C. and Compo, G. P.: A Practical Guide to Wavelet Analysis, Bull. Am. Meteorol. Soc., 79(1), 61–78, doi:10.1175/1520-0477(1998)079<0061:APGTWA>2.0.CO;2, 1998.

[Figure]

**Figure 1(DWT):** Synthetic temperature perturbation profiles (a, $T'_{w1}(z)$ and $T'_{w2}(z)$ are represented by black and red lines, respectively) and the corresponding winds (b, zonal; c, meridional) and shears (d, zonal; e, meridional). The winds and shears are, respectively, calculated from the prescribed amplitudes and wavenumbers (black line, labelled as "th") and reconstructed by spectral decomposition method (red dashed line, labelled as "rc"). All panels have the same y-axis scale.

[Figure]

**Figure 1(DFT):** Same caption as Figure 1(DWT) but for the results derived from DFT.

5. Finally, since only the along-track direction is sampled, the horizontal wavelength will be overestimated for any waves propagating in an off-track direction. The resulting underestimate of the horizontal wavenumber artificially increases the value of omega prime (equation 12) and leads to an overestimate of the vertical shear (equations 8 and 9). Given that, it seems possible that the apparent seasonal variations and latitudinal variations in the derived shears are a reflection of seasonal and latitudinal changes in preferred wave propagation directions rather than actual changes in the magnitudes of the shears.

**Response:** Following your comment, we have added this point in the last paragraph of Sec. 4.2 as "The GW-perturbed shears, which is derived from the projection of actual GWs in the along-track direction, are overestimated as compared to the actual GW-perturbed shears. The extent of overestimation depends the actual GW propagations, which have seasonal and latitudinal preferences. Thus the seasonal and latitudinal variations of GW-perturbed shears may be influenced by the preferred GWs propagation directions to some extent".

6. A minor comment is that the title of the paper, which uses the terms "gravity wave induced shears", suggests an effect in which waves are accelerating the flow to produce the shears, i.e., inducing a shear by momentum deposition for example, but that is not what the manuscript describes since the analysis focuses only on the superposition of shears that are naturally a part of the wave spectrum without any mean-flow acceleration.

**Response:** You are right. We changed the "gravity wave induced shears" to "gravity wave perturbed shears" throughout the text.

[revised manuscript text omitted]

---

## Author Comment (AC1) · 26 Aug 2020

Xiao Liu has submitted the point-to-point resonponse to RC1 and on behalf of all Co-Authors. Please see the responses in the Supplement, which containes one PDF file combined by the responses and reviseved manuscript.

Please also note the supplement to this comment:
https://acp.copernicus.org/preprints/acp-2020-515/acp-2020-515-AC1-supplement.pdf

---

## Author Comment (AC3) · 26 Aug 2020

Xiao Liu has submitted the point-to-point resonponse to RC2 and on behalf of all Co-Authors. Please see the responses in the Supplement, which contains one PDF file combined by the responses and reviseved manuscript.

Please also note the supplement to this comment:
https://acp.copernicus.org/preprints/acp-2020-515/acp-2020-515-AC3-supplement.pdf

---

## Author Response (AR1)

**Responses to Referee#1:**

The paper by Liu et al. provides climatologies of vertical wind shear. This can be obtained essentially from temperature amplitude multiplied by vertical wave number and is thus different from climatologies of GW amplitudes (without the m) and GW momentum flux which scales m*T^2. Both m and T-amplitude can, in principle gained from vertical profiles obtained from satellites at good accuracy. Corrections are applied for the ratio of intrinsic frequency to Coriolis parameter. For these, however, quite a number of assumptions have to be made. These are either not explained at all or not very well in the paper. A further major point is that the scaling with m emphasizes the noise of the observations. That fact needs to be discussed. My recommendation is therefore to return to the authors for major revisions.

**Response:** Thanks for your valuable comments on our manuscript. According to your comments, the main revisions are: (1) wavelet transformation method is used to decompose GWs and reconstruct shears; (2) the results under the medium frequency assumption are provided. These revisions do not change our conclusion. Please see below for the point-to-point responses.

**Major comments**

1. All the problems in the theory part arise out of the fact that you apply the omega/f correction. My recommendation therefore is to present also the results for mid-frequency assumption (omega » f), for figure 4 in the main text, for figures 5-9 in the appendix. That would allow the reader to estimate for herself/himself whether the assumptions which need to be made (at least for current satellites) are important to the findings. Are they just amplifying the shears in an about uniform way or are they contributing to the shape of the distributions?

   **Response:** Sure. According to Eq. (4-5),

$$\frac{\partial u'_j}{\partial z} = \frac{g}{N} m_j \left(1 - f^2/\widehat{\omega}_j^2\right)^{-1/2} \tilde{T}_j e^{i(\varphi_j + \pi)}, \tag{4}$$

$$\frac{\partial v'_j}{\partial z} = \frac{f}{\widehat{\omega}_j} \frac{g}{N} m_j \left(1 - f^2/\widehat{\omega}_j^2\right)^{-1/2} \tilde{T}_j e^{i(\varphi_j + \pi/2)} \tag{5}$$

and the mid-frequency assumption $\widehat{\omega}_j \gg f$, we get $f/\widehat{\omega}_j \approx 0$ and

$$\frac{\partial u'_j}{\partial z} \approx m_j \frac{g}{N} \tilde{T}_j e^{i(\varphi_j + \pi)}, \tag{15}$$

$$\frac{\partial v'_j}{\partial z} = 0. \tag{16}$$

The GW-perturbed shears are contributed only by $\partial u'_j/\partial z$. Under the assumption of $f/\widehat{\omega}_j \neq 0$, from Eq. (4-5), the GW-perturbed shear is,

$$S_A = \sqrt{\left(\frac{\partial u'_j}{\partial z}\right)^2 + \left(\frac{\partial v'_j}{\partial z}\right)^2} = \frac{g m_j}{N} \left(1 - f^2/\widehat{\omega}_j^2\right)^{-1/2} \left(1 + f^2/\widehat{\omega}_j^2\right)^{-1/2}.$$

Under the mid-frequency assumption of $f/\hat{\omega}_j \approx 0$, from Eq. (4a), the GW-perturbed shear is,

$$S_B = \left|\frac{\partial u'_j}{\partial z}\right| = \frac{g m_j}{N}.$$

Thus, the ratio between $S_B$ and $S_A$ is

$$R = \left[(1 - f^2/\hat{\omega}_j^2)/(1 + f^2/\hat{\omega}_j^2)\right]^{1/2}. \tag{17}$$

The ratio $R$ is less than 1 if $f/\hat{\omega}_j \neq 0$ and is equal to 1 if $f/\hat{\omega}_j = 0$. Thus, the medium frequency reduces the GW-perturbed shears as compared to that of $f/\hat{\omega}_j \neq 0$. Fig. R1 shows the variations of the ratio ($R$) with vertical and horizontal wavelength at several latitudes for the background temperatures of 180 K (upper row) and 280 K (lower row). It can be seen that the ratio ($R$) decreases sharply with the decreasing vertical wavelengths, especially at high latitudes. The ratio ($R$) is approximately independent of background and thus the buoyancy frequency.

[Figure]

**Fig. R1.** The variations of the ratio (R) with vertical and horizontal wavelength at several latitudes for the background temperatures of 180 K (upper row) and 280 K (lower row). The contour interval is 0.025.

Following your suggestion, we performed same calculation of the GW-induces shears under the mid-frequency assumption. The results are shown in below and are added in the discussion part of the text.

To fully explore the differences of magnitudes of $S$ derived under the assumptions of $f/\widehat{\omega}_j \neq 0$ and $f/\widehat{\omega}_j \approx 0$, we show the GW-perturbed shears for $f/\widehat{\omega}_j \approx 0$ in a same manner as those in Fig. (4-9). Such that we can judge that whether the assumptions which need to be made. In a same manner as Fig. 4, we present in Fig. R2 the latitude-height contours of the zonal means (the first row) and standard deviations (std, the second row) of GW-perturbed shears and the top 10% largest shears (the third row) under the medium frequency assumption $f/\widehat{\omega}_j = 0$. The stds and the top 10% largest shears are not shown here since they have similar patterns as the zonal mean of GW-perturbed shears but have smaller maxima than those shown in Fig. 4. From Fig. R2 we can see that the latitude-height patterns of the GW-perturbed shears are the same as those shown in Fig. 4. However, the maxima of the zonal mean GW-perturbed shears decrease from 12-17 ms$^{-1}$km$^{-1}$ for $f/\widehat{\omega}_j \neq 0$ to 11-15 ms$^{-1}$km$^{-1}$ for $f/\widehat{\omega}_j \approx 0$. The maxima of the zonal mean GW-perturbed shears at latitudes higher than 50°N are at a higher height for $f/\widehat{\omega}_j \approx 0$ than that for $f/\widehat{\omega}_j \neq 0$. In a same manner as Fig. 5, we show in Fig. R3 the profiles of $S$ as well as the top 10% and 1% largest $S$ derived under the assumption of $f/\widehat{\omega}_j = 0$ during January and July at around 40°N. From Fig. R3 we can see that the height variations of these profiles are similar to those shown in Fig. 5. However, the magnitudes of these profiles are slightly smaller than those shown in Fig. 5.

In a same manner as Fig. (6-7), we show in Fig. (R4-R5) the zonal means (the first row) and standard deviations (std, the second row) of GW-perturbed shears and the top 10% largest shears (the third row) , respectively, during the four composite seasons and six yaw cycles derived under the medium frequency assumption of $f/\widehat{\omega}_j = 0$. From Fig. (R4-R5) we can see these results have similar patterns as those for $f/\widehat{\omega}_j \neq 0$ but have slightly smaller magnitudes. Moreover, the maxima of these results at high latitudes of summer hemispheres are at a higher height for $f/\widehat{\omega}_j \approx 0$ than those for $f/\widehat{\omega}_j \neq 0$. The peaks circled by blue rectangles are very weak and almost disappear in Fig. (R4-R5) as compared to those shown in Fig. (6-7). The stds and the top 10% largest shears are not shown here since they have similar patterns as the zonal mean of GW-perturbed shears but have smaller maxima than those shown in Fig. (6-7).

In a same manner as Fig. (8-9), we show in Fig. (R6-R7), respectively, the latitude-height contours and time-height contours of the zonal means of the GW-perturbed shears derived under the medium frequency assumption of $f/\widehat{\omega}_j = 0$. Comparing between Fig. (8-9) and Fig. (R6-R7), we can see that the GW-perturbed shears derived under the medium frequency assumption of $f/\widehat{\omega}_j = 0$ have similar patterns as those for $f/\widehat{\omega}_j \neq 0$ but have slightly smaller magnitudes.

In summary, the GW-perturbed shears derived under the assumptions of $f/\widehat{\omega}_j \neq 0$ and $f/\widehat{\omega}_j = 0$ have similar patterns on the aspects of latitude-height, time-height contours. The

magnitudes of the GW-perturbed shears derived under the assumption of $f/\hat{\omega}_j = 0$ are slightly smaller than those under the assumption of $f/\hat{\omega}_j \neq 0$

[Figure]

Figure R2: Same caption as Fig. 4 but for the zonal mean of GW-perturbed shears derived under the medium frequency assumption $f/\hat{\omega}_j = 0$.

[Figure]

Figure R3: Same caption as Fig. 5 but for the GW-perturbed profiles derived under the medium frequency assumption $f/\hat{\omega}_j = 0$ and at 40°N

[Figure]

Figure R4: Same caption as Fig. 6 but for the zonal means of GW-perturbed shears derived under the medium frequency assumption $f/\hat{\omega}_j = 0$.

[Figure]

Figure R5: Same caption as Fig. 7 but for the zonal means of GW-perturbed shears derived under the medium frequency assumption $f/\widehat{\omega}_j = 0$.

[Figure]

Figure R6: Same caption as Fig. 9 but the GW-perturbed shears derived under the medium frequency assumption $f/\widehat{\omega}_j = 0$.

[Figure]

Figure R7: Same captions as Fig.9 but for the GW-perturbed shears derived under the medium frequency assumption $f/\hat{\omega}_j = 0$.

2. Rewrite the theory part and properly name the assumptions (see also detailed comments below). How does your picture change, if you assume that the different waves you superpose have different propagation directions? The following more in the line of suggestions, which may be beyond the scope of the paper: It would be nice to have a Monte Carlo simulation showing the effect of superposing waves of different omega/f and different propagation direction, but I admit that the result would again depend on some assumptions about the real distributions. You could also try to use simulated observations through the model (just sampling, no radiative transfer) to get some idea on the size of effects. Maybe it also would be worthwhile to assume first that one could infer propagation direction as well, as in principle such instruments could be build.

**Response:** Thanks for your good suggestion. According to your suggestion, we have clarified the assumptions in the theory part (Sec. 2). The main revisions are:

(1) The assumptions on GW propagation directions: These monochromatic GWs may not propagate in the same direction. However, from two adjacent GW profiles, one cannot get the actual horizontal wavenumber ($k_{j,hr}$) of each monochromatic, but can get the projection of actual wave

vector in the along-track direction ($k_{j,ha}$). The inconsistency between $k_{j,hr}$ and $k_{j,ha}$ introduces uncertainties of GW-perturbed shears, which are discussed in Sec. 2.3. Thus, $u'$ can be expressed as the vector sum of the projections of all the actual monochromatic GWs in the along-track direction and formulated as,

$$u' = \sum_j u_j' = \sum_j \tilde{u}_j e^{i\varphi_j} = \frac{g}{N}\sum_j \left[\left(1 - f^2/\hat{\omega}_j^2\right)^{-1/2} \tilde{T}_j e^{i(\varphi_j + \pi/2)}\right].$$

(2) Due to the uncertainties in determining the GWs' propagation direction, only the magnitudes of GW-perturbed shears are analyzed in this work.

(3) It should be noted that the $k_{j,h}$ is the projection of the actual GW's horizontal wavenumber in the along-track direction and is underestimated. This underestimates the intrinsic frequency and thus overestimates the wind and shears based on Eq. (6-9) and will be discussed detail in Sec. 2.3.

(4) However, GWs may not propagate only in the along-track direction. Our assumption will introduce some uncertainties and will be discussed below.

It is a good suggestion to perform the Monte Carlo simulation to show the effect of superposing waves of different omega/f and different propagation direction. We plan to perform the simulation in our future work.

3.  Discuss the influence of noise. In particular it is known that SABER temperature retrievals have a noise problem at the summer mesopause. That is stated on the SABER web-page and also discussed for GW analyses by Ern et al. 2018. The results for this particular region have to be treated with care. At least some discussion must be included in the paper.

    **Response:** We have added some discussion on this point in Sec. 3.1 as below:

The GW-perturbed shears might be influenced by temperature uncertainties of SABER measurements, which are much larger at around the cold summer mesopause (Remsberg et al., 2008). We removed waves with vertical wavelengths shorter than 5 km, to minimize the noise introduced by uncertainties of SABER measurements (Ern et al., 2011, 2018).

4.  The paper must be self-explaining. Explain how you arrive at your vertical wavelengths and amplitudes. Anyway: You should not use FFT over the whole altitude regime. There is so much Doppler shift, critical level filtering and oblique propagation going on that the case where you observe the same gravity wave at all altitudes and furthermore with similar vertical wavelength is the absolute exception.

    **Response:** You are right. For a single GW, its amplitudes and wavelengths may vary when they propagate due to the Doppler shift, critical level filtering and oblique propagation, etc. A GW profile

extracted from observations is a superposition of many GWs, which may come from different locations and have height dependent amplitudes and wavelengths. We have replaced the DFT with discrete wavelet transform (DWT, Torrence & Compo, 1998) in this version. Such that we can get the height dependent vertical wavelengths and amplitudes. Here we show in Figure 1(DWT) the GW-perturbed shears derived through DWT. For comparison, we show in Figure 1(DFT) the GW-perturbed shears derived through DFT. Comparing between Figure 1(DWT) and Figure 1(DFT), similar GW-perturbed shears can be obtained from both methods. However, the slightly difference occurs at z=85-95 km, where the small-scale perturbations of GW-perturbed shears are clear for the DFT method but not for the DWT method. On the statistical sense, the GW-perturbed shears exhibit similar magnitudes and latitude-height and time-height patterns (see text for detailed results).

We have revised the method in the Sec.2.2.

Torrence, C. and Compo, G. P.: A Practical Guide to Wavelet Analysis, Bull. Am. Meteorol. Soc., 79(1), 61–78, doi:10.1175/1520-0477(1998)079<0061:APGTWA>2.0.CO;2, 1998.

[Figure]

**Figure 1(DWT):** Synthetic temperature perturbation profiles (a, $T'_{w1}(z)$ and $T'_{w2}(z)$ are represented by black and red lines, respectively) and the corresponding winds (b, zonal; c, meridional) and shears (d, zonal; e, meridional). The winds and shears are, respectively, calculated from the prescribed amplitudes and wavenumbers (black line, labelled as "th") and reconstructed by spectral decomposition method (red dashed line, labelled as "rc"). All panels have the same y-axis scale.

[Figure]

**Figure 1(DFT):** Same caption as Figure 1(DWT) but for the results derived from DFT.

5.   In particular: how do you deal with tides? All tides must be removed before the analysis as they are global wave modes with wind amplitudes maximizing at a different latitude than temperature amplitudes.

**Response:** According to Preusse et al. (2002) and Ern et al. (2004, 2018), due to the slowing precessing of the SABER measurement (a full cycle is about 120 days), migrating tides will appear as stationary zonal wave patterns if data from ascending and descending nodes are taken separately. By removing these stationary wave patterns separately for ascending and descending nodes, tides can be removed from the observed temperature fluctuations.

In the first paragraph of Sec. 2.3, we present the method of extraction methods of GW profiles: "First, the daily SABER temperature profiles $T(z)$ in a latitude band of 5° are selected. Second, at each altitude, these selected data are fitted by harmonics with zonal wavenumbers ranging from 0 to 6, which are mainly planetary waves and non-migrating tides and are removed from $T(z)$ to get the residual temperature $T_r(z)$. The component of wavenumber 0 is considered as the zonal mean temperature $\bar{T}(z)$. Due to the slowing precessing of the SABER measurement (a full cycle is about 120 days), migrating tides will appear as stationary zonal wave patterns if data from ascending and descending nodes are taken separately. The above two steps are applied on both the ascending and descending nodes, respectively, such that it minimizes the influences of migrating tides on the residual temperature $T_r(z)$ and thus GW profiles (Preusse et al., 2002, Ern et al., 2004, 2018)".

6.   Discussions: for GWMF I think it is a valid assumption that wind modulation causes most of the relevant waves to propagate opposite to the winds. You can find some of this expressed in AIRS

results and also GW resolving global models. However, for shear I don't think that is further valid. Therefore, I would include some of the discussion (very much shortened) in sec. 2.

**Response:** For the wave propagation directions, we have added some discussions in Sec. 2 as below:

These monochromatic GWs may not propagate in the same direction. However, from two adjacent GW profiles, one cannot get the actual horizontal wavenumber ($k_{j,hr}$) of each monochromatic, but can get the projection of actual wave vector in the along-track direction ($k_{j,ha}$). The inconsistency between $k_{j,hr}$ and $k_{j,ha}$ introduces uncertainties of GW-perturbed shears, which are discussed in Sec. 2.3. Thus, $u'$ can be expressed as the vector sum of the projections of all the actual monochromatic GWs in the along-track direction and formulated as,

$$u' = \sum_j u'_j = \sum_j \tilde{u}_j e^{i\varphi_j} = \frac{g}{N} \sum_j \left[ \left(1 - f^2/\hat{\omega}_j^2\right)^{-1/2} \tilde{T}_j e^{i(\varphi_j + \pi/2)} \right].$$

**Specific comments**

1. P2L4 It is now accepted ... A larger part of the shear and the winds is associated with tides. You need to mention them here. In particular sporadic E is primarily associated with tides (e.g. Arras et al., 2009, Arras et al. 2018).

   **Response:** We have revised "The large winds and shears play important…" as "The large winds and shears are associated with tides and GWs and play important…". Moreover, we have added the references of Arras et al. (2009), Arras and Wickert (2018), and Jacobi et al. (2019) to show the relations between sporadic E and tides.

   Arras, C., Jacobi, C., and Wickert, J.: Semidiurnal tidal signature in sporadic E occurrence rates derived from GPS radio occultation measurements at higher midlatitudes, Ann. Geophys., 27, 2555–2563, https://doi.org/10.5194/angeo-27-2555-2009, 2009.

   Arras, C. and Wickert, J.: Estimation of ionospheric sporadic E intensities from GPS radio occultation measurements, J. Atmos. Sol.-Terr. Phys., 171, 60–63, https://doi.org/10.1016/j.jastp.2017.08.006, 2018.

   Jacobi, C., Arras, C., Geißler, C., and Lilienthal, F.: Quarterdiurnal signature in sporadic E occurrence rates and comparison with neutral wind shear, Ann. Geophys., 37, 273–288, https://doi.org/10.5194/angeo-37-273-2019, 2019.

2. P2L9 Please explain this equation: what is the motivation? Static stability of saturated mid frequency GW?

**Response:** "Based on the theory of dynamic instability, Liu (2007) showed that the maximum wind shears derived from $S = 2N$ ($N$: the buoyancy frequency, $S$: the vertical shear) coincide well with the observed large wind shears" have been revised as:

Based the definition of the Richardson number $Ri = N^2/S^2$, the atmosphere is dynamic stable when $Ri > 1/4$ and dynamic instable when $Ri \leq 1/4$. Here $N^2 = g/\bar{T}\left(d\bar{T}/dz + g/c_p\right)$ is the static stability, $S^2 = (\partial u/dz)^2 + (\partial v/dz)^2$ is the wind shear; $\bar{T}$ is the background temperature; $g$ and $c_p$ are the gravitational acceleration and specific heat for dry air at constant pressure, respectively; $u$ and $v$ are the zonal and meridional winds, respectively. The threshold of dynamic instability is $Ri = 1/4$, which means that the maximum wind shear allowed by the background static stability should be $S = 2N$. According to $S = 2N$, Liu (2007) showed that the maximum wind shears coincide well with the observed large wind shears.

3. P2L34 "poorly known" Motivate. What are the limitations of the wind observations? Noise? Altitude resolution? Combination to vector winds?

   **Response:** We have clarified this point as "However, the global characteristics of shears are poorly known due to either the limited altitude coverage or the altitude resolution or large noises of the satellite observations".

4. P3 Equ 1: Under the assumption of conservative propagation and no refraction. Reasonable for some waves in a limited altitude range, but must not be taken over the whole MA as is done below.

   **Response:** We have added the assumption as "For a conservative propagation and no refraction monochromatic linear GW, it has the form of…".

5. P3L20/L25 These polarization relations are for winds u parallel and v perpendicular to the wavevector of the GW. You can do the assumption, but must say so in the text.

   **Response:** We have revised this as "$u'_j$ and $v'_j$ are the horizontal wind perturbations parallel and perpendicular to the wavevector of the GW, respectively".

6. P4 Equ. 6 and Equ. 7 These equations make the much stronger assumption that the wave vectors are all pointing in the same direction. While that seems a plausible assumption, if you look for the largest amplitudes (preferential orientation against the mean wind), I don't see why this should be the case for wind shear. You showed in Equs 4 and 5 that shear is proportional to m as well as to the amplitude. Handwaving, these two effects should largely cancel with regards to phase speed. Thus, you would not expect a direction preference.

**Response:** You are right. The wave vectors are not pointing in the same direction and determined by the horizontal wavenumber $k_{j,h}$. However, we cannot get the actual wave vector using a GWs' pair but get the projection of actual wave vector in the along-track direction. Thus, Eq. (6-7) are the vector sum of the projection of all the actual monochromatic GWs in the along-track direction. The inconsistency between the actual wave vector and the projected wave vector induces the uncertainties of shears as discussed in Sec. 2.3

According to your comment, we clarified this point before Eq. (6) as below:

These monochromatic GWs may not propagate in the same direction. However, from two adjacent GW profiles, one cannot get the actual horizontal wavenumber ($k_{j,hr}$) of each monochromatic GW, but can get the projection of actual wave vector in the along-track direction ($k_{j,ha}$). The inconsistency between $k_{j,hr}$ and $k_{j,ha}$ introduces uncertainties of GW-perturbed shears, which are discussed in Sec. 2.3. Thus, $u'$ can be expressed as the vector sum of the projections of all the actual monochromatic GWs in the along-track direction and formulated as,

$$u' = \sum_j u'_j = \sum_j \tilde{u}_j e^{i\varphi_j} = \frac{g}{N}\sum_j \left[\left(1 - f^2/\hat{\omega}_j^2\right)^{-1/2}\tilde{T}_j e^{i(\varphi_j + \pi/2)}\right]. \tag{6}$$

7.  P4 Equ. 8 and 9 then are not valid

**Response:** Eq. (8-9) are the vertical gradient of $u'$ and $v'$. Due to the linearity of vector sum operation, the partial derivation the vector sum is equivalent to the vector sum of all the components. To clarify this point, we revised Eq. (8-9) as:

$$\frac{\partial u'}{\partial z} = \frac{\partial}{\partial z}\left(\sum_j u'_j\right) = \sum_j \frac{\partial u'_j}{\partial z} = \frac{g}{N}\sum_j \left[m_j\left(1 - f^2/\hat{\omega}_j^2\right)^{-1/2}\tilde{T}_j e^{i(\varphi_j + \pi)}\right], \tag{8}$$

$$\frac{\partial v'}{\partial z} = \frac{\partial}{\partial z}\left(\sum_j v'_j\right) = \sum_j \frac{\partial v'_j}{\partial z} = \frac{g}{N}\sum_j \left[\frac{fm_j}{\hat{\omega}_j}\left(1 - f^2/\hat{\omega}_j^2\right)^{-1/2}\tilde{T}_j e^{i(\varphi_j + \pi/2)}\right]. \tag{9}$$

8.  P5 Equ 12 is fine, but is introduced in a way that makes belief that you could get the horizontal wavenumber from the satellite. Instead it is the one projected to the track. This will introduce overestimate of lambda_h, underestimate of k_h and thus underestimate of nhat{nomega}. You introduce some of these arguments below, but this must be more clearly ordered.

**Response:** We have added a note to clarify this point as "It should be noted that the $k_{j,h}$ is the projection of the actual GW's horizontal wavenumber in the along-track direction and is underestimated. This underestimates the intrinsic frequency and thus overestimates the wind and shears based on Eq. (6-9) and will be discussed detail in Sec. 2.3."

9.  P5L8 No you can not! This is not the zonal and meridional component!

**Response:** We have revised this as "we can get $u'_j$ and $v'_j$ and their shears".

10. P5L15 And why should the wavevector point along the orbital track? That is as unlikely.

**Response:** This point has been clarified as "Here we assume that $u'_j$ and $v'_j$ are the winds along and cross the orbit track directions, respectively". Moreover, we have added "However, GWs may not propagate only in the along-track direction. The assumption will introduce uncertainties and will be discussed below" in the end of this paragraph.

11. P5L25 There is no reason to believe that the waves at 20 km have anything to do with the waves at 100 km. Please explain your method. Phase differences should be gained from "local" phases determined for a limited altitude range.

**Response:** You are right. We used discrete wavelet transform to decompose GWs and reconstruct GW-perturbed shears. Such that we can get the height dependent amplitudes and vertical wavelengths. The local phase shift is computed by the cospectrum of two adjacent GW profiles (Ern et al., 2004, Alexander, 2008). In the text we have revised as:

The first step is to evaluate the amplitude and vertical wavenumbers of each GW profile by the method of discrete wavelet transform (short for DWT, Torrence & Compo, 1998) such that we can get the height dependent amplitudes and vertical wavelengths. For the GW profiles of $T'_{w1}(z)$ and $T'_{w2}(z)$, then at each height and vertical wavelength ($\lambda_z = 2\pi/m$), their DWT are $\hat{T}_{w1}(z, \lambda_z)$ and $\hat{T}_{w2}(z, \lambda_z)$. Then, their cospectrum $C_{1,2}$ is computed,

$$C_{1,2} = \hat{T}_{w1}\hat{T}^*_{w2} = \tilde{T}_{w2}\tilde{T}_{w2} \exp(i\Delta\varphi_{1,2}),  \tag{13}$$

Here, $\tilde{T}_{w1} = |\hat{T}_{w1}|$ and $\tilde{T}_{w2} = |\hat{T}_{w2}|$ are the amplitudes of $T'_{w1}(z)$ and $T'_{w2}(z)$, $\hat{T}^*_{w2}$ is the complex conjugation of $\hat{T}_{w2}$. The phase shift is calculated by $\Delta\varphi_{1,2} = \tan^{-1}[\text{Im}(C_{1,2})/\text{Re}(C_{1,2})]$. When performing DWT, we restrict the vertical wavelength ranging from ~5 km to ~30 km for a vertical extent of 90 km, which is the height coverage (18-108 km) of the SABER temperature profiles.

It should be noted that, the GW-perturbed shears derived from DWT and DFT are very similar. Please see the response to the **Major commets#4**. Thus, changing the method from DFT to DWT does not change our conclusions.

12. P8L6 can you please make these number codes plain text 01 July - 31 July which Year?

**Response:** We have revised this as "(e.g., 31 days during 01 July-31 July in 2018 and 62 days during 29 June-30 August in 2018). The same is true during the intervals of 06 May-27 June and 29 June-30 August in 2018, respectively, at latitudes of 52.5°S-52.5°N. The GW-perturbed shears during the continuous two yaw cycles (06 May-27 June and 29 June-30 August)". We also revised the number codes as plain text at other places in the text.

13. P8L15 This is a region problematic for retrievals due to very cold temperatures and low IR signals (cf. Remsberg, Ern). Some of what you see there may be real, but some is very likely due to increased noise. A discussion is needed.

    **Response:** We have added a discussion on this point in the end of this paragraph as "The GW-perturbed shears might be influenced by temperature uncertainties of SABER measurements, which are much larger at around the cold summer mesopause (Remsberg et al., 2008). We removed waves with vertical wavelengths shorter than 5 km, to minimize the noise induced by uncertainties of SABER measurements (Ern et al., 2011, 2018)".

14. P9L16 You mean the zonal mean of the individual shear profiles? Please be precise, as the zonal mean shear due to GWs should be zero. But then: what does the lidar provide? Average of abs.(shears) or shear of average?

    **Response:** This is my mistake and should be clarified. Both the means shears and the zonal mean shears are the means of the shear's magnitude (i.e., mean of abs.(shears), no direction) throughout the manuscript. We have added a note on Equation (10) as:

    Finally, the magnitude of GW-perturbed shear can be written as

    $$S = \sqrt{(\partial u'/\partial z)^2 + (\partial v'/\partial z)^2}. \tag{10}$$

    Due to the uncertainties in determining the GWs' propagation direction, only the magnitudes of GW-perturbed shears are analyzed in this work."

15. P9L25 You do not discuss here the visibility filter, which also should reduce amplitudes

    **Response:** We have added this point in the text as "(3) the observational filter should underestimate the amplitudes of GWs, especially for GWs with shorter vertical wavelengths (Ern et al., 2018), and thus underestimate the GW-perturbed shears".

16. P12L17 And also on what you emphasize due to the scaling with m: shear (short vertical wavelength), amplitudes (mid lz) or GWMF (long lz).

    **Response:** We have added the point of "influences of the vertical wavenumber m on the differences of the heights at which phase shifts occur" in the text and revised as:

    "However, the heights at which phase shifts occur are different. One reason is that the background temperature and static stabilities (Liu et al., 2020). The other reason is that the GWMF is inverse proportion to the vertical wavenumber $m$, while the GW-perturbed shear is proportion to the $m$. The resulting height of the phase shift of GWMF at a lower height than that of GW-perturbed shears since $m$ increases with height below z=90 km (Ern et al. (2018)."

17. P12L21 also Schmidt et al.

    **Response:** We have added "Schmidt et al. (2016)" here.

Schmidt, T., Alexander, P., and de la Torre, A.: Stratospheric gravity wave momentum flux from radio occultations, J. Geophys. Res.-Atmos., 121, 4443–4467, https://doi.org/10.1002/2015 JD024135, 2016.

18. P13L11 SABER "sees" GWs with horizontal wavelengths > 200km, much the same as WACCM, though with somewhat reduced amplitude. I would check by mid-frequency approach, whether the scaling and profile distance is really of importance. In addition, as you said above, if there is an influence, it would be an overestimation.

    **Response:** Below are Equation (8) and (12),

$$\frac{\partial u'}{\partial z} = \sum_j \frac{\partial u_j'}{\partial z} = \frac{g}{N}\sum_j \left[ m_j\left(1 - f^2/\widehat{\omega}_j^2\right)^{-1/2} \tilde{T}_j e^{i(\varphi_j + \pi)} \right], \tag{8}$$

$$\widehat{\omega}_j^2 = N^2 \frac{k_{j,h}^2}{m_j^2} + f^2. \tag{12}$$

According to Equation (8), we can see that the longer horizontal wavelength (i.e., smaller horizontal wavenumber) induces smaller intrinsic frequency $\widehat{\omega}_j$. This increases the magnitude of $\partial u'/\partial z$ and overestimates the GW-perturbed shears based on Equation (8).

However, what we want to express is that the longer sampling distances miss the GWs with shorter horizonal wavelengths, which might also contribute the GW-perturbed shears.

We did not express this point clearly in the text and revised it in the new version as "The longer sampling distances miss the GWs with shorter horizonal wavelengths, which might also contribute the GW-perturbed shears and thus reduces the magnitudes of GW-perturbed shears".

**Technical corrections**

1. P1L28 ... (or more precisely vertical wind shears) ...

   **Response:** This has been revised as "…large horizontal winds and their vertical shears (or more precisely vertical wind shears, short for shears)…".

2. P1L30 ... and from ground-based ...

   **Response:** We have added "from" before "ground-based".

3. P2L2 ... strong gravity wave (GW) activity.

   **Response:** We have revised "gravity waves (GWs)" as "gravity wave (GW)".

4. P2L2 You probably want to refer to a generally knowledge on GWs rather than on evidence from the aforementioned papers for the sources: Most prominent sources of GWs are convection, orography and jets and fronts in the troposphere as well as spontaneous adjustment and secondary wave generation in the stratosphere. Amplitudes and shears increase when the GWs propagate ...

   **Response:** Thanks. We have revised "These GWs had increasing wave amplitude and shears when they propagated into the lower thermosphere" as "Most prominent sources of GWs are convection, orography and jets and fronts in the troposphere as well as spontaneous adjustment and secondary wave generation in the stratosphere (Fritts and Alexander, 2003). Amplitudes and shears increase when the GWs propagate into the lower thermosphere".

5. P2L13 driving -> are the cause of

   **Response:** We have revised "important in driving" as "are the cause of".

6. P2L17 reproduced the large wind shears seen in the observations.

   **Response:** We have revised "reproduced the large wind shears" as "reproduced the large wind shears seen in the observations".

7. P2L22 Observations ... were made by sounding ... for a very limited number of locations only and hence cannot ...

   **Response:** We have revised "Large winds and shears have been observed both by sounding rocket and ground based lidar and radar, all locally and cannot provide a global morphology" as "Observations on large winds and shears were made by sounding rocket and ground-based lidar and radar for a very limited number of location only and hence cannot provide a global morphology".

8. P2L24 still challenging

   **Response:** We have revised "still a challenging" as "still challenging".

9. P3L4 of deriving shears from GW analyses of temperature observations.

   **Response:** We have revised "…. of deriving shears from GWs, while the GWs are derived from temperature profiles. The temperature profiles, which are measured by the Sounding…" as "…. of deriving shears from GW analyses of temperature observations…".

10. P3L6 SABER luckily still measures: Temperature profiles measured by .... from 2002 to 2019 are used for this study providing an 18 year period.

**Response:** We have revised "The temperature profiles, which are measured by the Sounding of the Atmosphere using Broadband Emission Radiometry (SABER) instrument (Russell et al., 1999) onboard the TIMED satellite, have covered a period of 18 years (2002-2019) and are remarkably stable until now (Mlynczak et al., 2020)" as "Temperature profiles measured by the Sounding of the Atmosphere using Broadband Emission Radiometry (SABER) instrument (Russell et al., 1999) onboard the TIMED satellite from 2002 to 2019 are used for this study providing an 18-year period". Temperature profiles are remarkably stable until now (Mlynczak et al., 2020)."

11. P3L8 ... website based on Remsberg et al. (2008).

    **Response:** We have revised "...website (Remsburg et al., 2008)" as "...website based on Remsberg et al. (2008)".

12. P3L10 omit () at this point

    **Response:** We have deleted "(short for GW-induced shears)".

13. P4L3 In reality, usually spectrum of GWs is observed formed by superposition of several monochromatic GWs.

    **Response:** We have revised "In real atmosphere, a GW profile consists multiple spectra and can be regarded as a superposition of monochromatic GWs" as "In reality, usually spectrum of GWs is observed formed by superposition of several monochromatic GWs".

14. P5L7 and the amplitude and vertical wavenumber determined from the satellite observations for the individual monoc. GWs, we ...

    **Response:** We have revised "and the prescribed amplitude and vertical wavenumber of each monochromatic GW, we can get the zonal and meridional winds components and their shears" as "and the prescribed amplitude and vertical wavenumber, which will be determined from the satellite observation and described below, for the individual monochromatic GW, we can get $u'_j$ and $v'_j$ and their shears".

15. P6L26 https://en.wikipedia.org/wiki/Flowchart Please use different wording here
    **Response:** We replaced "flow chart" by "procedure" here and figure caption.

16. P7L20 a factor 0.2 would be underestimation overestimated by 20%?
    **Response:** We have revised "which are overestimated by a factor of 0.2" as "which are overestimated by 20%".

17. P8L9 comment -> common

    **Response:** We have revised "comment" as "common".

18. P9L15 slightly larger

    **Response:** We have revised "slight larger" as "slightly larger".

**Responses to Referee#2:**

SABER temperature profiles are used as input to a calculation of gravity wave parameters, including the vertical and horizontal wavenumbers, the intrinsic frequency, and the horizontal velocity perturbations associated with each wave component in the decomposed wave spectrum. The goal is to estimate the vertical shears associated with the wave spectrum over the extended time periods and greater geographical extent provided by the satellite data.

The presentation of the material and methodology is clear, the discussion of past work is useful and very complete, and the figures represent the results well. Nonetheless, there are several major questions related to the validity of the analysis.

The shears obtained from the SABER data are calculated and are not directly measured. A broader global climatology, including the seasonal variations, is valuable to the community and represents a worthwhile goal. There are a number of critical assumptions required, however, for the calculations to be valid and meaningful. Those assumptions are especially likely to break down in the altitude range around the mesopause, which is where the largest shears occur that are a focus of the analysis. The assumptions are not adequately addressed in the paper, and the effect of those assumptions are therefore also not addressed. The recommendation is to return the manuscript to the authors for revisions.

**Response:** Thanks for your valuable comments on our manuscript. Following your comments below, we have addressed the assumptions and used wavelet transform to reconstruct the wind shears in this version. Please see below for detailed responses.

**Specific comments are as follows**

1. Tides are in some sense global-scale gravity waves constrained by spherical boundary conditions, but they clearly do not conform to the simple linearized polarization relations given in equations (2) and (3). The authors introduce some filtering to reduce or eliminate larger-scale modes, but it is not clear that the tidal modes, especially the higher order modes, are eliminated prior to extracting the gravity wave spectral components. This is especially important since the altitude range where the large shears occur near the mesopause is also an altitude range where a number of tidal modes have large amplitudes, perhaps not coincidentally?

   **Response:** According to Preusse et al. (2002) and Ern et al. (2004, 2018), due to the slowing precessing of the SABER measurement (a full cycle is about 120 days), migrating tides will appear as stationary zonal wave patterns if data from ascending and descending nodes are taken separately.

By removing these stationary wave patterns separately for ascending and descending nodes, tides can be removed from the observed temperature fluctuations.

In the first paragraph of Sec. 2.3, we present the method of extraction methods of GW profiles: "First, the daily SABER temperature profiles $T(z)$ in a latitude band of 5° are selected. Second, at each altitude, these selected data are fitted by harmonics with zonal wavenumbers ranging from 0 to 6, which are mainly planetary waves and non-migrating tides and are removed from $T(z)$ to get the residual temperature $T_r(z)$. The component of wavenumber 0 is considered as the zonal mean temperature $\bar{T}(z)$. Due to the slowing precessing of the SABER measurement (a full cycle is about 120 days), migrating tides will appear as stationary zonal wave patterns if data from ascending and descending nodes are taken separately. The above two steps are applied on both the ascending and descending nodes, respectively, such that it minimizes the influences of migrating tides on the residual temperature $T_r(z)$ and thus GW profiles (Preusse et al., 2002, Ern et al., 2004, 2018)".

2.  A further concern is that the region of large shears near the mesopause is also a region of large winds that represents a height range with critical levels for a large fraction of the gravity wave spectrum, especially since the background winds rotate with height through that altitude range, so there are critical levels for a broad range of propagation directions. It appears questionable to apply the linearized polarization relations in regions with critical levels where the vertical wavelength is changing rapidly with height and the dynamics are almost certainly nonlinear. The validity of the analysis in those regions is not discussed.

**Response:** Thanks for your good suggestion. According to your suggestion, we have added a paragraph in the end of Sec. 2.3 as below:

The large winds and shears in the MLT region, where both tides and GWs reach large amplitudes and may interact nonlinearly (Fritts and Vincent, 1987; Li et al., 2009; Liu et al., 2014a) and break (Fritts et al., 2004; Liu and Vadas, 2013; Vadas and Liu, 2013). The large winds may rotate with height and act as critical levels, which filter out a broad spectral range of GWs. These filtered GWs may break and deposit their momenta in the background atmosphere, which create body force to general secondary GWs. Moreover, the vertical wavelengths of these filtered GWs change rapidly with height and in the nonlinear regime. These nonlinear GWs may also produce large winds and shears around the mesopause regions. However, these nonlinear GWs cannot be described by linear GW theory proposed here. Thus, the GW-perturbed shears underestimate the actual shears in the MLT region due to the unrepresented nonlinear GWs.

3.  The large shear region is also a region where many gravity waves break, producing turbulence and, in some cases, secondary waves, as discussed in at least one of the references cited in the

manuscript. The critical levels mentioned in the previous comment can produce breaking, but breaking waves can also be a result of the amplitude growth with height, which by itself leads to amplitudes large enough to produce such effects at heights near the mesopause.

**Response:** We added the discussion on the limitation of our method, which is only valid for the linear GW. For the GW break and turbulence, as well as the secondary waves, they are in the nonlinear regime and out of the scope of the linear GW theory. This point has been addressed in the response to comment#2 and added a paragraph in the end of Sec. 2.3.

4. If we assume a localized source in the lower atmosphere that produces a spectrum of waves, then the lowest-frequency gravity waves propagate at a lower elevation angle while the higher-frequency waves propagate at angles closer to vertical. A vertical profile extending from the ground to the lower thermosphere is therefore unlikely to have the contents from a single set of monochromatic waves contributing at all heights. As an example, several studies have been published in the past with special cases in which waves have been traced from the troposphere to the thermosphere when there has been an especially strong source, such as a strong line of thunderstorms, but those waves become displaced horizontally as they propagate vertically so that a vertical sample would only intercept the wave packet in a small part of the complete altitude range. In the general case the altitude profile measured by the satellite will likely have contributions from an extended range of geographically-distributed sources and therefore also different phasing dominating the vertical wavelength contribution in each part of the altitude range. If so, it is not clear that an individual wavelength contribution extracted via a DFT represents the wave structure accurately and applying the polarization relations as if that particular component represents a single monochromatic wave seems questionable.

**Response:** You are right. For a single GW, its amplitudes and wavelengths may vary when they propagate due to the Doppler shift, critical level filtering and oblique propagation, etc. A GW profile extracted from observations is a superposition of many GWs, which may come from different locations and have height dependent amplitudes and wavelengths. We have replaced the DFT with discrete wavelet transform (DWT, Torrence & Compo, 1998) in this version. Such that we can get the height dependent vertical wavelengths and amplitudes. Here we show in Figure 1(DWT) the GW-perturbed shears derived through DWT. For comparison, we show in Figure 1(DFT) the GW-perturbed shears derived through DFT. Comparing between Figure 1(DWT) and Figure 1(DFT), similar GW-perturbed shears can be obtained from both methods. However, the slightly difference occurs at z=85-95 km, where the small-scale perturbations of GW-perturbed shears are clear for the DFT method but not for the DWT method. On the statistical sense, the GW-perturbed shears exhibit similar magnitudes and latitude-height and time-height patterns (see text for detailed results).

We have revised the method in the Sec.2.2.

Torrence, C. and Compo, G. P.: A Practical Guide to Wavelet Analysis, Bull. Am. Meteorol. Soc., 79(1), 61–78, doi:10.1175/1520-0477(1998)079<0061:APGTWA>2.0.CO;2, 1998.

[Figure]

**Figure 1(DWT):** Synthetic temperature perturbation profiles (a, $T'_{w1}(z)$ and $T'_{w2}(z)$ are represented by black and red lines, respectively) and the corresponding winds (b, zonal; c, meridional) and shears (d, zonal; e, meridional). The winds and shears are, respectively, calculated from the prescribed amplitudes and wavenumbers (black line, labelled as "th") and reconstructed by spectral decomposition method (red dashed line, labelled as "rc"). All panels have the same y-axis scale.

[Figure]

**Figure 1(DFT):** Same caption as Figure 1(DWT) but for the results derived from DFT.

5. Finally, since only the along-track direction is sampled, the horizontal wavelength will be overestimated for any waves propagating in an off-track direction. The resulting underestimate of the horizontal wavenumber artificially increases the value of omega prime (equation 12) and leads to an overestimate of the vertical shear (equations 8 and 9). Given that, it seems possible that the apparent seasonal variations and latitudinal variations in the derived shears are a reflection of seasonal and latitudinal changes in preferred wave propagation directions rather than actual changes in the magnitudes of the shears.

**Response:** Following your comment, we have added this point in the last paragraph of Sec. 4.2 as "The GW-perturbed shears, which is derived from the projection of actual GWs in the along-track direction, are overestimated as compared to the actual GW-perturbed shears. The extent of overestimation depends the actual GW propagations, which have seasonal and latitudinal preferences. Thus the seasonal and latitudinal variations of GW-perturbed shears may be influenced by the preferred GWs propagation directions to some extent".

6. A minor comment is that the title of the paper, which uses the terms "gravity wave induced shears", suggests an effect in which waves are accelerating the flow to produce the shears, i.e., inducing a shear by momentum deposition for example, but that is not what the manuscript describes since the analysis focuses only on the superposition of shears that are naturally a part of the wave spectrum without any mean-flow acceleration.

**Response:** You are right. We changed the "gravity wave induced shears" to "gravity wave perturbed shears" throughout the text.

[revised manuscript text omitted]

删除了: GW-induced

删除了: and

删除了: GW-induced

删除了: GW-induced

删除了: GW-induced

删除了: GW-induced

删除了: GW-induced

删除了: GW-induced

删除了: GW-induced

continuous two composite yaw cycles may have overlaps, with the longest overlap time of about 20 days. Thus, the composite yaw cycle can represent the results during two months around the center date of each composite yaw cycle.

From Fig. 6, we can see that the zonal means of the GW-perturbed shears (the first row) increase with the increasing height and latitude in general. The peaks are ~10-15 ms$^{-1}$km$^{-1}$ above 90 km at latitudes of 82.5°S-50°N (50°S-82.5°N) during spring and summer (autumn and winter). Moreover, the wind shears have peaks at a lower height ($z$~80-90 km) and at latitudes of 82.5°S-50°S (50°N-82.5°N) during autumn and winter (spring and summer). These lower height peaks during spring and autumn (highlighted by blue rectangles) are weaker than those during summer and winter (highlighted by red rectangles). Comparing with Fig. 7 (the second and third columns, 0228-0512, 0502-0715), we find that the weak peak at $z$~80-90 km during spring is contributed from the wind shears during May (highlighted by a red rectangle in Fig. 7 during the yaw cycle of 0502-0715) since there is no peak at similar location during the yaw cycle of 0228-0512. The same is true during autumn, when the weak peak at z~80-90 km is contributed from the wind shears during November (highlighted by a red rectangle in Fig. 7 during the yaw cycle 1031-114) since there is no peak at similar location during the yaw cycle of 1228-0318. The stronger peaks during summer and winter in Fig. 6 are contributed from those during the yaw cycles of 0502-0715 and 1031-0114, respectively. The stronger peaks above 90 km and at ~$z$=80-90 km (marked by red rectangles) are both at around the mesopause as referred to the zonal mean temperature (contour lines in the second rows of Figs. 6 and 7).

The std and the top 10% largest shears, which are shown in the second and third rows of Figs. 6 and 7, respectively, have similar patterns as that of zonal mean shears. The maxima of the std and the top 10% largest shears are, respectively, ~12 ms$^{-1}$km$^{-1}$ and ~30 ms$^{-1}$km$^{-1}$, which are slightly less than that shown in Fig. 4. This is because the sampling profiles in Figs. 6 and 7 (composite season or yaw cycle over 18-year) are much larger than those in Fig. 4 (only one yaw cycle in one year).

Since the patterns of zonal mean and stds of the GW-perturbed shears and the top 10% largest shears are similar to each other (as shown in Figs. 4, 6, 7), only the zonal mean shears during each summer from 2002 to 2019 are shown in Fig. 8. It can be seen that the latitude-height distributions of GW-perturbed shears, including the peaks at lower heights (around the mesopause region) of high latitudes, are similar to the 18-year's mean results shown in Figs. 6 and 7. However, the GW-perturbed shear magnitudes (shown in Fig. 8) exhibit year-to-year variations. For example, at the SH high latitudes, the wind shears above 90 km are strongest during 2008 and 2019 and weakest during 2002. At the NH high latitudes, the GW-perturbed shears at ~$z$=85-95 km vary with year more greatly and have smaller values, as compared to those at around 80 km.

**4.2 Intra-annual Oscillations of GW-perturbed Shears**

Since the GW-perturbed shears are prominent around the mesopause region, their intra-annual oscillations will be studied at ~$z$=60-100 km. Figure 9 shows the monthly zonal mean GW-perturbed shears at four latitudes bands of the NH and SH from 2002 to 2019. A general feature of time-height variations GW-perturbed shears are the annual (AO) and semiannual oscillations (SAO). To quantify the exact amplitudes and phases of AO and SAO, harmonic fitting is applied on the GW-

删除了: GW-induced

删除了: GW-induced

删除了: GW-induced

删除了: GW-induced

删除了: GW-induced

删除了: GW-induced

删除了: GW-induced

删除了: GW-induced

删除了: GW-induced

删除了: GW-induced

[revised manuscript text omitted]

---

## Author Response (AR2)

**Responses to Referee#1:**

For final publication, the manuscript should be "accepted subject to technical corrections" Response: Thanks for your help in improving the level of manuscript.

**Suggestions for revision or reasons for rejection (will be published if the paper is accepted for final publication)**

The revised paper describes the methods and potential caveats very thoroughly. In particular, the added figures allow to estimate the influence of the most critical part, i.e. the influence of the frequency inferred from the horizontal wavelength estimate. The findings are original and relevant for the field. I hence recommend publication basically as is. A few technical corrections/suggestions are given below, but I also would like to suggest additional language editing by the Copernicus staff, who usually do a very good job on this.

Response: Thanks for your efforts in evaluating our manuscript.

**Corrections**

- P1L30 these are still a very few selected sites -> omit globally Response: We have deleted "globally".
- P2L3 ascribed -> attributed ; (Gs) -> (GWs)
   Response: We have revised "Gs" as "GWs".
- 3. P2L12 Based on ...

Response: We have replaced "Based" as "Based on".

4. P2L21 are are

Response: We have deleted one "are".

5. P3L11 noises -> noise

Response: We have revised "noises" as "noise".

6. P3L12 omit 'and' before 'in the MLT'; ... and since they are likely ...

**Response:** This sentence has been revised as "Since large shears play important roles in the atmospheric dynamics in the MLT region and ionospheric E region, and since they are likely caused by GWs, it is should be possible...".

P3L19 Retrieval of temperature has reached a very good quality ??? Long term stability of the SABER instrument is very high over the recent 18 years of measurements ???
 Response: Following your suggestion, we have added the sentence "This indicates the stability

of the SABER instrument is very high over the recent 18 years of measurements.".

 P3L30 For conservative propagation and without refraction, a linear monochromatic GW can be described as ()

**Response:** We have revised "For a conservative propagation and no refraction monochromatic linear GW, it has the form of ()" as "For conservative propagation and without refraction, a linear monochromatic GW can be described as".

9. P4L17 That's a jump: you first need to say how you actually want to use k and in particular the propagation direction

**Response:** Following your suggestion, we have added the sentence "Here we only consider the GWs propagating or projecting in the along-track direction" at the beginning of this paragraph.

- 10. P5L15 Omit: While
- 11. Response: We have deleted "While".
- 12. P6L17 (in short ...)**Response:** We have revised "short for DWT" as "in short DWT".
- 13. P9L5 is chosen to include data from both north-viewing and south-viewing yaw periods and hence to cover ...

**Response:** We have revised "The longer date coverage is chosen to cover a wider latitude range" as "The longer date coverage is chosen to include data from both north-viewing and south-viewing yaw periods and hence to cover a wider latitude range".

14. P9L6 for calculating zonal means

Response: We have revised "for zonal mean" as "for calculating zonal means".

15. P9L13 ... (cf. contour ...Response: We have revised "referred to contour lines" as "cf. contour lines".

**16. P13L5 and peaks in June**

**Response:** We have revised "and has peak in June" as "and peaks in June".

17. P16L22 Make it two sentences again: Due to the important role large vertical wind shears play in the dynamics and electrodynamics of the MLT, there is a need for global observations. In response to this need, a method ...

**Response:** Following your suggestion, we have revised the sentence "Due to the importance roles of the large shears in the MLT region on the atmospheric dynamics and electrodynamics, a method of..." as "Due to the important role the large vertical wind shears play in in the dynamics and electrodynamics of the MLT, there is a need for global observation. In response to this need, a method of...".

18. P16L24 of linear GWs. Data employed are SABER ...

**Response:** We have revised "... of a linear GW. The data are the SABER..." as "... of linear GWs. Data employed are SABER...".

19. P16L11 "new techniques" of analysis would not really help us. I think you have done what is possible with the current data. For new observations you could, if you like, be more explicit: ... develop new techniques for remote sensing temperatures or winds from space such as e.g. limb imaging techniques allowing to infer temperatures in 3D along the orbital track at high horizontal and vertical resolution. As said, if you like. Limb imaging techniques are mature, we need the opportunity to bring them to space and they would be a tremendous step forward.

[revised manuscript text omitted]

删除了:s 删除了:and

 删除了: a

 删除了: no

 删除了: linear

 删除了:, it has the form of

 $\left(u_{j}^{\prime},v_{j}^{\prime}|T_{j}^{\prime}/\overline{T}\right)=\left(\widetilde{u}_{j},\widetilde{v}_{j},\widetilde{T}_{j}\right)\times e^{i\varphi_{j}+z/2H},$

1

(1)

here,  $i = \sqrt{-1}$  is the imaginary unit. The subscript *j* denotes a monochromatic GW.  $u'_j$  and  $v'_j$  are the horizontal wind perturbations parallel and perpendicular to the wavevector of the GW, respectively.  $T'_j$  and  $\overline{T}$  are the perturbation and background temperatures, respectively.  $\tilde{u}_j$ ,  $\tilde{v}_j$  and  $\tilde{T}_j$  are the amplitudes of  $u'_j$ ,  $v'_j$  and  $T'_j/\overline{T}$ , respectively.  $\varphi_j = k_j x + l_j y +$

5  $m_j z - \omega_j t$  is the phase of GW.  $k_j$ ,  $l_j$  and  $m_j$  are the wavenumbers in the horizontal (x, y) and vertical (z) directions, respectively.  $\omega_j$  and t are the ground-based frequency and time.

Based on the polarization of the monochromatic linear GWs with lower and medium frequencies (Fritts & Rastogi, 1985; Fritts & Alexander, 2003), the relations between  $\tilde{u}_j$ ,  $\tilde{v}_j$  and  $\tilde{T}_j$  can be derived as (Eckermann et al., 1995; Liou et al., 2003; Gubenko et al., 2008),

10
$$\tilde{u}_{j} \approx i \frac{g}{N} \left( 1 - f^{2} / \hat{\omega}_{j}^{2} \right)^{-1/2} \tilde{T}_{j},$$
 (2)
 $\tilde{v}_{j} = \frac{f}{\hat{\omega}_{i} N} \left( 1 - f^{2} / \hat{\omega}_{j}^{2} \right)^{-1/2} \tilde{T}_{j}.$  (3)

here  $\hat{\omega}_j$  and  $f = 2\Omega \sin \phi$  ( $\Omega = 7.292 \times 10^{-5} \text{s}^{-1}$ ,  $\phi$  is latitude) are the intrinsic and inertial frequencies, respectively. The

wind shear of each monochromatic GW can be written as,

$$\frac{\partial u'_j}{\partial z} = \frac{g}{N} m_j \left(1 - f^2 / \hat{\omega}_j^2\right)^{-1/2} \tilde{T}_j e^{i(\varphi_j + \pi)}, \tag{4}$$

$$\frac{\partial v'_j}{\partial z} = \int_{-\infty}^{\infty} g_{ij} e^{-i(\varphi_j - \pi)} f_{ij} e^{i(\varphi_j + \pi)},$$

[revised manuscript text omitted]